# Switching Latent Bandits

**Alessio Russo**                                                             *alessio.russo@polimi.it*
*Department of Electronics, Information and Bioengineering*
*Politecnico di Milano*

**Alberto Maria Metelli**                                       *albertomaria.metelli@polimi.it*
*Department of Electronics, Information and Bioengineering*
*Politecnico di Milano*

**Marcello Restelli**                                               *marcello.restelli@polimi.it*
*Department of Electronics, Information and Bioengineering*
*Politecnico di Milano*

**Reviewed on OpenReview:** *https://openreview.net/forum?id=4ZGqCXcUqR*

## Abstract

We consider a Latent Bandit problem where the latent state keeps changing in time according to an underlying Markov chain, and every state is represented by a specific Bandit instance. At each step, the agent chooses an arm and observes a random reward but is unaware of which MAB he is currently pulling. As typical in Latent Bandits, we assume to know the reward distribution of the arms of all the Bandit instances. Within this setting, our goal is to learn the transition matrix determined by the Markov process. We propose a technique to tackle this estimation problem that results in solving a least-square problem obtained by exploiting the knowledge of the reward distributions and the properties of Markov chains. We prove the consistency of the estimation procedure, and we make a theoretical comparison with standard Spectral Decomposition techniques. We then discuss the dependency of the problem on the number of arms and present an offline method that chooses the best subset of possible arms that can be used for the estimation of the transition model. We ultimately introduce the SL-EC algorithm based on an Explore then Commit strategy that uses the proposed approach to estimate the transition model during the exploration phase. This algorithm achieves a regret of the order $\mathcal{O}(T^{2/3})$ when compared against an oracle that builds a belief representation of the current state using the knowledge of both the observation and transition model and optimizes the expected instantaneous reward at each step. Finally, we illustrate the effectiveness of the approach and compare it with state-of-the-art algorithms for non-stationary bandits and with a modified technique based on spectral decomposition.

## 1 Introduction

The Multi-Armed Bandit (MAB) (Lattimore & Szepesvári, 2020) framework is a well-known model used for sequential decision-making with little or no information. This framework has been successfully applied in a large number of fields, such as recommender systems, advertising, and networking. In the general MAB formulation, a learner sequentially selects an action among a finite set of different ones. The choice over the arm to select is made by properly balancing the exploration-exploitation trade-off with the goal of maximizing the expected total reward over a horizon $T$ and guaranteeing the *no-regret* property, thus meaning that the loss incurred by not knowing the best arm is increasing sublinearly over time. Standard MAB literature requires the payoff of the available actions to be stationary (i.e., rewards come from a fixed distribution) in order to design efficient no-regret algorithms.

However, in many real-life applications, the stationarity assumption may not necessarily hold as data may be subject to changes over time. In some applications, it is also possible to identify different data distributions,

each one corresponding to a specific working regime that can be modeled as a MAB. In cases of large availability of historical data appearing in the form of past user interactions, it is possible to learn *offline* the observation models associated with the different arms for each working regime. Exploiting the knowledge of observation models leads to many advantages over the *fully online exploration* setting where no prior information is available at the beginning, and a massive number of interactions is required to learn the observation models associated with each working regime. It is often the case that the underlying working regime (state) cannot be directly observed and the non-stationarity of the process is caused by the continuous change over time of the underlying regimes. By knowing how these regimes are characterized, it is possible to learn the dynamics of the changes by repeatedly interacting with the evolving environment. Inferring the underlying state accelerates the adaptation of the agent to the environment, thus leading to improved performances over time.

Learning observation models independently and prior to transition models can be a viable option when computational resources are limited. Indeed, we will show in the following that spectral decomposition (SD) techniques (Anandkumar et al., 2014), which are used to learn jointly the observation and the transition model, typically require a large number of samples and involve computationally intensive operations. Other scenarios where we can assume that the observation models are already known are those where the models are learned offline from samples generated by simulators. Once these models are deployed in an environment that is characterized by changes, the dynamics can be learned by interacting with the environment. We can consider, for example, the problem of resource allocation such as the electricity allocation in a specific residential area. This problem can be modeled as a Bandit where each arm represents a specific resource allocation, while the rewards represent the extent to which the allocation has been optimal. Obviously, the optimality of the allocation depends on the state of the system, which may be conditioned by several factors such as environmental conditions, community trends, and seasonality.

Another possible scenario that suits our setting is the one of *Transfer Learning*, where partial knowledge of the system (in our case, the observation model) can be used in a context with different dynamics (in this case the transition model, representing the dynamics of the system, needs to be learned). Building on the previously mentioned scenario, we can consider using the same observation models in a new residential area, with a structure analog to the first one (thus justifying the use of the same observation model) but located in a different place, with potentially different weather conditions and inhabitants having different behaviors (modeled using a different transition model).

Assuming the existence of a finite set of discrete latent states is a relevant choice when approaching the modeling of complex real-life problems characterized by different and recurrent working regimes. Regimes of this type can typically be observed in domains such as the financial market and online advertising, generally marked by high volatility and specific seasonality patterns (M. et al., 2022; Heston & Sadka, 2008; Guo et al., 2021).

Considering the financial market example, these regimes might include bull markets (characterized by rising prices), bear markets (characterized by falling prices), and periods of high or low volatility. The actual regime at any given time is not directly observable (hidden state), but we can infer it from observable data. In this example, the different actions available are the decisions whether to sell or buy different amounts of stocks and the observations can be the different returns, trading volumes, or stock prices.

Concerning the online advertising example, the hidden states can be represented by the interests of users, which are not directly observable. The interests may vary according to seasonal patterns, new trends, or exogenous variables that modify the market, and we can model it as a Markov chain. The actions are the different types of content that can be displayed to the users, while the observations may be represented by factors such as conversions or interactions with the ads (metrics such as the click-through rate could be considered).

Past works focused on this state identification problem under the assumption of knowing the conditional observation models (Maillard & Mannor, 2014; Zhou & Brunskill, 2016) and defined theoretically optimal UCB algorithms. Follow-up work by Hong et al. (2020a) provided more practical Thompson Sampling algorithms, also considering the problem of model misspecification and came up with an analysis on the Bayes regret.

The works cited above assume that the latent state does not change during the interaction process: once the real state is identified, the agent can act optimally. Differently, we embrace a more realistic scenario and assume that the latent state can change through time. In accordance with the latent bandits setting, we assume that the learning agent is aware of the observation models of the arms conditioned on each latent state. A setting similar to ours has been considered also in Hong et al. (2020b), the key difference is that they assume to have either full or partial knowledge of both the observation model and the transition model. We instead focus on the problem of learning the transition model given the knowledge of the observation models and maximizing the cumulative reward over $T$ interaction steps.

More specifically, our problem is modeled by assuming the existence of a finite set $\mathbb{S}$ of different MABs all sharing the same set of finite arms $\mathbb{I}$, each generating rewards (observations) in a finite set $\mathbb{V}$. Each state $s \in \mathbb{S}$ represents a different instance of a MAB. At each time step $t$, there is a transition from latent state $s_{t-1}$ to the new latent state $s_t$ according to the transition matrix governing the process. The action $a_t$ selected in $t$ will thus generate a reward conditioned on the latent state $s_t$.

**Contributions and Outline**  We introduce the Related Works in Section 2 and the Preliminaries in Section 3. After that, in Section 4, we define the formulation of the problem that considers known Bandit instances that switch through time according to an underlying Markov process. The information about the reward distributions of the bandit instances is encoded into a suitable observation matrix, while the transition model of the chain needs to be estimated. The learning objective of the agent is to maximize at each instant the expected instantaneous reward given the estimated belief over the current Bandit. After this part, we introduce the main assumptions that hold in our setting, motivate the reasons behind them, and show how they can be relaxed for the estimation of the transition model (Section 4.1).

Section 5.1 presents the estimation procedure of the transition model that uses samples collected using a round-robin procedure for selecting arms. Then, we propose an offline arm selection strategy that chooses a subset of the available arms for the estimation approach, with the objective of promoting diversity between observation distributions induced by the selected arms. In Section 5.2, we detail the SL-EC algorithm that employs an Explore then Commit approach and uses the proposed estimation procedure for learning the transition model during the exploration phase.

Finally, Section 7 shows numerical simulations on synthetic and semi-synthetic data. We provide additional experiments that highlight the difference in performance between our estimation procedure and a technique based on SD approaches. We complement the numerical simulation with further experiments in Appendix A while we present a comparison with SD approaches on the theoretical side in Appendix D.

## 2   Related Works

**Non-stationary Bandits**  Non-stationary behaviors are closer to real-world scenarios, and this has induced a vast interest in the scientific community leading to the formulation of different methods that consider either abruptly changing environments (Garivier & Moulines, 2011), smoothly changing environments (Trovò et al., 2020), or settings with a bounded variation of the rewards (Besbes et al., 2014). It is known that when rewards may arbitrarily change over time, the problem of Non-Stationary Bandits is intractable, meaning that only trivial bounds can be derived on the dynamic pseudo-regret. That is the main reason why in the literature, there is a large focus on non-stationary settings enjoying some specific structure in order to design algorithms with better guarantees. Non-stationary MAB approaches typically include both passive methods in which arm selection is mainly driven by the most recent feedback (Auer et al., 2019; Besbes et al., 2014; Trovò et al., 2020) and active methods where a change detection layer is used to actively perceive a drift in the rewards and to discard old information (Liu et al., 2017; Cao et al., 2018). Works such as Garivier & Moulines (2011) provide a $\mathcal{O}(\sqrt{T})$ regret guarantee under the assumption of knowing the number of abrupt changes. Other works, such as Besbes et al. (2014), employ a fixed budget to bound the total variation of expected rewards over the time horizon. They are able to provide a near-optimal frequentist algorithm with pseudo-regret $\mathcal{O}(T^{2/3})$ and a distribution-independent lower bound. All the above methods are not suited for environments that switch between different regimes as they do not keep in memory past interactions but rather tend to forget or discard the past.

A particular type of non-stationary Bandit problem related to our work includes the *restless Markov* setting (Ortner et al., 2014; Slivkins & Upfal, 2008) where each arm is associated with a different Markov process and the state of each arm evolves independently of the learner's actions. Differently, Fiez et al. (2018) investigate MAB problems with rewards determined by an unobserved Markov chain where the transition to the next state depends on the action selected at each time step, while Zhou et al. (2021) focus on MAB problems where the state transition dynamics evolves independently of the chosen action. This last work has many similarities with our setting. The main difference lies in the fact that they do not assume to know the conditional reward models and learn them jointly with the transition matrix. They make use of SD techniques (Anandkumar et al., 2014) and use this tool in a regret minimization algorithm achieving a $\mathcal{O}(T^{2/3})$ regret bound. Their setting is more complex than ours but involves additional assumptions, like the invertibility of the transition matrix that defines the chain. Furthermore, spectral methods need a vast amount of samples in order to provide reasonable estimation errors and can hardly be used in large problems. A detailed discussion on the differences between the estimation procedure used in Zhou et al. (2021) and ours is presented in Appendix D.

A setting similar to the one we consider has also been described in Kontorovich et al. (2013) in the context of *Hidden Markov Model*. They adopt an estimation approach of the transition model that shares some similarities with our estimation approach since they consider couples of consecutive observations, but their procedure is more involved and leads to worse convergence guarantees than ours.

**Latent Bandits**  More similar lines of work are related to bandit studies where latent variables determine the distribution of rewards (Maillard & Mannor, 2014; Zhou & Brunskill, 2016). In these works, the unobserved state is fixed across different rounds, and the conditional rewards depend on the latent state. Maillard & Mannor (2014) developed UCB algorithms without context, considering the two different cases in which the conditional rewards are either known or need to be estimated. This line of work has been extended to the contextual bandit case in Zhou & Brunskill (2016) where there is an offline procedure to learn the policies and a selection strategy to use them online. Hong et al. (2020a) proposed a TS procedure in the contextual case that updates a prior probability over the set of states in order to give a higher probability to the real latent state. A non-stationary variant of this setting is proposed in Hong et al. (2020b) where the latent states are assumed to change according to an underlying Markov chain. They develop TS algorithms under different cases when both the reward and transition models are completely known and when partial information about them is available. For the partial information case, they provide an algorithm based on particle filters, which will be used for comparison in the experimental section. Differently from Hong et al. (2020b), we do not assume any prior information about the transition matrix, and we learn it through interactions with the environment using the information about the reward models.

Another interesting work associated with latent bandits is the one from Kwon et al. (2022) where, differently from previously cited works, they assume an episodic setting with a fixed horizon $H$. At the beginning of each episode, a specific MAB instance is sampled from a fixed mixing distribution, and the agent interacts with the sampled MAB until the end of the episode without being aware of the MAB she is interacting with. The goal is to learn both the mixture weights and the reward distributions associated with each MAB. The relevant difference with our work is that they consider an episodic setting, while we consider a continuous one. Another main difference is that they provide results in terms of sample complexity needed in order to learn a near-optimal policy, not taking into account the suffered regret.

In Appendix C, we provide further discussion comparing our setting with related works and we report a table summarizing the differences with respect to the most related ones.

## 3   Preliminaries

In the following, we will present the main elements that are useful to understand what will follow. We will denote with $\Delta(\mathbb{X})$ the simplex of a finite space $\mathbb{X}$, and we will use the bold symbol $\boldsymbol{P}$ to denote the transition matrix and the probabilities associated with a Markov chain (see Section 3.2).

### 3.1 Multi-Armed Bandits

A *I*-armed stochastic bandit (Lattimore & Szepesvári, 2020) is a collection of distributions $\nu = (\Pr(\cdot|a) \in \Delta(\mathbb{V}) : a \in \mathbb{I})$ where $\mathbb{I}$ is the set of available actions with cardinality $I$ and $\mathbb{V}$ is a finite set of possible rewards with cardinality $V$. A learning agent sequentially interacts with the environment over $T$ rounds. For each round $t \in \{1, \ldots, T\}$, the learner chooses an action $a_t \in \mathbb{I}$ and the environment gives as output a reward $r_t \in \mathbb{V}$. The goal of the learner is to maximize the sum of cumulative rewards $\sum_{t=1}^{T} r_t$, which is a random quantity that depends on the stochasticity of both the environment and the choice of the agent's actions. The behavior of an agent interacting with an environment is defined by a policy $\theta : \mathcal{H} \to \Delta(\mathcal{A})$ that maps the observed history[1] to actions. In general, the performance of a bandit algorithm is measured using the notion of regret, which is defined as the deficit suffered by the learning agent with respect to the optimal policy. The regret of a policy $\theta$ on a bandit instance is defined as:

$$\mathcal{R}_T(\theta) = T\mu^* - \mathbb{E}\left[\sum_{t=1}^{T} r_t\right], \tag{1}$$

where $\mu^*$ defines the maximum expected reward among the available arms, while the expectation is taken with respect to the policy $\theta$ and the stochasticity of the environment.

### 3.2 Markov Chains

A Markov Chain (MC) (Feller, 1968) is defined by a tuple $\mathcal{M} := (\mathbb{S}, \boldsymbol{P}, \boldsymbol{\nu})$, where $\mathbb{S}$ is a finite state space ($|\mathbb{S}| = S$), $\boldsymbol{P} : \mathbb{S} \to \Delta(\mathbb{S})$ is the transition model, such that $\boldsymbol{P}(s, s')$ denotes the probability of reaching state $s' \in \mathbb{S}$ when being in state $s \in \mathbb{S}$, and $\boldsymbol{\nu}$ is the initial state distribution. We will denote with $\boldsymbol{P}$ the stochastic matrix of size $S \times S$ representing the transition model.

A Markov chain is said to be *ergodic* if its associated transition matrix consists of a single recurrent class containing all states (Puterman, 1994). Ergodic Markov chains satisfy the properties of being *irreducibile*, thus meaning that it is possible to reach any state from any other state with positive probability in a finite number of steps and *aperiodic*, meaning that the chain does not follow a regular, repeating pattern in its transitions.

We state the following result for ergodic Markov chains.

**Proposition 3.1.** *Let $\boldsymbol{P}$ be the transition matrix of an ergodic Markov Chain and $\boldsymbol{\nu}$ an arbitrary probability vector. Then:*

$$\lim_{n \to \infty} \boldsymbol{\nu} \boldsymbol{P}^n = \boldsymbol{\pi},$$

*where $\boldsymbol{P}^n$ represents the transition kernel induced after $n$ steps, $\boldsymbol{\pi}$ is the unique stationary distribution of the chain, and the components of the vector $\boldsymbol{\pi}$ are all strictly positive. By definition, this distribution satisfies the equation $\boldsymbol{\pi} \boldsymbol{P} = \boldsymbol{\pi}$.*

Since the stationary distribution of the Markov chain is unique, it follows that there is only one eigenvalue with unitary value ($\lambda_{\max} = 1$). Let's define the set of ordered moduli of the eigenvalues of transition matrix $\boldsymbol{P}$ as $(|\lambda_i|)_{i=1}^{S}$. By denoting $|\lambda_{\max}| = |\lambda_1|$, we have the following relation:

$$|\lambda_1| = 1 > |\lambda_2| \geq \cdots \geq |\lambda_S|,$$

where the inequality between $|\lambda_1|$ and $|\lambda_2|$ is strict for ergodic chains. The quantity $1 - |\lambda_2|$ is defined as the absolute spectral gap of the Markov chain and controls the rate of convergence of the chain towards its stationary distribution (Krishnamurthy, 2016). In what will follow, we will use the symbol $\lambda$ to denote the modulus of the second largest eigenvalue, namely $\lambda = |\lambda_2|$.

---

[1]We define a history $h := (a_j, r_j)_{j=0}^{t} \in \mathcal{H}_t$ with $\mathcal{H}_t$ being the space of histories of length $t$ and we denote instead with $\mathcal{H}$ the space of the histories of arbitrary length.

## 4 Switching Latent Bandits

We consider to have a finite set $\mathbb{S} := \{s_1, \ldots, s_S\}$ of $S = |\mathbb{S}|$ different MAB instances. Each MAB is characterized by the same set of discrete arms $\mathbb{I} := \{a_1, \ldots, a_I\}$ with cardinality $I = |\mathbb{I}|$ and the same set of finite rewards $\mathbb{V} = \{r_1, \ldots, r_V\}$ with cardinality $V = |\mathbb{V}|$. Whenever an arm $a \in \mathbb{I}$ is pulled, a corresponding reward $r \in \mathbb{V}$ is generated by the environment. We consider each reward $r \in \mathbb{V}$ bounded for simplicity in the range $[0, 1]$. The distribution of rewards $\Pr(\cdot|s, a)$ conditioned on MAB instance $s$ and action $a$ is categorical[2]. In particular, we assume to know the parameters characterizing these distributions and to store this information into matrix $\boldsymbol{O} \in \mathbb{R}^{IV \times S}$, which we call action observation matrix. Each row of this matrix encodes a specific action-reward pair $(a, r) \in \mathbb{I} \times \mathbb{V}$. Then, for any pair $(a, r) \in \mathbb{I} \times \mathbb{V}$ and any state $s \in \mathbb{S}$, we have:

$$\boldsymbol{O}\big((a, r), s\big) = \Pr(r|s, a), \tag{2}$$

where $\Pr(r|s, a)$ represents the probability value of observing reward $r$ while pulling action $a$ from MAB $s$. At each step $t$, only one MAB $s_t \in \mathbb{S}$ is active, and it determines the reward $r_t$ that is received when the agent pulls action $a_t$. The choice over the active MAB is determined by an underlying Markov chain with transition matrix $\boldsymbol{P} \in \mathbb{R}^{S \times S}$. More precisely, the probability over the next active MAB $s_{t+1}$ is determined by the distribution $\boldsymbol{P}(s_t, \cdot) \in \Delta(\mathbb{S})$ and is thus independent of the chosen action $a_t$. The setting we consider assumes that the agent is not able to observe the active MAB at each step, and the objective is to learn the transition matrix $\boldsymbol{P}$ characterizing the underlying process while knowing the observation model $\boldsymbol{O}$.

**Learning objective** As already seen, the agent does not observe the sequence of MAB instances, but by deriving an estimate of the transition matrix $\boldsymbol{P}$, a belief representation over the current active MAB $s \in \mathbb{S}$ can be defined. In the following, we will report the update rule of the belief vector $\mathbf{b}_t \in \Delta(\mathbb{S})$ under the knowledge of the observation model $\boldsymbol{O}$ and the transition model $\boldsymbol{P}$. The update of the belief derives from the typical correction and update step of the Bayes rule, where the correction step adjusts the current belief $\mathbf{b}_t$ using the reward $r_t$ obtained by pulling arm $a_t$ and the prediction step computes the new belief $\mathbf{b}_{t+1}$ simulating a transition step of the chain. More formally, for each element $\mathbf{b}_{t+1}(s)$ of the belief vector $\mathbf{b}_{t+1}$, the update step is as follows:

$$\mathbf{b}_{t+1}(s) = \frac{\sum_{s' \in \mathbb{S}} \mathbf{b}_t(s') \boldsymbol{O}\big((a_t, r_t), s'\big) \boldsymbol{P}(s', s)}{\sum_{s'' \in \mathbb{S}} \mathbf{b}_t(s'') \boldsymbol{O}\big((a_t, r_t), s''\big)}. \tag{3}$$

After having defined the update rule of the belief vector $\mathbf{b}_t$, we introduce, for each action $a \in \mathbb{I}$, vector $\boldsymbol{\mu}(a) \in \mathbb{R}^S$ where element $\boldsymbol{\mu}(a, s)$ referred to state $s \in \mathbb{S}$ contains the expected reward obtained when pulling arm $a$ while being in state $s$. More formally:

$$\boldsymbol{\mu}(a, s) = \sum_{r \in \mathbb{V}} r \, \boldsymbol{O}\big((a, r), s\big). \tag{4}$$

Given the belief $\mathbf{b}_t$ over the states, the objective of the agent is to pull the action that maximizes the instantaneous expected reward such that:

$$a_t = \arg\max_{a \in \mathbb{I}} \sum_{s \in \mathbb{S}} \boldsymbol{\mu}(a, s) \mathbf{b}_t(s) = \arg\max_{a \in \mathbb{I}} \langle \boldsymbol{\mu}(a), \mathbf{b}_t \rangle, \tag{5}$$

where $\langle \cdot, \cdot \rangle$ denotes the scalar product between the two vectors.
From the considerations reported above, we are now ready to formulate the notion of regret we try to minimize:

$$\mathfrak{R}_T = \sum_{t=1}^{T} \max_{a \in \mathbb{I}} \langle \boldsymbol{\mu}(a), \mathbf{b}_t \rangle - \max_{a \in \mathbb{I}} \langle \boldsymbol{\mu}(a), \widehat{\mathbf{b}}_t \rangle, \tag{6}$$

where $\mathbf{b}_t$ and $\widehat{\mathbf{b}}_t$ denote the belief vectors updated using the real transition matrix $\boldsymbol{P}$ and the estimated one $\widehat{\boldsymbol{P}}$ respectively. Here, we used symbol $\mathfrak{R}$ to characterize the regret defined in Equation 6, in order to discriminate from the standard notion of regret $\mathcal{R}$ introduced in Section 3.1.

---

[2]In Appendix F, we will see how this formulation can be extended to continuous distributions.

### 4.1 Assumptions

We need now to introduce some assumptions that should hold in our setting:

**Assumption 4.1.** *The smallest element of the transition matrix defining the Markov chain is $\epsilon :=$ $\min_{s,s' \in \mathbb{S}} \boldsymbol{P}(s, s') > 0$.*

This assumption ensures a non-null probability of transitioning from any state to any other in one step. It is possible to show that under this assumption, the induced Markov chain is ergodic, thus guaranteeing the presence of a unique stationary distribution, as shown in Proposition 3.1. Under the ergodic condition, the chain is able to reach its stationary distribution $\boldsymbol{\pi}$ geometrically fast, regardless of its initial distribution Krishnamurthy (2016). Our assumption on the minimum entry is not a necessary condition for the two aforementioned motivations but a sufficient one. However, we require this condition to bound the error between the belief computed using the real transition matrix and an estimated one. This result is presented in Proposition E.6 and builds on the original result present in De Castro et al. (2017). This one-step reachability assumption is always present in works dealing with partial observability that show results in terms of regret in non-episodic scenarios. Notably, it has been used in similar works such as Zhou et al. (2021); Jiang et al. (2023); Mattila et al. (2020) and also employed in the more complex POMDP setting (Xiong et al., 2022). Works not using this assumption either do not need it since they use a less powerful class of policies such as memoryless ones[3] (Azizzadenesheli et al., 2016) or they directly impose an error of the estimated belief that adequately decreases with the number of collected samples (Jafarnia Jahromi et al., 2022).

**Assumption 4.2.** *The action observation matrix $\boldsymbol{O} \in \mathbb{R}^{IV \times S}$ is full column rank.*

This second assumption, instead, is related to the identifiability of the parameters of the problem and has been largely used in works using spectral decomposition techniques (Zhou et al., 2021; Azizzadenesheli et al., 2016; Hsu et al., 2012). A robust version of this assumption, called weakly-revealing[4] is also present in other works involving learning parameters in POMDPs (Liu et al., 2022; Jin et al., 2020). In the following, we will see that this is a necessary condition in order to recover matrix $\boldsymbol{P}$. Indeed, we will see that the error of the estimation procedure has an inverse dependency on the minimum singular value $\sigma_{\min}(\boldsymbol{O})$ of the action observation matrix $\boldsymbol{O}$ and through Assumption 4.2, we implicitly require that $\sigma_{\min}(\boldsymbol{O}) > 0$.

## 5 Proposed Approach

As clarified in the previous section, our goal is to minimize the regret formulated in Equation 6. To reach this objective, we need to define a good estimate $\widehat{\boldsymbol{P}}$ of the transition matrix that in turn results in a more accurate update of the belief vector $\widehat{\mathbf{b}}_t$. We will now show how the transition model can be learned by exploiting the knowledge of the observation model $\boldsymbol{O}$ (Section 5.1) and we will present the SL-EC algorithm that makes use of the presented estimation approach in order to minimize the regret (Section 5.2).

### 5.1 Transition Model Estimation

The Markov chain estimation procedure presented in this section holds under weaker assumptions than those presented in Section 4.1. In particular, we relax the one-step reachability assumption (Assumption 4.1) and we only require the ergodicity of the transition matrix $\boldsymbol{P}$.

**Stationary Distribution of Consecutive States** We start with a consideration about the transition matrix that defines the chain. Building on Proposition 3.1, an ergodic chain admits a unique stationary distribution $\boldsymbol{\pi}$.
From the uniqueness of this distribution, it can be easily shown that there exists as well a unique stationary distribution on consecutive states that we represent with a matrix $\boldsymbol{W} \in \Delta(\mathbb{S}^2)$ having dimension $S \times S$.

---

[3]A memoryless policy defines the action to choose only based on the last observation seen. For this reason, it does not require a notion of belief over the states.

[4]The $\alpha$-weakly revealing assumption defines a lower bound $\alpha$ to the minimum singular value of the observation matrix $\boldsymbol{O}$, such that $\sigma_{\min}(\boldsymbol{O}) \geq \alpha$.

Its elements are obtained as $\boldsymbol{W}(s, s') = \pi(s)\boldsymbol{P}(s, s')$. By defining with $\boldsymbol{\Pi} = diag(\boldsymbol{\pi})$ the diagonal matrix of size $S \times S$ having values of the stationary distribution $\boldsymbol{\pi}$ along its diagonal, we can express matrix $\boldsymbol{W}$ of the stationary distribution of consecutive states as follows:

$$\boldsymbol{W} = \boldsymbol{\Pi}\boldsymbol{P},$$

which is obtained by multiplying each row of the transition matrix $\boldsymbol{P}$ by the associated probability value of the stationary distribution. The reverse procedure that allows retrieving matrix $\boldsymbol{P}$ from $\boldsymbol{W}$ is defined by the following equation:

$$\boldsymbol{P}(s, s') = \frac{\boldsymbol{W}(s, s')}{\sum_{s'' \in \mathbb{S}} \boldsymbol{W}(s, s'')}, \tag{7}$$

which shows that the rows of matrix $\boldsymbol{P}$ are obtained by normalizing the rows of matrix $\boldsymbol{W}$ such that they sum to 1, as required for stochastic matrices.

The next paragraph shows how the matrix $\boldsymbol{W}$ of stationary distribution of consecutive states relates to the stationary distribution of consecutive rewards.

**Stationary Observation-State Relation**   Let's choose an arm $a \in \mathbb{I}$: we will denote with $\boldsymbol{d}_a \in \Delta(\mathbb{V})$ the stationary distribution of rewards conditioned on pulling action $a$ when the chain has mixed[5]. Vector $\boldsymbol{d}_a$ has dimension $V$ and its elements are characterized as follows:

$$\boldsymbol{d}_a(r) = \sum_{s \in \mathbb{S}} \boldsymbol{O}\big((a, r), s\big)\boldsymbol{\pi}(s) \qquad \forall\, r \in \mathbb{V}, \tag{8}$$

where we recall that $\boldsymbol{\pi}(s)$ represents the probability of state $s$ taken from the stationary distribution of the chain and $\boldsymbol{O}\big((a, r), s\big)$ represents the probability of observing reward $r$ while pulling action $a$ and being in state $s$. A similar rationale can be extended to consecutive rewards $(r, r') \in \mathbb{V}^2$ conditioned on pulling a couple of consecutive actions $(a, a') \in \mathbb{I}^2$. We denote with $\boldsymbol{d}_{a,a'} \in \Delta(\mathbb{V}^2)$ the distribution over consecutive rewards conditioned on pulling the pair of arms $(a, a')$. We represent it with a vector of size $V^2$ and define it as follows:

$$\boldsymbol{d}_{a,a'}((r, r')) = \sum_{s, s' \in \mathbb{S}^2} \boldsymbol{O}\big((a, r), s\big)\,\boldsymbol{O}\big((a', r'), s'\big)\,\boldsymbol{W}(s, s') \qquad \forall\, r, r' \in \mathbb{V}, \tag{9}$$

where we recall that matrix $\boldsymbol{W} \in \Delta(\mathbb{S}^2)$ represents the stationary distribution of consecutive states.

By considering the different vectors of type $\boldsymbol{d}_{a,a'}$, we define vector:

$$\boldsymbol{d} = \left(\boldsymbol{d}_{a,a'}\right)_{(a,a') \in \mathbb{I}^2} \tag{10}$$

where the term on the right denotes a concatenation of vectors $\boldsymbol{d}_{a,a'}$ for all $(a, a') \in \mathbb{I}^2$ and the resulting vector $\boldsymbol{d}$ has size $I^2V^2$.

We define now a new matrix $\boldsymbol{A} \in \mathbb{R}^{I^2V^2 \times S^2}$ to which we will refer as reference matrix. It extends the information contained in the action observation matrix $\boldsymbol{O}$ considering consecutive pairs of elements and it is characterized as follows:

$$\boldsymbol{A} = \boldsymbol{O} \otimes \boldsymbol{O}, \tag{11}$$

where symbol $\otimes$ refers to the Kronecker product (Loan, 2000). Since we assume knowledge of the observation model $\boldsymbol{O}$, we can directly compute the reference matrix by applying the Kronecker operator.

As a last step before presenting the main result, we transform matrix $\boldsymbol{W}$ and vectorize[6] it to obtain vector $\boldsymbol{w} \in \Delta(\mathbb{S}^2)$ having dimension $S^2$. By using the quantities just defined, we can finally reformulate Equation 9 so that it can be extended to all pairs of actions. Using vector notation, we have:

$$\boldsymbol{d} = \boldsymbol{A}\boldsymbol{w}. \tag{12}$$

---

[5]The distribution of states in a mixed chain corresponds by definition to its stationary distribution $\boldsymbol{\pi}$.

[6]The vectorization operation used here creates a new vector $\boldsymbol{w}$ by concatenating each row of matrix $\boldsymbol{W}$.

Basically, this equation relates the stationary probability distribution of consecutive observations with the stationary probability distribution of consecutive latent states. The next paragraph shows how to obtain an estimate $\widehat{\boldsymbol{d}}$ of vector $\boldsymbol{d}$, from which, by reversing Equation 12, an estimate $\widehat{\boldsymbol{w}}$ of the stationary distribution of consecutive states can be computed.

**Transition Model Estimation**  We will now see how to concretely compute an estimate of $\boldsymbol{w}$ using Equation 12. Going back to vectors $\boldsymbol{d}_{a,a'}$, we can build an estimate $\widehat{\boldsymbol{d}}_{a,a'}$ for each pair of actions $(a, a') \in \mathbb{I}^2$. For this purpose, let's take a pair of action $(a, a')$ and repeatedly pull it. We can count the number of occurrences of each pair of observed rewards $(r, r') \in \mathbb{V}^2$ and store this information in a suitable count vector $\boldsymbol{n}_{a,a'}$ of size $V^2$. We can easily derive an estimate of vector $\boldsymbol{d}_{a,a'}$ as follows:

$$\widehat{\boldsymbol{d}}_{a,a'} = \frac{\boldsymbol{n}_{a,a'}}{N}, \tag{13}$$

where $N$ represents the number of times the pair of consecutive arms $(a, a')$ has been pulled.
We propose an estimation procedure that pulls each pair of arms $(a, a') \in \mathbb{I}^2$ in a round-robin fashion and stores the observed pairs of rewards in the corresponding vector count $\boldsymbol{n}_{a,a'}$. The choice of a round-robin approach highlights interesting properties in the theoretical analysis, as will be shown later in Section 6. By executing $N$ different rounds, thus meaning that each pair of arms is pulled exactly $N$ times and by exploiting the knowledge of the reference matrix $\boldsymbol{A}$, we can derive:

$$\widehat{\boldsymbol{w}} = \boldsymbol{A}^{\dagger} \widehat{\boldsymbol{d}} = \boldsymbol{A}^{\dagger} \frac{\boldsymbol{n}}{N}, \tag{14}$$

where $\boldsymbol{A}^{\dagger}$ is the Moore–Penrose inverse of reference matrix $\boldsymbol{A}$, while vectors $\widehat{\boldsymbol{d}}$ and $\boldsymbol{n}$ are obtained by concatenating the different vectors $\widehat{\boldsymbol{d}}_{a,a'}$ and $\boldsymbol{n}_{a,a'}$ as also done in Equation 10. The second equality is derived by extending Equation 13 to the concatenated vectors. The stated equation shows that the estimation procedure involves solving a simple least-square problem, which can be done in a computationally efficient way.
Once an estimate $\widehat{\boldsymbol{w}}$ is computed, the corresponding matrix $\widehat{\boldsymbol{W}}$ can be obtained by reverting the vectorization operation and eventually an estimate $\widehat{\boldsymbol{P}}$ of the transition model is computed using Equation 7.
The pseudocode of the presented estimation procedure is detailed in Algorithm 1.

We acknowledge that other approaches can be devised for choosing the action policy used during estimation. Some approaches have been devised for the Latent Bandit setting such as the one in Kinyanjui et al. (2023) which face a pure exploration problem. However, their method is tailored for the stationary setting and they update their policy as new information is acquired. Instead, in our scenario, it is necessary to select a prior an action policy and keep it constant during the interaction with the environment: approaches that work in this direction and that could be potentially used in our setting are those based on Experimental Design (Kiefer & Wolfowitz, 1960).

### 5.1.1 Arm Selection Strategy

In Algorithm 1, we propose a simple approach for choosing the arms to pull. Each pair of arms is indeed pulled the same number of times during the exploration phase by using a deterministic approach. However, it can be shown that the estimation procedure proposed in Section 5.1 can be extended to a more flexible arm selection policy. We may randomize the arm choice by assigning non-uniform probabilities to each pair of arms. In principle, this aspect allows exploiting the knowledge of the known reward distribution of each arm, for example, giving at the beginning a higher probability to the pairs of arms that are more rewarding. For example, this arm selection policy may be beneficial if we plug our estimation approach into an iterative two-phase exploration and exploitation algorithm, as the one used in Zhou et al. (2021).

**Offline arm selection**  In problems with a large number of available arms, a round-robin approach among all possible pairs of arms may be detrimental as it considers all arms equivalently. There may be cases where some actions are less useful for state identification. The extreme case is an action that induces the same observation distribution for all the Bandit instances. Indeed, pulling that action will not provide any additional information on the current MAB and the effect will only be to slow down the estimation

---

**Algorithm 1:** Estimation Procedure

**Input:** Action Observation matrix $\boldsymbol{O}$, number of rounds $N$

**1** Build Reference matrix $\boldsymbol{A}$ using Equation 11
**2** Initialize vector of counts $\boldsymbol{n}_{a,a'}$ with zeroes for all $(a, a') \in \mathbb{I}^2$
**3** $k = 0$
**4 while** $k < N$ **do**
**5**      $t = k * I^2$
**6**      **foreach** $(a, a') \in \mathbb{I}^2$ **do**
**7**          Pull arm $a_t = a$
**8**          Observe reward $r_t = r$
**9**          Pull arm $a_{t+1} = a'$
**10**         Observe reward $r_{t+1} = r'$
**11**         $\boldsymbol{n}_{a,a'}(r, r') = \boldsymbol{n}_{a,a'}(r, r') + 1$
**12**         $t = t + 2$

**13** Compute $\widehat{\boldsymbol{d}}_{a,a'}$ for all $(a, a') \in \mathbb{I}^2$ using Equation 13
**14** Obtain $\widehat{\boldsymbol{d}}$ by concatenating all the different $\widehat{\boldsymbol{d}}_{a,a'}$ (as done in 10)
**15** Estimate $\widehat{\boldsymbol{w}}$ from Equation 14
**16** Reshape vector $\widehat{\boldsymbol{w}}$ to obtain matrix $\widehat{\boldsymbol{W}}$
**17** Compute $\widehat{\boldsymbol{P}}$ using Equation 7

---

procedure. In general, actions that induce *similar* observation distributions for all the MABs will provide *less* information with respect to actions that induce highly different distributions for all the MABs.

A more convenient approach, in this case, would be to select a subset of different arms to be used during the exploration phase. Intuitively, the arm selection procedure tends to promote diversity among arms conditioned on the latent states, with the objective of increasing the identifiability capabilities deriving from the actions. It turns out that we are able to get an understanding of the information loss we suffer by selecting specific arms, given the knowledge of the action observation matrix $\boldsymbol{O}$. In particular, in Section 6 devoted to the theoretical analysis, we will see that the quality of the estimation highly depends on the minimum singular value $\sigma_{\min}(\boldsymbol{O})$ of the action observation matrix $\boldsymbol{O}$. We can thus use this value to drive the choice of the best subset of arms to use.

In particular, by fixing a number $J < I$ of arms to use among those available, the choice over the best subset of size $J$ can be done as follows. We consider all the possible subsets of arms of size $J$, and for each of these subsets, we derive a reduced action observation matrix $\boldsymbol{G}$ of size $JV \times S$ that is obtained by simply removing from the original matrix $\boldsymbol{O}$ all the rows associated to the actions not belonging to the considered subset of arms. Having defined a new action observation matrix for each generated subset, a good candidate subset of arms is the one yielding the reduced action observation matrix $\boldsymbol{G}$ with the highest $\sigma_{\min}(\boldsymbol{G})$. Understandably, this approach implies that the reduced action observation matrix $\boldsymbol{G}$ derived from the subset of selected arms should be full-column rank, thus satisfying Assumption 4.2.

### 5.2 SL-EC Algorithm

Having established an estimation procedure for the transition matrix $\widehat{\boldsymbol{P}}$, we will now provide an algorithm that makes use of this approach in a regret minimization framework.

We consider a finite horizon $T$ for our problem. We propose an algorithm called *Switching Latent Explore then Commit* (SL-EC) that proceeds using an EC approach where the exploration phase is devoted to finding the best estimation of the transition matrix $\widehat{\boldsymbol{P}}$, while during the exploitation phase, we maximize the instantaneous expected reward following the formulation provided in Equation 5. The exploration phase lasts for $T_0$ episodes, where $T_0$ is optimized w.r.t. the total horizon $T$, as will be seen in Section 6. The pseudocode of the SL-EC Algorithm is presented in Algorithm 2.

---

**Algorithm 2:** SL-EC Algorithm

---
**Input:** Observation model $\boldsymbol{O}$, Exploration horizon $T_0$, Total horizon $T$

**1** Define number of rounds $N = T_0/2I^2$

**2** $\widehat{\boldsymbol{P}} = EstimationProcedure(\boldsymbol{O}, N)$ (Algorithm 1)

**3** $\mathbf{b}_0 = UniformOverStates()$

**4** Compute $\widehat{\mathbf{b}}_{T_0}$ using samples collected during Algorithm 1

**5** $t \leftarrow T_0$

**6 while** $t \leq T$ **do**

**7**     $a_t = \arg\max_{a \in \mathbb{I}} \langle \boldsymbol{\mu}(a), \widehat{\mathbf{b}}_t \rangle$

**8**     Observe reward $r_t$

**9**     $\widehat{\mathbf{b}}_{t+1} = UpdateBelief(\widehat{\mathbf{b}}_t, a_t, r_t)$ (Equation 3)

**10**     $t = t + 1$

---

Basically, the exploration phase pulls each pair of arms in a round-robin fashion and uses the estimation procedure presented in Algorithm 1. When the exploration phase is over, an estimation of the transition matrix $\widehat{\boldsymbol{P}}$ is computed. After that, a belief vector $\mathbf{b}_0$ is initialized by assigning a uniform probability to all the states (Line 3), and it is updated using Equation 3 and the estimated $\widehat{\boldsymbol{P}}$, considering the history of samples collected from the beginning up to $T_0$ (Line 4). Finally, the exploitation phase starts, as described in the pseudocode of the algorithm.

## 6 Theoretical Analysis

Having defined the estimation procedure of the transition model in Section 5.1 and having introduced the SL-EC algorithm, we will now provide theoretical guarantees for them.

### 6.1 Analysis of Estimation Procedure in Algorithm 1

We start with a concentration bound on the transition matrix $\widehat{\boldsymbol{P}}$ computed from the estimation procedure in Algorithm 1. As already highlighted, this estimation procedure only requires the ergodicity of the chain, thus relaxing Assumption 4.1.

**Lemma 6.1.** *Suppose Assumption 4.2 holds and suppose that the Markov chain with transition matrix $\boldsymbol{P}$ is ergodic, such that $\pi_{\min} := \min_{s \in \mathbb{S}} \boldsymbol{\pi}(s) > 0$ with $\boldsymbol{\pi} \in \Delta(\mathbb{S})$ being the stationary distribution of the chain. By assuming that the chain starts from an arbitrary distribution $\boldsymbol{\nu} \in \Delta(\mathbb{S})$, by pulling each pair of arms in a round-robin fashion for $N$ rounds and using the estimation procedure reported in Algorithm 1, we have that with probability at least $1 - \delta$ the estimation error of the transition matrix $\boldsymbol{P}$ will be:*

$$\|\boldsymbol{P} - \widehat{\boldsymbol{P}}\|_F \leq \frac{2I}{\sigma_{\min}^2(\boldsymbol{O})\pi_{\min}} \sqrt{\frac{2S(C + \log(CI^2/\delta))}{(1 - \lambda^{2I^2})N}}, \tag{15}$$

where $\|\cdot\|_F$ represents the Frobenius norm (Golub & Van Loan, 1996), $\sigma_{\min}(\boldsymbol{O})$ represents the minimum singular value of the action observation matrix $\boldsymbol{O}$, constant $C$ is defined as $C := \|\frac{\boldsymbol{\nu}}{\boldsymbol{\pi}}\|_\infty$ where $\frac{\boldsymbol{\nu}}{\boldsymbol{\pi}}$ represents the vector of the element-wise ratio between the two probability distributions, while $\lambda$ represents the modulus of the second highest eigenvalue of matrix $\boldsymbol{P}$. As reported in the statement of the Lemma, $N$ denotes the number of times each pair of arms is pulled, thus meaning that the stated error guarantee holds when interacting with the environment for a total number of $2I^2N$ steps, where the $I^2$ term arises from the total number of pairs of arms while the constant value 2 accounts for considering pairs of arms.

As a last remark, we note that the presented Lemma assumes that the chain starts from an arbitrary distribution. This fact leads to the further constant $C$ in the bound. Indeed when the chain starts from the stationary distribution, we have $C = 1$. This result comes from Proposition E.4 that uses a concentration result derived from Fan et al. (2021).

Here, we will provide a sketch of the proof of the presented Lemma. A more detailed version of this proof is reported in Appendix B.

***Sketch of the proof.*** The proof of Lemma 6.1 builds on two principal results. The former comprises a relation that links the estimation error of the matrix $\boldsymbol{P}$ with the estimation error of matrix $\boldsymbol{W}$, while the latter is a concentration bound on the estimated $\widehat{\boldsymbol{W}}$ from the true one $\boldsymbol{W}$. Concerning the first result, we can say that:

$$\|\boldsymbol{P} - \widehat{\boldsymbol{P}}\|_F \leq \frac{2\sqrt{S}\|\boldsymbol{W} - \widehat{\boldsymbol{W}}\|_F}{\pi_{\min}}. \tag{P.1}$$

This result follows from a sequence of algebraic manipulations and makes use of Lemma E.1 appearing in Appendix E.

We now need to define a bound on $\|\boldsymbol{W} - \widehat{\boldsymbol{W}}\|_F$. In order to bound this quantity, we resort to the vectorized versions $\boldsymbol{w}$ and $\widehat{\boldsymbol{w}}$ of the two matrices and use the result $\|\boldsymbol{W} - \widehat{\boldsymbol{W}}\|_F = \|\boldsymbol{w} - \widehat{\boldsymbol{w}}\|_2$. We proceed as follows:

$$
\begin{aligned}
\|\boldsymbol{w} - \widehat{\boldsymbol{w}}_{T_0}\|_2 &= \left\|\boldsymbol{A}^\dagger(\boldsymbol{d} - \widehat{\boldsymbol{d}})\right\|_2 \\
&\leq \|\boldsymbol{A}^\dagger\|_2\|\boldsymbol{d} - \widehat{\boldsymbol{d}}\|_2 \\
&= \frac{1}{\sigma_{\min}(\boldsymbol{A})}\|\boldsymbol{d} - \widehat{\boldsymbol{d}}\|_2 = \frac{1}{\sigma_{\min}^2(\boldsymbol{O})}\|\boldsymbol{d} - \widehat{\boldsymbol{d}}\|_2,
\end{aligned} \tag{P.2}
$$

where the first equality follows from Equation 14. In the inequality instead, we used the consistency property for the spectral norm of matrix $\boldsymbol{A}^\dagger$, while in the last equality we used a property of the Kronecker product for which it holds that:

$$\sigma_{\min}(\boldsymbol{A}) = \sigma_{\min}(\boldsymbol{O})\sigma_{\min}(\boldsymbol{O}) = \sigma_{\min}^2(\boldsymbol{O}).$$

Let's now consider the estimation error of each vector $\boldsymbol{d}_{a,a'}$ that represents the stationary distribution over consecutive rewards conditioned on pulling the pair of arms $(a, a')$. From Equation 10, we know that by concatenating each of these vectors, we obtain the quantity $\boldsymbol{d}$. Thus, by definition, we have:

$$\|\boldsymbol{d} - \widehat{\boldsymbol{d}}\|_2 = \sqrt{\sum_{(a,a')\in\mathbb{I}^2} \|\boldsymbol{d}_{a,a'} - \widehat{\boldsymbol{d}}_{a,a'}\|_2^2}. \tag{P.3}$$

The estimation error of each $\boldsymbol{d}_{a,a'}$ can be bounded by using a result shown in Proposition E.4 and inspired by the work of Hsu et al. (2012). It bounds the estimation error of categorical distributions when the observed samples derive from a Markov chain. With probability at least $1 - \delta/I^2$ we have that:

$$\|\boldsymbol{d}_{a,a'} - \widehat{\boldsymbol{d}}_{a,a'}\|_2 \leq \sqrt{\left(\frac{1 + \lambda^{2I^2}}{1 - \lambda^{2I^2}}\right)\frac{C + \log(CI^2/\delta)}{N}}.$$

The exponential term $2I^2$ that appears to the modulus of the second highest eigenvalue $\lambda$ has been introduced thanks to the adoption of the round-robin procedure for the choice of combinations of arms. Notably, each pair of arms is pulled every $2I^2$ steps of the Markov Process, thus resulting in a faster mixing of the chain. For more details, please refer to Appendix B.

By combining the last obtained bound with P.2 and P.3 and using a union bound for the estimation error of all vectors of type $\boldsymbol{d}_{a,a'}$, we have that with probability at least $1 - \delta$:

$$
\begin{aligned}
\|\boldsymbol{w} - \widehat{\boldsymbol{w}}\|_2 &\leq \frac{1}{\sigma_{\min}^2(\boldsymbol{O})}\sqrt{\left(\frac{1 + \lambda^{2I^2}}{1 - \lambda^{2I^2}}\right)\frac{I^2(C + \log(CI^2/\delta))}{N}} \\
&\leq \frac{I}{\sigma_{\min}^2(\boldsymbol{O})}\sqrt{\frac{2\left(C + \log(CI^2/\delta)\right)}{(1 - \lambda^{2I^2})N}}.
\end{aligned}
$$

Ultimately, by putting together the bound in P.1 with the one just obtained, we are able to obtain the final result stated in the Lemma. □

**Dependency on the Problem Parameters**   Lemma 6.1 shows the dependencies of the result obtained from the proposed estimation approach. The estimation error scales almost linearly with the number of arms $I$. This may seem concerning when dealing with problems involving a high number of arms. However, we have already observed in Section 5.1.1 that when the number of arms is large, it is possible to reduce this dependency by using the arm selection strategy detailed in Section 5.1.1. We further have a dependency on the minimum value of the induced stationary distribution $\pi_{\min}$, which is common in the partially observable setting.

Concerning instead the dependency on the minimum singular value $\sigma_{\min}(\boldsymbol{O})$, it is related to Assumption 4.2. It defines the amount of identifiability of different states given the information provided by the observations. This dependency characterizes the class of weakly-revealing POMDPs and is unavoidable in order to have a tractable problem (Chen et al., 2023).

The dependency on the modulus of the second largest eigenvalue $\lambda$ derives from the fact that observed samples are not independent but come from a Markov chain. The dependency on this term is thus unavoidable and we believe it to be tight for the considered setting. It derives from recent results from the work of Fan et al. (2021) which improves over the existing concentration results of samples coming from Markov Chains.

For a thorough comparison of our estimation approach with standard Spectral Decomposition techniques, we refer to Appendix D.

**Continuous Reward Distributions**   The presented setting tackles the case of discrete observations. It appears that handling continuous reward distributions within this framework is not feasible and this is true if we apply our approach as is. However, we can discretize the observation distributions and consider the discretized distribution as a categorical one. The process of discretization involves dividing the continuous observation distributions into a predetermined number $U$ of distinct consecutive intervals. Each interval is assigned a probability value that represents the likelihood of a sample drawn from the continuous distribution and belonging to that interval. Throughout this discretization procedure, we can define an action observation matrix of dimension $IU \times S$ and then apply Algorithm 1. More details on this aspect can be found in Appendix F.

## 6.2   Analysis of the SL-EC Algorithm

Having established the results on the estimation matrix $\boldsymbol{P}$, we can now provide regret guarantees for Algorithm 2. We recall that the oracle we use is aware of both the observation model $\boldsymbol{O}$ and the transition model $\boldsymbol{P}$ but does not observe the hidden state. As shown in the definition of the regret in Equation 6, it builds a belief over the states, using the formulation defined in Equation 3 and selects the arm that maximizes the expected instantaneous reward. The derived regret upper bound is provided in the following:

**Theorem 6.1.** *Suppose Assumptions 4.1 and 4.2 hold and suppose that the Markov chain with transition matrix $\boldsymbol{P}$ has stationary stationary distribution $\boldsymbol{\pi} \in \Delta(\mathbb{S})$. By assuming that the chain starts from an arbitrary distribution $\boldsymbol{\nu} \in \Delta(\mathbb{S})$ and by considering a finite horizon $T$, there exists a constant $T_0$, with $T > T_0$, such that with probability at least $1 - \delta$, the regret of the SL-EC Algorithm satisfies:*

$$\mathfrak{R}(T) \leq 2 \left( \frac{2LI^2}{\sigma_{\min}^2(\boldsymbol{O})\pi_{\min}} \sqrt{\frac{S(C + \log(CI^2/\delta))}{(1 - \lambda^{2I^2})}} \cdot T \right)^{2/3}, \tag{16}$$

*where $L = \frac{4S(1-\epsilon)^2}{\epsilon^3} + \sqrt{S}$ is a constant that is used to bound the error in the estimated belief (more details in Proposition E.6 in Appendix E). The presented regret has an order of $\mathcal{O}(T^{2/3})$ w.r.t. the horizon $T$, as common when using an Explore then Commit algorithm. A detailed proof of this theorem can be found in Appendix B. The presented bound on the regret can be achieved by appropriately choosing the exploration horizon $T_0$. More specifically, we set it as follows:*

$$T_0 = \left( \frac{2LTI^2}{\sigma_{\min}^2(\boldsymbol{O})\pi_{\min}} \sqrt{\frac{S(C + \log(CI^2/\delta))}{(1 - \lambda^{2I^2})}} \right)^{2/3}. \tag{17}$$

To compute $T_0$, we need to have information about the minimum value of the stationary distribution $\pi_{\min}$ and about the modulus of the second highest eigenvalue $\lambda$. If they are not available, a slightly different

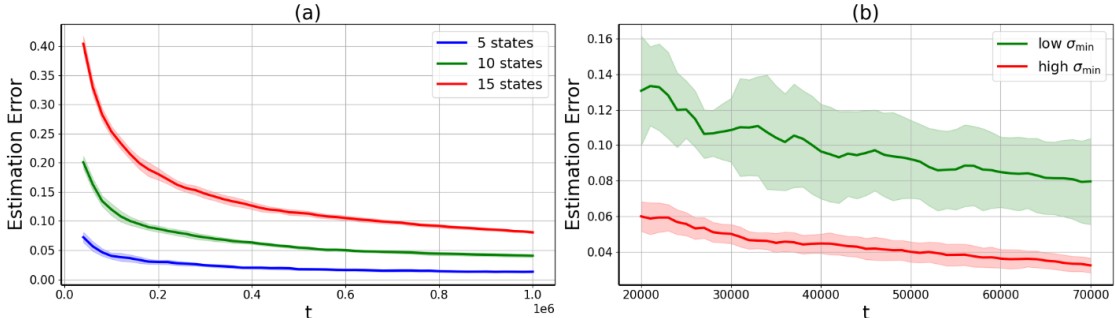

Figure 1: (a) Difference between the estimated and real transition matrix with an increasing number of samples. The metric used is $\|\cdot\|_F$ (10 runs, 95% c.i.), (b) Difference between real and estimated transition matrix using two different subsets of arms of size $J = 3$ arms from the 8 available on a problem with 5 states. The metric used is $\|\cdot\|_F$ (10 runs, 95% c.i.).

version of the bound can be derived so that $T_0$ can be optimized by only requiring the knowledge of $\epsilon$ from Assumption 4.1. More details are reported in Section B.3 of Appendix B.

**Choice of Explore then Commit**  The choice of an Explore then Commit type of algorithm is mainly due to the simplicity of this approach. We believe that this scaling of the regret with respect to time cannot be further improved within the considered class of problems and this is mainly due to the identifiability condition reported in Assumption 4.2. Indeed this class of problems includes worst-case instances where we are guaranteed to acquire information only by pulling all the available arms, thus a forced exploration phase is always required. A similar approach has been used in Zhou et al. (2021): they devise the SEEU algorithm which alternates between full exploration and full exploitation phases reaching a $\widetilde{\mathcal{O}}(T)^{2/3}$ regret guarantee. Instead of the phased algorithm, we opted for an EC approach.

## 7  Numerical Simulations

In this section, we provide numerical simulations on synthetic and semi-synthetic data based on the Movie-Lens 1M (Harper & Konstan, 2015) dataset, demonstrating the effectiveness of the proposed Markov chain estimation procedure. Specifically, we show the efficiency of the offline arm selection procedure described in Section 5.1.1 and conduct a comparison between our SL-EC Algorithm and several baselines in non-stationary settings. In Section 7.3, we provide additional experiments that highlight the performance difference between our approach and a modified technique based on Spectral Decomposition. Finally, in Appendix A we provide different sets of experiments showing the regret comparison under different exploration horizons (Appendix A.2) and we provide some numerical simulations showing the estimation error incurred when the provided observation model is misspecified (Appendix A.3).

### 7.1  Estimation Error of Transition Matrix

The first set of experiments is devoted to showing the error incurred by the estimation procedure of the transition matrix in relation to the number of samples considered and the set of actions used for estimation. The left side of Figure 1 illustrates the estimation error of the transition matrix given different instances of Switching Bandits with an increasing number of states. In particular, we fix the number of total actions $I = 10$ and number of observations $V = 10$ and consider three instances with $S = 5$, $S = 10$ and $S = 15$ number of states. As expected, we can see that as the number of states increases, the problem becomes more complex, and more samples are needed to improve the estimation. Figure 1 reports the Frobenius norm $\|\cdot\|_F$ of the error between the true and the estimated transition matrix. We can notice that the estimation procedure is particularly efficient leading to low error values even with a limited number of samples, as can be observed from the steep error drop appearing in the first part of the plot.

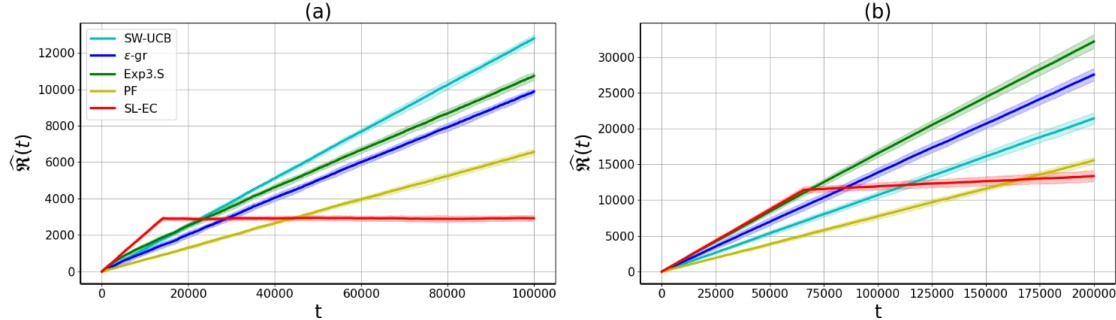

Figure 2: Plots of regret comparing the SL-EC Algorithm with some non-stationary bandit algorithms using: (a) synthetic data with parameters $S = 3$ states, $I = 4$ actions and $V = 5$ observations (5 runs, 95% c.i.); (b) data from MovieLens assuming $S = 5$ states, $I = 18$ actions and $V = 5$ observations (5 runs, 95% c.i.).

The right plot in Figure 1, instead, shows the estimation error obtained by using a different subset of arms. As mentioned in previous sections, it is not always beneficial to use all the available actions during the estimation procedure, but selecting a subset of actions may be preferable. Furthermore, we show that by selecting specific subsets of arms, we can improve the estimation w.r.t using other subsets. For this experiment, we consider $J = 3$ arms among the $I = 8$ available for a Switching MAB instance with $S = 5$ states. We then identify the optimal subset of arms of size $J$ and initiate the estimation process using the selected subset. In order to find the best one, we generate all matrices of type $G$, as described in Section 5.1.1 and choose the matrix with the highest $\sigma_{\min}(G)$. The subset of arms generating that matrix will be used for estimation. The estimation error of the best subset of arms is represented in the plot with the red line, while we represent in green the estimation error of the subset having the lowest $\sigma_{\min}(G)$. The figure clearly exhibits the performance difference between the two choices, thereby validating our claims. Additional details about the characteristics of the matrices used in the experiments are provided in Appendix A.1.

## 7.2 Algorithms Comparisons

In this second set of experiments, we compare the regret suffered by our SL-EC approach with other algorithms specifically designed for non-stationary environments. Following the recent work of Zhou et al. (2021), we consider the subsequent baseline algorithms: the simple $\epsilon$-*greedy* heuristics, a sliding-window algorithm such as *SW-UCB* (Garivier & Moulines, 2011) that is generally able to deal with non-stationary settings and the *Exp3.S* (Auer et al., 2002) algorithm. The parameters for all the baseline algorithms have been properly tuned according to the settings considered. For the specific experiments considered, we adopted scaled values for the exploration horizon $T_0$ w.r.t. the result derived from the theory.[7]
It is worth noting that, unlike our SL-EC algorithm, the baselines do not assume knowledge of the observation model or the underlying Markov chain. In contrast, our approach utilizes the observation model to estimate the transition matrix and to update the belief over the current state. Additionally, we compare our approach with a particle filter algorithm proposed in Hong et al. (2020b) about non-stationary Latent Bandits. They consider two settings: one with complete knowledge of both the observation and transition models and another that incorporates priors on the parameters of the models to account for uncertainty. We compare against a mixture of these two settings by providing their algorithm with full information about the observation model (as it is for our case) and an informative prior about the true transition model. The comparison is made in terms of the empirical cumulative regret $\widehat{\mathfrak{R}}(t)$ averaged over multiple independent runs.

---

[7] We used the value suggested by Equation 17 divided by $(10L)^{2/3}$. Scaling the values obtained by theory is common in the scientific literature and mostly translates into bigger multiplicative constants in the final regret bound or in similar bounds but holding with smaller probability. For our case, under the reduced exploration value, the regret presented in Theorem 6.1 increases by a multiplicative factor of $(10L)^{1/3}$.

### 7.2.1 Synthetic Experiments

These experiments have been conducted on various problem configurations with different numbers of states $S$, actions $I$, and observations $V$. The regret results for one configuration are shown in Figure 2(a). From the figure, it is clear that most of the baseline algorithms display a linear time dependence for the regret. This is expected since these algorithms do not take into account the underlying Markov chain that governs the process. The particle filter algorithm, despite being given a good initial prior on the transition model, does not achieve the same performance of SL-EC in the long run. Conversely, we can notice a quite different behavior for our algorithm that, in line with an Explore then Commit approach, initially accumulates a large regret and then experiences a drastic slope change when the exploitation phase begins. The regret shown in each plot is the average over all the runs. For further information regarding the generation of the transition and observation models, as well as the hyperparameters used for the baseline algorithms, we refer the reader to Appendix A.1.

As a remark, our algorithm outperforms the others when the absolute spectral gap $1 - \lambda$ of the chain has values closer to 1. Indeed, if this is not the case, simple exploration heuristics such as $\epsilon$-greedy would lead to comparable performance. A clear example is when the transition matrix $\boldsymbol{P}$ defining the chain assigns equal probability to all transitions. In this scenario, all states can be considered independent and identically distributed, and we get no advantage from the knowledge of the matrix $\boldsymbol{P}$ over the use of an algorithm such as $\epsilon$-greedy.

### 7.2.2 MovieLens Experiments

We also perform some experiments on semi-synthetic data based on MovieLens 1M (Harper & Konstan, 2015), a well-known collaborative filtering dataset where users rate different movies each belonging to a specific set of genres. We adopt a procedure similar to the one used in Hong et al. (2020b). The dataset is initially filtered to include only users who rated at least 100 movies and the movies that have been rated by at least 100 users. After that, we combine the available information in order to obtain a table where each row contains the mean of the ratings for each observed genre for each user (user-genre-rating table). If the user didn't observe any movie belonging to a specific genre, the cell is empty. From the obtained matrix, we select 70% of all ratings as a training dataset and use the remaining 30% as a test set. The sparse matrices so obtained are completed using least-squares matrix completion (Mnih & Salakhutdinov, 2007) using rank 10 and leading to a low prediction error.

Having defined the appropriate rank, we use the predictions on the empty cells of the original *user-genre rating* matrix to fill the entire table. We define a switching bandit instance by using the notion of a *superuser* inspired by Hong et al. (2020b). We use $k$-means to cluster users using the rows of the user-genre-rating matrix. The users belonging to the same cluster define a superuser that embeds a set of users with similar tastes. The information about the users belonging to the same clusters is then combined and used to generate categorical distributions on the rating, given each superuser and each possible genre (our actions). We choose $k = 5$ for the number of superusers as it is the one that yields clusters with more similar dimensions, and we use $I = 18$ for the actions since it represents the number of identified genres. The number of observations $V = 5$ corresponds to the 5 possible ratings that a movie can get. The transition matrix that governs the dynamics with which superusers alternate is defined by giving higher probabilities to transitions to similar states and also giving higher weights to self-loops in order to avoid too frequent changes. The interaction goes as follows. At each step, a new superuser $s_t$ is sampled based on $s_{t-1}$ and the transition model. The agent chooses an action $a_t$ corresponding to a genre to propose and gets a rating that is sampled from the categorical distribution defined by vector $\boldsymbol{O}\big((a_t, \cdot), s_t\big)$.

As for the synthetic case, our algorithm is compared to other baselines. From Figure 2(b), we can see that our SL-EC still outperforms the other baselines in the considered horizon. However, we highlight that our goal is not to beat the baselines since the comparison is not fair as most of them do not take into account the underlying Markov process, but we aim to show the difference w.r.t. other algorithms belonging to state of the art. More details about the experiments on Movielens are reported in Appendix A.

Table 1: Comparison with Nearly Deterministic Models.

| 2 States | 3K samples | 6K samples | 9K samples | 15K samples |
|---|---|---|---|---|
| *SD O* | 0.0493 (0.0097) | 0.0379 (0.0103) | 0.0335 (0.0097) | 0.0259 (0.0081) |
| *SD T* | 0.0342 (0.0185) | **0.0189** (0.0097) | 0.0149 (0.0032) | 0.0101 (0.007) |
| *Alg. 1* | **0.0234** (0.015) | 0.02 (0.0203) | **0.0119** (0.009) | **0.008** (0.0032) |
| 3 States | 150K samples | 300K samples | 600K samples | 900K samples |
| *SD O* | 0.0165 (0.0044) | 0.0113 (0.0036) | 0.0097 (0.0033) | 0.0085 (0.0018) |
| *SD T* | 0.1547 (0.0517) | 0.154 (0.0532) | 0.1544 (0.0534) | 0.1541 (0.0532) |
| *Alg. 1* | **0.0066** (0.0026) | **0.0046** (0.0012) | **0.0037** (0.0018) | **0.0031** (0.0008) |
| 5 States | 150K samples | 300K samples | 600K samples | 900K samples |
| *SD O* | 0.0681 (0.0178) | 0.0513 (0.0111) | 0.0354 (0.0127) | 0.0283 (0.0082) |
| *SD T* | 0.2409 (0.0633) | 0.2484 (0.0584) | 0.243 (0.0603) | 0.2407 (0.0601) |
| *Alg. 1* | **0.0283** (0.0054) | **0.0195** (0.0036) | **0.0137** (0.0033) | **0.0115** (0.0034) |

## 7.3 Numerical Comparisons with a Modified Spectral Decomposition Technique

The focus of this last set of experiments is to show the difference between a modified Spectral Decomposition (SD) technique and our estimation approach detailed in Algorithm 1. Among the various applications, SD techniques are typically used for learning with Hidden Markov Models (HMM) where no information about the observation and transition model is provided. Zhou et al. (2021) make use of these techniques to get an estimation of both the observation and the transition model. It is important to highlight that SD methods are hardly used in practice because of their computational and sample complexity. Indeed, both the related works of Zhou et al. (2021) and Azizzadenesheli et al. (2016) include only proof-of-concept experiments with 2 hidden states and 2 possible actions.

To make the comparison fairer, we consider a modified SD technique that is provided with information about the observation model in order to help the estimation process, as will be briefly explained. The original SD technique to which we refer follows the procedures highlighted in Anandkumar et al. (2014) for HMM and makes use of the Robust Tensor Power (RTP) method for orthogonal tensor decomposition. In typical SD techniques, data is collected by sampling an action at each time step and updating the computed statistics with the observed realization. With the presented modified SD technique, at each step, we do not simply update the statistics with the observation obtained when pulling the arm, but we give information about the observation distribution for all the available arms, with this information being conditioned on the underlying current state. In this way, it is like pulling all the arms at each step and receiving full information about their associated reward distributions, given the underlying state.

We perform various experiments by fixing the number of arms ($I = 20$) and the number of possible rewards ($V = 5$) for each arm and by changing the number of states. Each experiment is performed over 10 different runs, where for each run a transition and observation model are generated. For each experiment, our estimation procedure uses 3 arms among the 20 available, which are selected using the offline arms selection strategy. The transition and observation matrices are created in two different ways: the former focuses on nearly-deterministic matrices (Table 1), while the latter considers more stochasticity for both of them (Table 2).

The results of the experiments are structured in the following way. Each of the two Tables contains mini-tables representing sets of experiments characterized by different number of states. By fixing the number of states for the experiments, each mini-table shows three rows: the first one (indicated with *SD O*) contains the Frobenius norm of the estimation error of the observation matrix with the modified SD technique, the second row (indicated with *SD T*) represents the Frobenius norm of the estimation error of the transition matrix with the modified SD technique, while the third row represents the Frobenius norm of the estimation error of the transition matrix computed with Algorithm 1. For each experiment, we report the mean error over the 10 runs and one standard deviation between parenthesis. The modified SD technique clearly enhances the accuracy of estimating the observation model compared to standard SD approaches: this aspect is

Table 2: Comparison with Higher Model Stochasticity.

| 2 States | 150K samples | 210K samples | 270K samples |
|---|---|---|---|
| *SD O* | 0.1500 (0.2639) | 0.1411 (0.2741) | 0.1455 (0.2665) |
| *SD T* | 0.1488 (0.1536) | 0.1699 (0.1742) | 0.1576 (0.1702) |
| **Alg. 1** | **0.0145** (0.0175) | **0.0145** (0.0134) | **0.0125** (0.0103) |
| 3 States | 300K samples | 600K samples | 900K samples |
| *SD O* | 0.2987 (0.2128) | 0.3078 (0.2177) | 0.2594 (0.2309) |
| *SD T* | 0.3916 (0.2804) | 0.4425 (0.2637) | 0.4187 (0.2728) |
| **Alg. 1** | **0.0077** (0.003) | **0.0063** (0.0023) | **0.0052** (0.002) |

evident from the relatively low estimation errors observed in the *SD O* rows. We present this information to illustrate that the comparison between our estimation procedure and SD approaches is now more fair due to the modified SD technique employed. Having clarified this aspect, we focus on the estimation error of the transition model between the two different methods: this information is indeed separated from *SD O* by a dashed line. We show the experiments with lower estimation errors in bold.

The results of this first set of experiments are reported in Table 1. As already anticipated, both the observation and transition matrices are almost deterministic, hence having high probability on a specific observation/state and low probabilities for all the others. For transition matrices, the highest probability is assigned to the probability of staying in the same state. Near-determinism is defined to simplify the problem by making states more distinguishable. By inspecting the results, it is clear that Algorithm 1 outperforms the modified SD technique in almost all the scenarios. Comparable results are only achieved in the experiment with 2 states.

Table 2 reports instead the experimental results obtained using both transition and observation matrices with less peaked distributions, thus higher stochasticity. The discrepancy between our approach and the modified SD technique is more evident in this scenario. This aspect can be justified by the theoretical comparison reported in Appendix D, where it can be observed that, compared to our estimation approach, SD techniques have a dependency of higher order on the minimum singular values of both the observation and the transition models. Thus, when the observation matrix is more stochastic, its $\sigma_{\min}(\boldsymbol{O})$ typically decreases, and this aspect results in a higher estimation error. Indeed, it can be noticed that the estimation error is significant and the number of samples required to perform this set of experiments is much higher than that used for the nearly-deterministic case. Experiments involving a higher number of states instead were not able to reach convergence with a number of samples of the order $10^5$ and, by trying to increase this quantity, there were memory space problems with the used hardware (Intel i7-11th and 16G RAM).

Again, we would like to emphasize that SD techniques are explicitly meant to work in a different setting, intrinsically more complex, where no information about either the transition or the observation model is provided. However, with this set of experiments, we wanted to show that if, instead, we have knowledge about the observation model, directly using this information in the SD techniques does not lead to performances comparable to our approach.

## 8 Discussion and Conclusions

This paper studies a Latent Bandit problem with latent states changing in time according to an underlying unknown Markov chain. Each state is represented by a different Bandit instance that is unobserved by the agent. As common in the latent Bandit literature, we assumed to know the observation model relating each MAB to the reward distribution of its actions, and by using some mild assumptions, we presented a novel estimation technique using the information derived from consecutive pulls of pairs of arms. As far as we know, we are the first to present an estimation procedure of this type aiming at directly estimating the stationary distribution $\boldsymbol{w}$ of consecutive states. The approach is easy to use and does not require specific hyperparameters to be set. We provided an offline arm selection that selects the best subsets of arms to speed

up the estimation process. We analyzed the dependence of the parameters on the complexity of the problem, and we showed how our estimation approach can be extended to handle models with continuous observation distributions. We used the presented technique in our SL-EC algorithm that uses an Explore then Commit approach and for which we proved a $\mathcal{O}(T^{2/3})$ regret bound. The experimental evaluation confirmed our theoretical findings showing advantages over some baseline algorithms designed for non-stationary MABs and showing good estimation performances even in scenarios with larger problems. Furthermore, we compared our approach both empirically and theoretically (Appendix D) with SD techniques, taking into account the differences between the two procedures.

We identify different future research directions for the presented work, such as designing new algorithms that are able to exploit the flexibility in the exploration policy determined by the defined procedure, allegedly in an optimistic way. It may also be interesting to deepen the understanding of this problem when dealing with continuous reward models, trying to design optimal ways to discretize them in order to reach faster estimation performances. We could also consider the extension to the continuous state space setting (e.g., linearMDPs): among the main challenges in this scenario, we consider the adoption of a different representation for the reference matrix that would otherwise not be computable with infinite states and the redefinition of the stationary distribution over consecutive states. In such a case, it might be beneficial to estimate the feature functions directly by means of which the linear MDP is defined. Finally, it might be worth considering a contextual version of the proposed setting. According to the assumptions made, for example, whether the context is discrete or continuous or whether it is related or not to the latent state, this aspect may bring another dimension to the observation space. Redefining the reference matrix by taking this feature into account will likely lead to more informative components and help with the estimation process.

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
