# A  Additional Simulations and Details

In this section, we will give some details on the experiments reported in the main paper and we will provide the results of a different set of experiments showing the regret comparison of the different algorithms under different exploration horizons (Section A.2). In Section A.3 we also provide numerical simulations showing the estimation error incurred when the provided observation model is different from the real one[8].

## A.1  Experiment Details

### A.1.1  Estimation Error of Transition Matrix

For the experiments related to the estimation error of the transition matrix in Figure 1, we generated a set of transition and observation matrices with the following characteristics:

- for the plot on the left, we fixed the number $I = 10$ of possible actions and $V = 10$ of finite observations. We then consider the estimation procedure for problems of different sizes with $S = 5$, $S = 10$ and $S = 15$ number of states respectively;

- for the plot on the right, the considered estimated problem has $S = 5$ states, $I = 8$ possible actions, $V = 10$ finite observations.

The transition and observation matrices have been generated using the mentioned hyperparameters as follows. An initial version of transition and observation matrices is generated with random elements and, subsequently:

- regarding the transition matrix, we add a tuned diagonal matrix to the initial random version and then normalize. In this way, we give more probability on self transitions;

- regarding the observation matrix, for each pair of states and actions, we choose a specific reward that will be drawn with higher probability, in order to avoid having too much stochasticity in the reward distributions.

The scheme just presented is also used for the generation of matrices in the experiments showing the regret of the different algorithms.
For the experiments in the plot on the right, let's denote with $\boldsymbol{G}_g$ and $\boldsymbol{G}_r$ the reduced action observation matrices containing the subset of arms for the green and the red lines respectively. Their values are $\sigma_{\min}(\boldsymbol{G}_g) \approx 0.14$ and $\sigma_{\min}(\boldsymbol{G}_r) \approx 0.27$.

### A.1.2  Algorithms Comparisons

**Synthetic Experiments**  For this set of experiments, the parameters used for the generation of the transition and observation matrix are $S = 3$ states, $I = 4$ possible actions and $V = 5$ finite observations. The generation of the matrices is not completely random and follows the same procedure explained in the previous paragraph for the experiment on the matrix estimation error. For the specific experiments considered, we adopted for the exploration horizon $T_0$ the result derived from theory divided by $(10L)^{2/3}$. For the plots shown in the main paper, the hyperparameters used are $\epsilon = 0.05$ for the $\epsilon$-greedy approach, a value of $L_w = 1000$ for the sliding-window UCB algorithm, and the suggested value $1/T$ for the $\alpha$ parameter in the Exp3.S algorithm. For the particle filter algorithm, we used 100 different particles and a resampling threshold of 25 for the *Effective Sample Size*.

**Movielens Experiments**  In this section, we give more details about the experiments on the Movielens 1M dataset. As detailed in the main body of the paper the transition matrix is constructed taking into account the similarity between superusers. In order to do that we use the cosine similarity to define the

---

[8]The code for the experiments can be found at `https://github.com/alesnow97/SwitchingLatentBandits`.

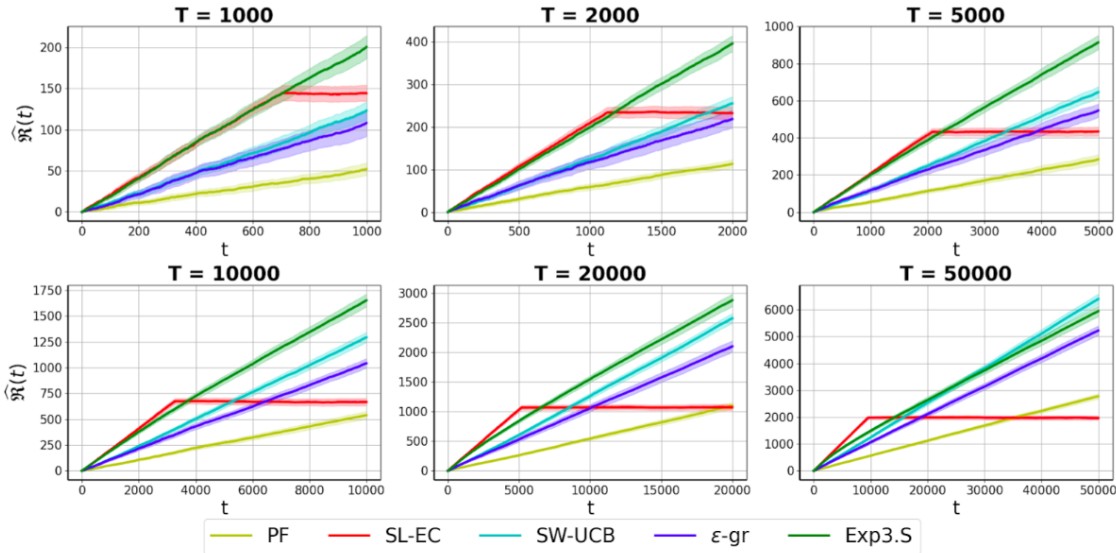

Figure 3: Empirical regret on a Switching Bandits instance with $S = 3$ states, $I = 4$ actions and $V = 5$ observations under different horizons (20 runs, 95%c.i.).

initial transition matrix. After that, we add a diagonal matrix to the previously obtained matrix in order to give higher values to self-loop probabilities. The final transition matrix is obtained by normalizing along each row. The self-loop probabilities are close to the value 0.98 thus representing a more realistic scenario with regimes not switching too often. The rewards are obtained by scaling the ratings obtained from a single movie in the range $[0, 1]$.

For the plots shown in the main paper, the hyperparameters used are $\epsilon = 0.05$ for the $\epsilon$-greedy approach, $L_w = 300$ for the sliding-window UCB algorithm, and the suggested value $1/T$ for the $\alpha$ parameter in the Exp3.S algorithm. For the particle filter algorithm, we used 300 different particles and a resampling threshold of 50 for the *Effective Sample Size*. Our SL-EC algorithm has been run using an offline arm selection procedure for choosing the arms to use during the exploration phase. The number of selected arms has been fixed to $J = 5$.

## A.2 Numerical Simulation on Regret Comparison under Different Horizons

The objective of this new set of experiments is to show the regret results of our approach under different values of the interaction horizon $T$. For practical reasons, we adopt a lower value for the exploration horizon $T_0$, following analogous considerations as those reported in Azizzadenesheli et al. (2016). We recall that using a different exploration horizon than the one suggested by theory would mostly result in having bigger multiplicative constants in the final regret bound or a similar bound but holding with a smaller probability.

We conducted two series of experiments:

1. the first set presents the regret over different horizons using as exploration horizon the one defined by Equation 17 divided by $(10L)^{2/3}$. The results are presented in Figure 3.

2. The second set of experiments instead shows the performance of the SL-EC algorithm when the exploration horizon is manually chosen. The results are presented in Figure 4.

**First Set of Experiments** For this first set of experiments, we employ the same instance used for the synthetic experiment in the main paper (reported in Figure 2). We run 20 experiments for each different value of the horizon. The horizon lengths are reported on each subplot.

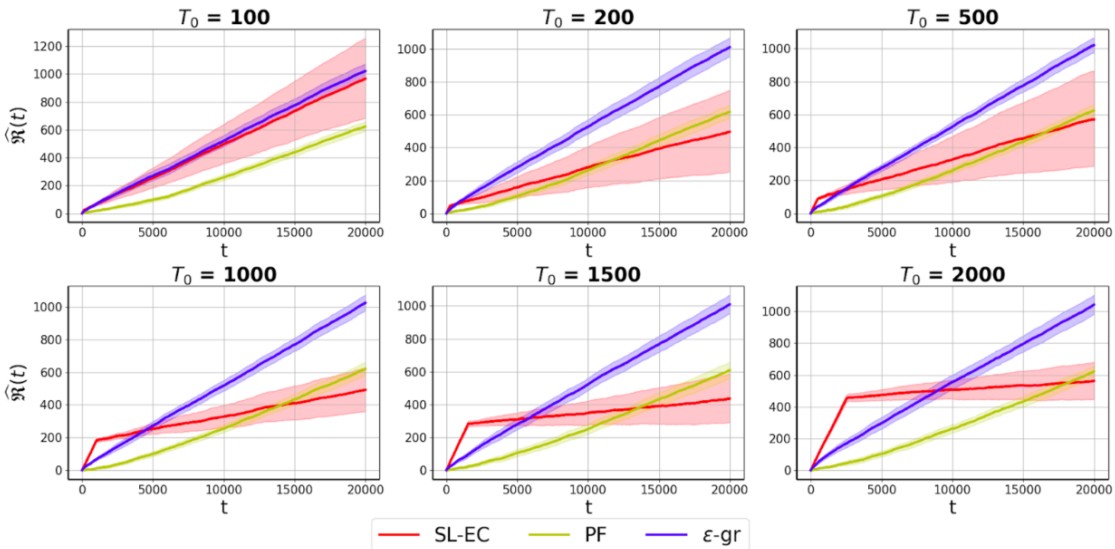

Figure 4: Empirical regret on a Switching Bandits instance with $S = 4$ states, $I = 8$ actions and $V = 5$ observations under manually tuned values for the exploration horizon $T_0$ (20 runs, 95%c.i.).

From Figure 3, we can observe that the SL-EC algorithm has generally better performances with respect to most of the considered baselines even with a reduced horizon. The algorithm based on particle filter has more advantages in shorter horizons since it does not have an initial exploration phase. However, besides the good initial prior we provide for the transition model, it is not able to reach a good enough estimation of the transition model and its advantage with respect to the SL-EC algorithm vanishes in the long run.

**Second Set of Experiments** The second set of experiments shows the performances of the SL-EC algorithm when the value of the exploitation horizon is small with respect to the total horizon. In these experiments, we consider a different Switching Bandit instance with $S = 4$ states, $I = 8$ actions, and $V = 5$ observations. We manually set the length of the exploration phases.
The different exploration values employed are reported as title of each subplot. For this set of experiments, we omit the result of both the EXP3S and the sliding Window UCB algorithm, since they both experience high error values that would make the plot visualization less clear. The experiments are reported using a total horizon of $T = 20000$.

From Figure 4, we can observe that when the number of samples used for exploration is small with respect to the problem space, the regret incurred during the exploitation phase is high since the estimated transition model has high error. Furthermore, the estimates present high variance since they are based on few samples and this leads to high variance as well in the exploitation phase of the SL-EC algorithm.

As the number of samples increases, we can see a reduction in the slope of the regret of the exploitation phase and a reduced variance as well. With these experiments, we would like to highlight that our algorithm can reach good results even when the number of samples used for estimation is particularly low with respect to the one suggested by theory.

## A.3 Numerical Simulation using Algorithm 1 with Misspecified Obsevation Model

In this set of experiments, we show the performance of the estimation algorithm (Algorithm 1) when the observation model used by the algorithm differs from the real one.

The experiments have been conducted on two different instances of Switching Latent Bandits, characterized as follows:

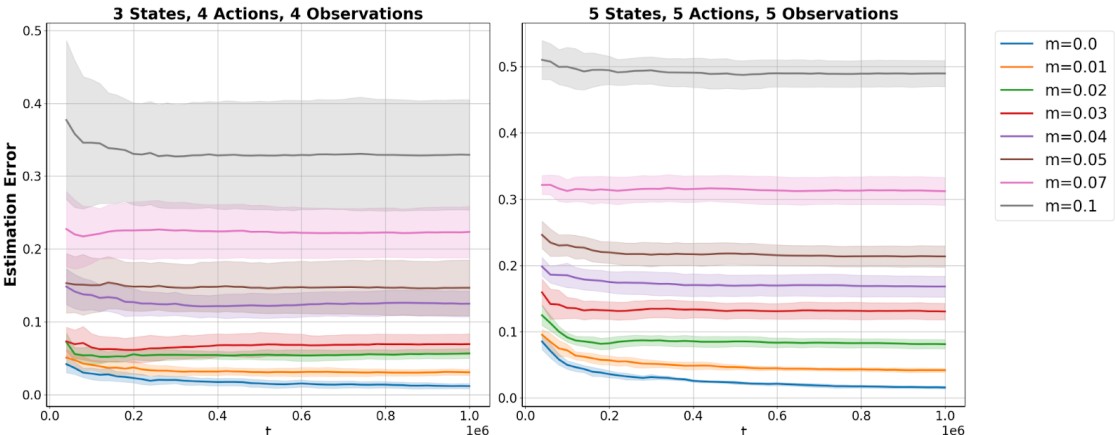

Figure 5: Frobenious norm of the estimation error of the transition model of 2 different Switching Bandits instances under different misspecification levels (10 runs, 95%c.i.).

- the first instance has 3 states, 4 actions, and 4 observations;

- the second instance has 5 states, 5 actions, and 5 observations.

Given the real observation model of each instance, we perturbed it according to different levels of intensity, using a parameter $m > 0$ that we call *misspecification level*. The perturbation is done as follows. For each combination of state $s$, arm $a$, and reward $r$, we sample a random value in the interval $[0, 1]$, we multiply the sampled quantity for the *misspecification level* parameter $m$ and we add it to the quantity $\Pr(r|s, a)$. So, the bigger the *misspecification level* is, the higher is expected to be the difference between the real and misspecified observation models. After this step, we normalize the newly obtained probabilities to make them sum to 1.

In the conducted experiments we did not apply the offline arm selection procedure, every arm is thus used during the estimation phase. Results are reported in terms of the Frobenius norm of the error between the real and the estimated transition matrix. For each one of the two instances, and for each value of $m$, 10 runs are performed. At each run a different perturbed observation model is generated using the procedure described above.

The results of the experiment are presented in Figure 5. By observing it, we can see that the proposed estimation algorithm is robust with respect to the misspecification of the observation model. Reasonably, we see the estimation error growing with model misspecification. In practice, if the reward model is accurate enough, a good estimation of the transition model can be performed.
Additionally, it can be noted that larger values of $m$ lead to higher variance in the experiments since the perturbed observation models differ more among themselves through the different runs.

## B    Theoretical Results

In this Section, we will provide the proofs of Lemma 6.1 (Section B.1) and Theorem 6.1 (Section B.2) presented in the main paper. Finally, Section B.3 shows how to compute the exploration horizon $T_0$ when only the minimum value of the transition matrix $\epsilon$ is known.

### B.1    Proof of Lemma 6.1

We will start by reporting Lemma 6.1 of the main paper and its proof.

**Lemma 6.1.** *Suppose Assumption 4.2 holds and suppose that the Markov chain with transition matrix $\boldsymbol{P}$ is ergodic, such that $\pi_{\min} := \min_{s \in \mathbb{S}} \boldsymbol{\pi}(s) > 0$ with $\boldsymbol{\pi} \in \Delta(\mathbb{S})$ being the stationary distribution of the chain.*

*By assuming that the chain starts from an arbitrary distribution $\boldsymbol{\nu} \in \Delta(\mathbb{S})$, by pulling each pair of arms in a round-robin fashion for $N$ rounds and using the estimation procedure reported in Algorithm 1, we have that with probability at least $1 - \delta$ the estimation error of the transition matrix $\boldsymbol{P}$ will be:*

$$\|\boldsymbol{P} - \widehat{\boldsymbol{P}}\|_F \leq \frac{2I}{\sigma_{\min}^2(\boldsymbol{O})\pi_{\min}}\sqrt{\frac{2S(C + \log(CI^2/\delta))}{(1 - \lambda^{2I^2})N}}, \tag{15}$$

*Proof.* The proof of the presented bound can be decomposed into two main parts. On one side, we can define the bound of the estimation error of the matrix $\boldsymbol{W}$ and, secondly, the error of the transition matrix $\boldsymbol{P}$ that derives from the estimation error of $\boldsymbol{W}$. We will first tackle this last part.

As a relevant note, in this proof we are assuming that all the values coming from the estimated matrix $\widehat{\boldsymbol{W}}$ are positive. However, being $\widehat{\boldsymbol{W}}$ the result of Equation Equation 14, it can also happen that, when not enough data are collected, the estimation is less accurate and some negative terms may appear. If this happen, we can simply set to 0 the negative terms while preserving all the theoretical steps presented here. Having clarified this aspect, we are ready to report the following steps:

$$\|\boldsymbol{P} - \widehat{\boldsymbol{P}}\|_F = \sqrt{\sum_{(s,s') \in \mathbb{S}^2}(\boldsymbol{P}(s,s') - \widehat{\boldsymbol{P}}(s,s'))^2} = \sqrt{\sum_{s \in \mathbb{S}}\|\boldsymbol{P}(s,\cdot) - \widehat{\boldsymbol{P}}(s,\cdot)\|_2^2}$$

$$= \sqrt{\sum_{s \in \mathbb{S}}\left\|\frac{\boldsymbol{W}(s,\cdot)}{\|\boldsymbol{W}(s,\cdot)\|_1} - \frac{\widehat{\boldsymbol{W}}(s,\cdot)}{\|\widehat{\boldsymbol{W}}(s,\cdot)\|_1}\right\|_2^2}$$

$$\leq \sqrt{\sum_{s \in \mathbb{S}}\left\|\frac{\boldsymbol{W}(s,\cdot)}{\|\boldsymbol{W}(s,\cdot)\|_2} - \frac{\widehat{\boldsymbol{W}}(s,\cdot)}{\|\widehat{\boldsymbol{W}}(s,\cdot)\|_2}\right\|_2^2} \tag{P.4}$$

$$\leq \sqrt{\sum_{s \in \mathbb{S}}\frac{4\|\boldsymbol{W}(s,\cdot) - \widehat{\boldsymbol{W}}(s,\cdot)\|_2^2}{\max\{\|\boldsymbol{W}(s,\cdot)\|_2, \|\widehat{\boldsymbol{W}}(s,\cdot)\|_2\}^2}} \tag{P.5}$$

$$\leq \sqrt{\sum_{s \in \mathbb{S}}\frac{4\|\boldsymbol{W}(s,\cdot) - \widehat{\boldsymbol{W}}(s,\cdot)\|_2^2}{\|\boldsymbol{W}(s,\cdot)\|_2^2}} \tag{P.6}$$

$$\leq \sqrt{\sum_{s \in \mathbb{S}}\frac{4S\|\boldsymbol{W}(s,\cdot) - \widehat{\boldsymbol{W}}(s,\cdot)\|_2^2}{\pi_{\min}^2}} \tag{P.7}$$

$$= \sqrt{\frac{4S\|\boldsymbol{W} - \widehat{\boldsymbol{W}}\|_F^2}{\pi_{\min}^2}} = \frac{2\sqrt{S}\|\boldsymbol{W} - \widehat{\boldsymbol{W}}\|_F}{\pi_{\min}}. \tag{P.8}$$

We recall that $\boldsymbol{P}(s,\cdot) \in \Delta(\mathbb{S})$ denotes the distribution over the next state when starting from state $s \in \mathbb{S}$ and we denote with $\boldsymbol{W}(s,\cdot)$ the vector of dimension $S$ representing the row of the matrix $\boldsymbol{W}$ of the stationary distribution of consecutive states. Line P.4 derives from the fact that $\|\boldsymbol{W}(s,\cdot)\|_1 \geq \|\boldsymbol{W}(s,\cdot)\|_2$, while line P.5 is obtained by using Lemma E.1. Line P.6 easily follows from

$$\max\{\|\boldsymbol{W}(s,\cdot)\|_2, \|\widehat{\boldsymbol{W}}(s,\cdot)\|_2\} \geq \|\boldsymbol{W}(s,\cdot)\|_2.$$

Line P.7 is instead derived from the following considerations. For any vector $\boldsymbol{W}(s,\cdot)$ of size $S$, it holds that:

$$\|\boldsymbol{W}(s,\cdot)\|_2^2 = \sum_{s' \in \mathbb{S}}\boldsymbol{W}(s,s')^2 \geq \frac{1}{S}\left(\sum_{s' \in \mathbb{S}}\boldsymbol{W}(s,s')\right)^2 = \frac{1}{S}\boldsymbol{\pi}(s)^2 \geq \frac{\pi_{\min}^2}{S},$$

where the first inequality in the expression above follows from the fact that $\sqrt{Y}\|\boldsymbol{y}\|_2 \geq \|\boldsymbol{y}\|_1 \ \forall \boldsymbol{y} \in \mathbb{R}^Y$. The second equality is instead derived from the definition of matrix $\boldsymbol{W}$, since the sum of the elements along the row associated with state $s$ corresponds to the probability value $\boldsymbol{\pi}(s)$ of state $s$ from the stationary distribution $\boldsymbol{\pi}$ induced by the chain. In the last inequality, we bound each of these probabilities by $\pi_{\min} = \min_{s \in \mathbb{S}}\boldsymbol{\pi}(s)$.

Finally, the first equality in line P.8 derives from $\sum_{s \in \mathbb{S}} \|\boldsymbol{W}(s, \cdot) - \widehat{\boldsymbol{W}}(s, \cdot)\|_2^2 = \|\boldsymbol{W} - \widehat{\boldsymbol{W}}\|_F^2$, which holds by the definition of the used quantities.

We will now derive the first part of the proof by defining a high probability bound on the estimation error of the stationary distribution over consecutive states represented by matrix $\boldsymbol{W}$. In order to do that, we use the relation $\|\boldsymbol{W} - \widehat{\boldsymbol{W}}\|_F = \|\boldsymbol{w} - \widehat{\boldsymbol{w}}\|_2$, where we recall that $\boldsymbol{w}$ is obtained by the vectorization of matrix $\boldsymbol{W}$. The bound is obtained assuming that each pair of arms is pulled in a round-robin fashion, as required by the algorithm. The derivation is as follows:

$$
\begin{aligned}
\|\boldsymbol{w} - \widehat{\boldsymbol{w}}\|_2 &= \left\| \boldsymbol{A}^\dagger (\boldsymbol{d} - \widehat{\boldsymbol{d}}) \right\|_2 \\
&\leq \|\boldsymbol{A}^\dagger\|_2 \|\boldsymbol{d} - \widehat{\boldsymbol{d}}\|_2 \\
&= \frac{1}{\sigma_{\min}(\boldsymbol{A})} \|\boldsymbol{d} - \widehat{\boldsymbol{d}}\|_2 = \frac{1}{\sigma_{\min}^2(\boldsymbol{O})} \|\boldsymbol{d} - \widehat{\boldsymbol{d}}\|_2,
\end{aligned} \tag{P.9}
$$

where vector $\widehat{\boldsymbol{d}}$ is obtained from the vector count $\boldsymbol{n}$ by assuming that each pair of arms has been pulled $N$ times (refer to Equation 13). The first inequality is obtained by the consistency property of matrices: the first norm represents the spectral norm of matrix $\boldsymbol{A}^\dagger$, while the second is a $\|\cdot\|_2$ of a vector. The last equality is instead obtained from the properties of the Kronecker product for which it holds that:

$$
\sigma_{\min}(\boldsymbol{A}) = \sigma_{\min}(\boldsymbol{O})\sigma_{\min}(\boldsymbol{O}) = \sigma_{\min}^2(\boldsymbol{O}). \tag{P.10}
$$

Let's consider now the estimation error of the different vectors $\boldsymbol{d}_{a,a'}$ constituting vector $\boldsymbol{d}$, as defined in Equation 10. We have seen in the main paper that each vector $\boldsymbol{d}_{a,a'}$ denotes the distribution over consecutive rewards conditioned on pulling the consecutive arms $(a, a')$ (see Equation 9).

Since the pairs of rewards are not i.i.d but depend on the underlying Markov process, we can use a suited concentration result for discrete distributions that appears in Proposition E.4 under the assumption that the chain does not start from its stationary distribution.

In particular, from Proposition E.4, it holds that $\forall (a, a') \in \mathbb{I}^2$, with probability at least $1 - \delta/I^2$:

$$
\|\boldsymbol{d}_{a,a'} - \widehat{\boldsymbol{d}}_{a,a'}\|_2 \leq \sqrt{\left( \frac{1 + \lambda}{1 - \lambda} \right) \frac{C + \log(CI^2/\delta)}{N}},
$$

with $N$ being the number of samples used for the estimation and $C$ is a constant intuitively representing the distance of the initial distribution from the stationary one. Considering the round-robin procedure used in the exploration phase, the result just presented can be slightly improved. Indeed, being $\boldsymbol{P}$ the transition matrix governing the chain and $\lambda$ the modulus of its second largest eigenvalue, since samples used to estimate each $\widehat{\boldsymbol{d}}_{a,a'}$ are collected every $2I^2$ time instants, this leads to an effective transition matrix $\boldsymbol{P}^{2I^2}$, with associated $\lambda^{2I^2}$ modulus of the second largest eigenvalue. This collection policy has thus the consequence of inducing less temporal dependence among samples. We can modify the previous result by having $\forall (a, a') \in \mathbb{I}^2$, with probability at least $1 - \delta/I^2$:

$$
\|\boldsymbol{d}_{a,a'} - \widehat{\boldsymbol{d}}_{a,a'}\|_2 \leq \sqrt{\left( \frac{1 + \lambda^{2I^2}}{1 - \lambda^{2I^2}} \right) \frac{C + \log(CI^2/\delta)}{N}}. \tag{P.11}
$$

We can now express the following relation that easily follows from the definition of the quantities involved:

$$
\|\boldsymbol{d} - \widehat{\boldsymbol{d}}\|_2 = \sqrt{\sum_{(a,a') \in \mathbb{I}^2} \|\boldsymbol{d}_{a,a'} - \widehat{\boldsymbol{d}}_{a,a'}\|_2^2}. \tag{P.12}
$$

By combining the results in P.9, P.11, P.12, and using a union bound, we have that with probability at least $1 - \delta$:

$$
\begin{aligned}
\|\boldsymbol{w} - \widehat{\boldsymbol{w}}\|_2 &\leq \frac{1}{\sigma_{\min}^2(\boldsymbol{O})} \sqrt{\left( \frac{1 + \lambda^{2I^2}}{1 - \lambda^{2I^2}} \right) \frac{I^2(C + \log(CI^2/\delta))}{N}} \\
&\leq \frac{I}{\sigma_{\min}^2(\boldsymbol{O})} \sqrt{\frac{2(C + \log(CI^2/\delta))}{(1 - \lambda^{2I^2})N}}.
\end{aligned} \tag{P.13}
$$

We now have all the elements needed to show the final result. Putting together P.8 and P.13, we have that with probability at least $1 - \delta$, it holds that:

$$
\|\boldsymbol{P} - \widehat{\boldsymbol{P}}\|_F \leq \frac{2I}{\sigma_{\min}^2(\boldsymbol{O})\pi_{\min}} \sqrt{\frac{2S(C + \log(CI^2/\delta))}{(1 - \lambda^{2I^2})N}},
$$

which concludes the proof. $\qquad \square$

## B.2 Proof of Theorem 6.1

We are now ready to derive the main result related to the regret of the SL-EC Algorithm. We will report here Theorem 6.1 of the main paper.

**Theorem 6.1.** *Suppose Assumptions 4.1 and 4.2 hold and suppose that the Markov chain with transition matrix $\boldsymbol{P}$ has stationary stationary distribution $\boldsymbol{\pi} \in \Delta(\mathbb{S})$. By assuming that the chain starts from an arbitrary distribution $\boldsymbol{\nu} \in \Delta(\mathbb{S})$ and by considering a finite horizon $T$, there exists a constant $T_0$, with $T > T_0$, such that with probability at least $1 - \delta$, the regret of the SL-EC Algorithm satisfies:*

$$\mathfrak{R}(T) \leq 2 \left( \frac{2LI^2}{\sigma_{\min}^2(\boldsymbol{O})\pi_{\min}} \sqrt{\frac{S(C + \log(CI^2/\delta))}{(1 - \lambda^{2I^2})}} \cdot T \right)^{2/3}, \tag{16}$$

*Proof.* The proof of the regret of the SL-EC Algorithm makes use of some of the results previously derived and it can be divided into the regret from the exploration and regret from the exploitation phase.
Considering an exploration phase of length $T_0$, the regret initially suffered can be trivially bounded as:

$$\mathfrak{R}_{1:T_0} = \sum_{t=1}^{T_0} \max_{a \in \mathbb{I}} \langle \boldsymbol{\mu}(a), \mathbf{b}_t \rangle - \mathrm{r}_t \leq \sum_{i=1}^{T_0} 1 = T_0, \tag{P.14}$$

where vector $\boldsymbol{\mu}(a)$ has dimension $S$ and its elements are defined in Equation 4.
For the exploitation phase, we compute a belief vector $\widehat{\mathbf{b}}_t$ at each step according to Equation 5 using the estimate $\widehat{\boldsymbol{P}}$ of the transition matrix. The belief vector is initialized uniformly over the states and updated starting from the initial samples up to those collected at the end of the exploration phase. The analysis of the regret in this part is as follows:

$$\begin{aligned}
\mathfrak{R}_{T_0:T} &= \sum_{t=T_0+1}^{T} \max_{a \in \mathbb{I}} \langle \boldsymbol{\mu}(a), \mathbf{b}_t \rangle - \max_{a \in \mathbb{I}} \langle \boldsymbol{\mu}(a), \widehat{\mathbf{b}}_t \rangle \\
&\leq \sum_{t=T_0+1}^{T} \max_{a \in \mathbb{I}} |\langle \boldsymbol{\mu}(a), \mathbf{b}_t - \widehat{\mathbf{b}}_t \rangle| \\
&\leq \sum_{t=T_0+1}^{T} \|\boldsymbol{\mu}(a)\|_\infty \|\mathbf{b}_t - \widehat{\mathbf{b}}_t\|_1 \\
&\leq \sum_{t=T_0+1}^{T} \|\mathbf{b}_t - \widehat{\mathbf{b}}_t\|_1 \\
&\leq \sum_{t=T_0+1}^{T} L\|\boldsymbol{P} - \widehat{\boldsymbol{P}}_{T_0}\|_F \\
&\leq \frac{4LTI^2}{\sigma_{\min}^2(\boldsymbol{O})\pi_{\min}} \sqrt{\frac{S(C + \log(CI^2/\delta))}{(1 - \lambda^{2I^2})T_0}}, \tag{P.15}
\end{aligned}$$

where in the second inequality we applied Hölder's inequality with norms $\infty$ and 1, while the third inequality is obtained from $\|\boldsymbol{\mu}(a)\|_\infty \leq 1 \, \forall a \in \mathbb{I}$. The fourth inequality is obtained by applying Proposition E.6, while the last inequality uses the concentration derived in Lemma 6.1, considering that the number of rounds is $N = T_0/(2I^2)$ if the exploration horizon has length $T_0$.

Combining together the regrets of the two phases derived in Equation P.14 and in Equation P.15 we have:

$$\mathfrak{R}(T) \leq T_0 + \frac{4LTI^2}{\sigma_{\min}^2(\boldsymbol{O})\pi_{\min}} \sqrt{\frac{S(C + \log(CI^2/\delta))}{(1 - \lambda^{2I^2})T_0}}. \tag{P.16}$$

We can now optimize this bound w.r.t. the exploration length $T_0$ by vanishing the derivative of the right-hand side of Equation P.16. What we get is the following term:

$$T_0 = \left( \frac{2LTI^2}{\sigma_{\min}^2(\boldsymbol{O})\pi_{\min}} \sqrt{\frac{S(C + \log(CI^2/\delta))}{1 - \lambda^{2I^2}}} \right)^{2/3}.$$

By substituting this value of $T_0$ into Equation P.16, we get the result of the Theorem. □

## B.3 Optimization of $T_0$

In order to be able to compute $T_0$, we need to have information about the minimum value of the stationary distribution $\pi_{\min}$ and about the modulus of the second highest eigenvalue $\lambda$. If they are not available, a slightly different version of the bound can be derived in order to compute $T_0$ by only using the knowledge of $\epsilon$ from Assumption 4.1.

Since Theorem 6.1 requires Assumption 4.1 to hold, which is stronger than the ergodicity of the chain, we can further characterize the result obtained in Lemma 6.1. In particular, we can lower bound the minimum probability of the stationary distribution $\pi_{\min}$ with the minimum quantity $\epsilon$ appearing in the transition model $\boldsymbol{P}$, such that:

$$\pi_{\min} \geq \epsilon. \tag{18}$$

A second step involves instead characterizing the value of $\lambda$ as a function of $\epsilon$. To do that, we resort to a quantity $\rho(\cdot)$ known as Dobrushin coefficient (Krishnamurthy, 2016) that defines the rate of convergence of ergodic chains towards their stationary distribution. Given a stochastic matrix $\boldsymbol{T}$ representing the dynamics of a Markov chain defined on a finite state space $\mathbb{X}$ and given two probability vectors in $\nu, \bar{\nu} \in \Delta(\mathbb{X})$, the Dobrushin coefficient satisfies the following relation:

$$\|\boldsymbol{T}^\top \nu - \boldsymbol{T}^\top \bar{\nu}\|_1 \leq \rho(\boldsymbol{T}) \|\nu - \bar{\nu}\|_1,$$

where $\boldsymbol{T}^\top$ represents the transpose of matrix $\boldsymbol{T}$. The inequality above says that the one-step evolution of two probability vectors induced by the same transition matrix $\boldsymbol{T}$ can be bounded by the quantity on the right where scalar $\rho(\boldsymbol{T}) \in [0, 1]$ represents the Dobrushin coefficient. For ergodic chains, this coefficient is always strictly smaller than 1. Hence, by iteratively applying the inequality, it is possible to ensure geometric convergence of the initial distance.

Among the properties of the Dobrushin coefficient (Krishnamurthy, 2016), we have that it is an upper bound to the modulus of the second largest eigenvalue $\lambda$, thus:

$$\lambda(\boldsymbol{T}) \leq \rho(\boldsymbol{T}), \tag{19}$$

where here we use $\lambda(\boldsymbol{T})$ to denote the modulus of the second largest eigenvalue of matrix $\boldsymbol{T}$. Furthermore, the Dobrushin coefficient corresponds to the following quantity:

$$\rho(\boldsymbol{T}) = 1 - \min_{i,j \in \mathbb{X}} \sum_{l \in \mathbb{X}} \min\{\boldsymbol{T}(i, l), \boldsymbol{T}(j, l)\}. \tag{20}$$

For a Markov chain with transition matrix $\boldsymbol{P}$ satisfying Assumption 4.1, we can set an upper bound to the Dobrushin coefficient as follows:

$$\rho(\boldsymbol{P}) = 1 - \min_{i,j \in \mathbb{S}} \sum_{l \in \mathbb{S}} \min\{\boldsymbol{P}(i, l), \boldsymbol{P}(j, l)\} \tag{21}$$

$$\leq 1 - \sum_{l \in \mathbb{S}} \epsilon = 1 - S\epsilon \tag{22}$$

where the inequality derives indeed from Assumption 4.1.
Using now 19, we have that:

$$\lambda(\boldsymbol{P}) \leq \rho(\boldsymbol{P}) \leq 1 - S\epsilon. \tag{23}$$

The bound on the regret of Theorem 6.1 can now be rewritten as:

$$\mathfrak{R}(T) \leq T_0 + \frac{4LTI^2}{\sigma_{\min}^2(\boldsymbol{O})\epsilon} \sqrt{\frac{S(C + \log(CI^2/\delta))}{(1 - (1 - S\epsilon)^{2I^2})T_0}}, \tag{24}$$

where we have used both the results in 18 and in 23. With this expression of the bound, the value of $T_0$ minimizing the formulation is:

$$T_0 = \left( \frac{2LTI^2}{\sigma_{\min}^2(\boldsymbol{O})\epsilon} \sqrt{\frac{S(C + \log(CI^2/\delta))}{1 - (1 - S\epsilon)^{2I^2}}} \right)^{2/3}.$$

which can now be obtained without requiring the knowledge of ether $\pi_{\min}$ or $\lambda$.

## C    Comparison with Related Works

Table 3 defines a comparison between our work and the most related works by considering different aspects for each of them.
Some details about the table are reported below. It contains three different sub-table: the first concerns the assumptions used, the second sub-table gives information about the employed estimation technique, and the latter sub-table gives information about the employed regret minimization algorithm.

We highlight some important aspects that can be observed from this comparison. First of all, we notice that, together with SD techniques, our approach can provide consistent estimation of the model parameters, while this is not achieved with techniques relying on Bayesian updates.

It is also relevant to highlight that we compare against the oracle acting according to Equation 5, which we call here *Optimal Belief-Based Action*. We opted for this choice since Equation 5 can be easily computed, differently from the more powerful optimal POMDP policy whose computation is notoriously intracatable Madani (1999)[9].

Table 3 reported below uses the following abbreviations: SD (Spectral Decomposition) and TS (Thompson Sampling). The *Best Action* oracle used in Hong et al. (2020b) refers to the oracle that always pulls the best action as if the underlying state of the environment is observable.
Furthermore, the meanings of the cells containing the following symbols are:

(*) : The agent knows a prior of both the observation and the transition model;

(**) : The assumption adopted in Jafarnia Jahromi et al. (2022) can be related to the full-rank assumption. It states that, for any two different transition models, a different distribution over action-observation sequences is induced;

(***) : Bayesian regret.

### C.1    Differences with Azizzadenesheli et al. (2016)

We highlight here some differences with respect to the work of Azizzadenesheli et al. (2016). Mainly:

1. They consider the more classical POMDP setting without knowledge of the observation or transition model and use SD approaches to estimate both models. Differently, in our work, we assume a Switching Latent Bandit setting and devise a new estimation technique tailored for learning the transition model while exploiting the knowledge of the observation model.

2. As can be seen in the Table, the regret in Azizzadenesheli et al. (2016) is defined with respect to the optimal memoryless stochastic policy, which is a weaker oracle than the optimal belief-based policy we use. The set of memoryless policies is less powerful than the set of belief-based policies since the former ones are Markovian with respect to the last observation received. Furthermore, the oracle

---

[9]If we instead chose to compare against the optimal POMDP policy and we assumed an optimization oracle that is able to compute the optimal policy given our estimate of the model given by Algorithm 1, by following analogous steps to those used in Zhou et al. (2021) for proof of the regret, it is possible to show that under the same assumptions we use here, we would suffer $\widetilde{\mathcal{O}}(T^{2/3})$ regret.

| | Azizza. et al. (2016) | Zhou et al. (2021) | Jafarnia et al. (2022) | Hong et al. (2020b) | SL-EC |
|---|---|---|---|---|---|
| Knowledge of Obs. Model | No | No | Yes | Yes* | Yes |
| Knowledge of Trans. Model | No | No | No | Yes* | No |
| Ergodicity of Induced Chain | Yes | Yes | Yes | - | Yes |
| Minimum Trans. Prob. | No | Yes | No | - | Yes |
| Invertible Trans. Model | Yes | Yes | No | - | No |
| Full-rank Obs. Model | Yes | Yes | Yes** | - | Yes |
| Minimum Action Prob. | Yes | No | No | No | No |
| Consistent Estimation of Trans. Model | No | No | Yes | - | No |
| Consistent Estimation of Belief | No | No | Yes | - | No |
| **Estimation Technique** | SD | SD | Bayesian Update | Bayesian Update | Algorithm 1 |
| **Consistent Estimation** | ✓ | ✓ | ✗ | ✗ | ✓ |
| **Oracle** | Optimal Stochastic Memoryless Policy | Optimal POMDP policy | Optimal POMDP policy | Best Action | Optimal Belief-Based Action |
| **Algorithm Type** | Optimistic | Alternating Explor. - Exploit. phases | TS-based | TS-based | Optimistic |
| **Regret** | $\widetilde{\mathcal{O}}(\sqrt{T})$ | $\widetilde{\mathcal{O}}(T^{2/3})$ | $\mathcal{O}(T^{2/3})$*** | $\mathcal{O}(T^{2/3})$*** | $\mathcal{O}(T^{2/3})$ |

Table 3: Tables comparing our work with most relevant related works. Note that the first subtable refers to the employed assumptions.

is stochastic since the set of considered policies includes policies having a minimum probability of taking each action. This aspect favors model estimation and is fundamental to applying Spectral techniques and obtaining a regret of order $\widetilde{\mathcal{O}}(\sqrt{T})$.

3. While we use an Explore then Commit algorithm, an optimistic approach is used in Azizzadenesheli et al. (2016) with each episode being characterized by an increasing length. The optimistic approach is supported by the fact that the used policies are exploratory enough to allow state identification since every action is constantly chosen with a minimum probability.

4. Concerning the assumptions, the minimum transition probability assumption (our Assumption 4.1) is not needed in Azizzadenesheli et al. (2016). We recall that this assumption is required in our setting to bound the error in the estimated belief, while memoryless policies do not use a notion of belief and in turn this allows Azizzadenesheli et al. (2016) to get rid of this assumption. Concerning the identifiability assumption, they also have a full-rank assumption on the observation matrix, which Spectral Decomposition techniques require. In addition, they require the transition matrix to be invertible, as a necessary condition to apply SD approaches.

## D   Comparison with Spectral Decomposition technique used in (Zhou et al., 2021)

We devote this section to the comparison of our estimation approach with spectral decomposition techniques (Anandkumar et al., 2014). In particular, we will focus on the comparison with the spectral procedure adopted in the work of Zhou et al. (2021) which faces a similar problem to the one we consider. We start by highlighting the main differences between their work and ours:

- they consider learning both the transition and the observation models, while we assume to know the latter.

- they have a further assumption compared to ours as they require the invertibility of the transition matrix $\boldsymbol{P}$.

- they assume to have access to an optimization oracle that returns the optimal policy for a given known model. Differently, our oracle optimizes the best instantaneous expected reward given the belief on the MABs at each timestep computed using the real transition and observation matrices.

The authors propose the SEEU (Spectral Exploration and Exploitation with UCB) algorithm that alternates between exploration phases used to make parameter estimation and exploitation phases where the actions are pulled according to the computed optimistic policy. During the exploration phases, they use standard spectral decomposition methods to estimate both the observation and the transition models. The guarantees they provide require both Assumption 4.1 and 4.2 and they further require the invertibility of the transition matrix, which is needed for the application of SD approaches. The algorithm they devise reaches $\widetilde{\mathcal{O}}(T^{2/3})$ regret, disregarding logarithmic terms.

To define a comparison with Spectral Decomposition techniques, we need to introduce some quantities that will be helpful in what will follow. We will report some results appearing in Appendix B of Zhou et al. (2021) on spectral decomposition techniques, based on the work of Anandkumar et al. (2014).
In particular, they introduce the following three matrices $\boldsymbol{B}_1, \boldsymbol{B}_2, \boldsymbol{B}_3 \in \mathbb{R}^{IV \times S}$ where each row is associated to a couple $(a, r) \in \mathbb{I} \times \mathbb{V}$ and each column is associated to a state $s \in \mathbb{S}$. They are defined as follows:

$$\boldsymbol{B}_1\big((a, r), s\big) = \Pr(a_{t-1} = a, r_{t-1} = r | s_t = s)$$
$$\boldsymbol{B}_2\big((a, r), s\big) = \Pr(a_t = a, r_t = r | s_t = s)$$
$$\boldsymbol{B}_3\big((a, r), s\big) = \Pr(a_{t+1} = a, r_{t+1} = r | s_t = s)$$

for any $(a, r) \in \mathbb{I} \times \mathbb{V}$ and for any state $s \in \mathbb{S}$.
We can now present the bound on the error of the estimated transition matrix. From Anandkumar et al.

(2014), it can be shown that with a sufficient number of samples $N$ and with probability at least $1 - \delta$, spectral approaches ensure that:

$$\|\boldsymbol{P} - \widehat{\boldsymbol{P}}\|_F \leq C_2 \sqrt{\frac{\log\left(6\frac{I^2 V^2 + IV}{\delta}\right)}{N}}, \tag{25}$$

with

$$C_2 = \frac{4}{\sigma_{\min}(\boldsymbol{B}_2)}\left(S + S^{3/2} * \frac{21}{\sigma_{1,-1}}\right)C_3$$

$$C_3 = 2G\frac{2\sqrt{2}+1}{(1-\theta)\sqrt{\pi_{\min}}}\left(1 + \frac{8\sqrt{2}}{\pi_{\min}^2\sigma^3} + \frac{256}{\pi_{\min}^3\sigma^2}\right),$$

where $\sigma_{1,-1}$ is the smallest nonzero singular value of a covariance matrix computed during the estimation process (see Section 3.1 in Zhou et al. (2021)) and $\sigma = \min\{\sigma_{\min}(\boldsymbol{B}_1), \sigma_{\min}(\boldsymbol{B}_2), \sigma_{\min}(\boldsymbol{B}_3)\}$, where $\sigma_{\min}(\boldsymbol{B}_i)$ represents the smallest nonzero singular value of the matrix $\boldsymbol{B}_i$, for $i = 1, 2, 3$. $\boldsymbol{\pi}$ represents the stationary distribution of the underlying chain and $\pi_{\min} := \min_s \boldsymbol{\pi}(s) \geq \epsilon$. Finally, $\theta$ and $G$ are some mixing rate parameters. Under Assumption 4.1, it is possible to show that we can take $G = 2$ thus having $\theta \leq 1 - \epsilon$. The term $C_2$ that we report here shows a further $\sqrt{S}$ term with respect to the $C_2$ term reported in Zhou et al. (2021). This is due to the fact that the bound reported in Equation 25 uses the Frobenious norm of the difference of the matrices, while Zhou et al. (2021) report the bound in terms of the spectral norm $\|\cdot\|_2$. The additional $\sqrt{S}$ of the $C_2$ term reported here is indeed due to the conversion between the two norms, for which it holds that $\|\boldsymbol{P} - \widehat{\boldsymbol{P}}\|_F \leq \sqrt{S}\|\boldsymbol{P} - \widehat{\boldsymbol{P}}\|_2$.

We can simplify the expression of $C_2$ by reporting a new constant $C_2'$ for which it can be easily shown that $C_2' \leq C_2$. It is defined as follows:

$$C_2' = \frac{16(2\sqrt{2}+1)}{\sigma_{\min}(\boldsymbol{B}_2)\sqrt{\pi_{\min}}}\left(S + S^{3/2} * \frac{21}{\sigma_{1,-1}}\right)\left(1 + \frac{8\sqrt{2}}{\pi_{\min}^2\sigma_{\min}^3(\boldsymbol{B}_2)} + \frac{256}{\pi_{\min}^3\sigma_{\min}^2(\boldsymbol{B}_2)}\right).$$

Having defined the quantity $C_2'$ containing the dependency on the problem parameters of the spectral decomposition approach used in Zhou et al. (2021), we can now consider the guarantees of the estimation procedure reported in Lemma 6.1.

Before proceeding with the comparison, we recall that Lemma 6.1 is expressed with respect to the number $N$ of pulls of each pair of arms, while the total number of samples used corresponds to $N_{tot} = 2NI^2$. By substituting the expression $N = N_{tot}/(2I^2)$ into the bound, we get:

$$\|\boldsymbol{P} - \widehat{\boldsymbol{P}}\|_F \leq \frac{4I^2}{\sigma_{\min}^2(\boldsymbol{O})\pi_{\min}}\sqrt{\frac{S(C + \log(CI^2/\delta))}{(1 - \lambda^{2I^2})N_{tot}}} \leq \frac{4I^2}{\sigma_{\min}^2(\boldsymbol{O})\pi_{\min}^{3/2}}\sqrt{\frac{S(1 + \log(1I^2/\delta))}{(1 - \lambda^{2I^2})N_{tot}}},$$

where in the second inequality we used the fact that the maximum value of $C$ is $1/\pi_{\min}$.

Using this new result, we define our constant $C_{our}$ as:

$$C_{our} = \frac{4I^2}{\sigma_{\min}^2(\boldsymbol{O})\pi_{\min}^{3/2}}\sqrt{\frac{SC}{1 - \lambda^{2I^2}}}.$$

It is now relevant to note the similarities between matrix $\boldsymbol{B}_2$ and our action observation matrix $\boldsymbol{O}$. Given any state $s \in \mathbb{S}$ and any action-reward pair $(a, r)$, we have that:

$$\boldsymbol{O}\big((a,r),s\big)\Pr(a_t = a|s_t = s) = \Pr(r_t = r|s_t = s, a_t = a)\Pr(a_t = a|s_t = s)$$
$$= \Pr(a_t = a, r_t = r|s_t = s)$$
$$= \boldsymbol{B}_2\big((a,r),s\big),$$

where the first equality follows by the definition of $\boldsymbol{O}$ in Equation 2. Furthermore, since the SEEU algorithm samples uniformly over the $I$ actions during the exploration phase, for any $(a, r) \in \mathbb{I} \times \mathbb{V}$ and for any state $s \in \mathbb{S}$, we have that:

$$\boldsymbol{O}\big((a,r), s\big) \Pr(a_t = a | s_t = s) = \boldsymbol{O}\big((a,r), s\big) \frac{1}{I} = \boldsymbol{B}_2\big((a,r), s\big).$$

Considering the minimum singular values of these matrices, the stated result allows also to say:

$$\sigma_{\min}(\boldsymbol{O}) = \sigma_{\min}(\boldsymbol{B}_2) I. \tag{26}$$

We can now rewrite constant $C_{our}$ as:

$$C_{our} = \frac{4I^2}{\sigma_{\min}^2(\boldsymbol{O}) \pi_{\min}^{3/2}} \sqrt{\frac{S}{1 - \lambda^{2I^2}}} = \frac{4}{\sigma_{\min}^2(\boldsymbol{B}_2) \pi_{\min}^{3/2}} \sqrt{\frac{S}{1 - \lambda^{2I^2}}},$$

where the equality directly follows from 26.

We are now ready to compare the constant $C_2'$ of SD approaches with constant $C_{our}$ appearing in our approach[10]. $C_2'$ has a dependency of order $-7/2$ with respect to $\pi_{\min}$ while $C_{our}$ enjoys a dependency of order $-3/2$. By considering instead the number of states $S$, constant $C_2'$ contains a term that scales with order $3/2$, while we have a dependency of order $1/2$. Finally, the dependency on the minimum singular value of the matrix $\boldsymbol{B}_2$ has order $-4$ in $C_2'$ and order $-2$ in $C_{our}$[11]. Again, we recall that these considerations are made on $C_2'$ which is a smaller value than the real one $C_2$ appearing in their bound.

From this analysis, we have shown that our approach enjoys better dependence with respect to SD approaches on all the problem parameters.

## E Useful Lemmas and Deviation Inequalities

This section is devoted to the presentation of some results that are useful in understanding some proofs appearing in Appendix B.

**Lemma E.1.** *(Lemma A.1 in Ramponi et al. (2020)) Let $\boldsymbol{x}, \boldsymbol{y} \in \mathbb{R}^d$ any pair of vectors, then it holds that:*

$$\left\| \frac{\boldsymbol{x}}{\|\boldsymbol{x}\|_2} - \frac{\boldsymbol{y}}{\|\boldsymbol{y}\|_2} \right\|_2 \leq \frac{2\|\boldsymbol{x} - \boldsymbol{y}\|_2}{\max\{\|\boldsymbol{x}\|_2, \|\boldsymbol{y}\|_2\}}.$$

*Proof.* The presented result follows from a sequence of algebraic manipulations:

$$\begin{aligned}
\left\| \frac{\boldsymbol{x}}{\|\boldsymbol{x}\|_2} - \frac{\boldsymbol{y}}{\|\boldsymbol{y}\|_2} \right\|_2 &= \left\| \frac{\boldsymbol{x}}{\|\boldsymbol{x}\|_2} - \frac{\boldsymbol{y}}{\|\boldsymbol{y}\|_2} \pm \frac{\boldsymbol{y}}{\|\boldsymbol{x}\|_2} \right\|_2 \\
&= \left\| \frac{\boldsymbol{x} - \boldsymbol{y}}{\|\boldsymbol{x}\|_2} - \frac{\boldsymbol{y}(\|\boldsymbol{y}\|_2 - \|\boldsymbol{x}\|_2)}{\|\boldsymbol{y}\|_2 \|\boldsymbol{x}\|_2} \right\|_2 \\
&\leq \frac{\|\boldsymbol{x} - \boldsymbol{y}\|_2}{\|\boldsymbol{x}\|_2} + \frac{|\|\boldsymbol{x}\|_2 - \|\boldsymbol{y}\|_2|}{\|\boldsymbol{x}\|_2} \\
&\leq 2 \frac{\|\boldsymbol{x} - \boldsymbol{y}\|_2}{\|\boldsymbol{x}\|_2},
\end{aligned}$$

where the triangular inequality has been applied in the third line and the reverse triangular inequality in the last one, i.e. $|\|\boldsymbol{x}\|_2 - \|\boldsymbol{y}\|_2| \leq \|\boldsymbol{x} - \boldsymbol{y}\|_2$. The result in the lemma can be derived by observing that, for symmetry reasons, the same derivation can be performed by getting $\|\boldsymbol{y}\|_2$. $\square$

---

[10]We only consider the terms that can be directly compared, thus disregarding the term $\sqrt{1 - \lambda^{2I^2}}$.

[11]The empirical impact of this dependency is evident in the numerical comparison in Section 7.3.

### E.1 Concentration Results for Markov Chains

We report here the concentration bound on Theorem 12 from Fan et al. (2021). It is derived from Markov chains with continuous space, but can as well be adapted to our finite space setting.

**Proposition E.1.** *(Concentration Bound for Markov Chains) Let $\{X_i\}_{i \geq 1}$ be a Markov chain with stationary distribution $\boldsymbol{\pi}$ and absolute spectral gap $1 - \lambda$. Suppose the initial distribution $\boldsymbol{\nu}$ is absolutely continuous with respect to the invariant measure $\pi$ and its derivative $\frac{d\nu}{d\pi}$ has a finite p-moment for some $p \in (1, \infty]$. Let $h = p/(p-1) \in [1, \infty)$ and*

$$C = C(\nu, n_0, p) := \begin{cases} 1 + 2^{2/p} \lambda^{2n_0/h} \left\| \frac{d\nu}{d\pi} - 1 \right\|_{\pi,p} & \text{if } p \in (1,2), \\ 1 + \lambda^{n_0} \left\| \frac{d\nu}{d\pi} - 1 \right\|_{\pi,2} & \text{if } p = 2, \\ 1 + 2^{2/h} \lambda^{2n_0/p} \left\| \frac{d\nu}{d\pi} - 1 \right\|_{\pi,p} & \text{if } p \in (2,\infty), \\ \left\| \frac{d\nu}{d\pi} \right\|_{\pi,\infty} = ess\ sup \left\| \frac{d\nu}{d\pi} \right\| & \text{if } p = \infty. \end{cases}$$

*Then, it follows that for any $\epsilon > 0$, uniformly for all bounded function $f : \mathcal{X} \to [a, b]$:*

$$\text{Pr}_{\boldsymbol{\nu}} \left( \frac{1}{n} \sum_{i=n_0+1}^{n_0+n} f(X_i) - \pi(f) > \epsilon \right) \leq C \exp \left( -\frac{1}{h} \cdot \frac{1-\lambda}{1+\lambda} \cdot \frac{n\epsilon^2}{2(b-a)^2/4} \right),$$

*where $Pr_{\boldsymbol{\nu}}$ emphasizes the dependence on the initial distribution, we have $\pi(f) = \int f(x)\pi(x)dx$ and also:*

$$\|g\|_{\pi,p} := (\pi(g^p))^{1/p} = \left( \int g^p(x)\pi(x)\, dx \right)^{1/p}.$$

### E.2 Concentration Results for Discrete Distributions

We present here some concentration bounds that can be derived from McDiarmid's inequality (McDiarmid, 1989). The first proposition considers samples that are i.i.d. while the second result considers samples coming from a Markov chain and is indeed used in the proof of Lemma 6.1.

**Proposition E.2.** *(Concentration for Discrete Distributions with Independent Samples (Proposition 19 (Hsu et al., 2012)) Let z be a discrete random variable that takes values in $\{1, \ldots, d\}$, distributed according to $\mathbf{q}$. We write $\mathbf{q}$ as a vector $\mathbf{q} = [Pr(z = j)]_{j=1}^d$. Assume we have $N$ i.i.d. samples, and that our empirical estimate of $\mathbf{q}$ is $[\widehat{q}]_j = \sum_{i=1}^N \mathbb{1}[z_i = j]/N$.*
*We have that $\forall \epsilon > 0$:*

$$Pr \left( \|\widehat{\mathbf{q}} - \mathbf{q}\|_2 \geq \frac{1}{\sqrt{N}} + \epsilon \right) \leq \exp \left( -N\epsilon^2 \right).$$

The presented result can be also written for dependent samples coming from a Markov chain. For this case, it is possible to discount the number of samples based on the modulus of the second largest eigenvalue of the transition matrix $\boldsymbol{P}$ (as reported in Appendix A of Hsu et al. (2012)). Also, in Fan et al. (2021), they state that $\frac{1+\lambda}{1-\lambda}N$ Markov chain samples are needed to achieve the same accuracy with $N$ independent samples in the naive Monte Carlo method, with $\lambda$ being the modulus of the second largest eigenvalue of $\boldsymbol{P}$.

**Proposition E.3.** *(Concentration for Discrete Distributions with Samples coming from a Markov Chain) Let z be a discrete random variable that takes values in $\{1, \ldots, d\}$, distributed according to $\mathbf{q}$. We write $\mathbf{q}$ as a vector $\mathbf{q} = [Pr(z = j)]_{j=1}^d$. Assume we have $N$ samples coming from a Markov process having a transition matrix with the modulus of the second largest eigenvalue $\lambda$, and assume that the Markov chain starts from its stationary distribution $\boldsymbol{\pi}$. Our empirical estimate of $\mathbf{q}$ is $[\widehat{q}]_j = \sum_{i=1}^N \mathbb{1}[z_i = j]/N$.*
*We have that $\forall \epsilon > 0$:*

$$Pr \left( \|\widehat{\mathbf{q}} - \mathbf{q}\|_2 \geq \sqrt{\left( \frac{1+\lambda}{1-\lambda} \right) \frac{1}{N}} + \epsilon \right) \leq \exp \left( -\frac{1-\lambda}{1+\lambda} \cdot N\epsilon^2 \right).$$

The previous result can be as well extended to the case where the starting distribution is arbitrary. In the following, we will provide the statement and its proof.

**Proposition E.4.** *(Concentration for Discrete Distributions with Samples coming from a Markov Chain Starting From Arbitrary Distribution) Let* z *be a discrete random variable that takes values in* $\{1, \ldots, d\}$, *distributed according to* $\mathbf{q}$. *We write* $\mathbf{q}$ *as a vector* $\mathbf{q} = [Pr(z = j)]_{j=1}^d$. *Assume we have* $N$ *samples coming from a Markov process having a transition matrix with the modulus of the second largest eigenvalue* $\lambda$, *assume that the Markov chain has stationary distribution* $\boldsymbol{\pi}$ *and that the chain starts from an arbitrary distribution* $\boldsymbol{\nu}$. *Our empirical estimate of* $\mathbf{q}$ *is* $[\widehat{q}]_j = \sum_{i=1}^N \mathbb{1}[z_i = j]/N$.
*We have that* $\forall \epsilon > 0$ :

$$Pr\left( \|\widehat{\mathbf{q}} - \mathbf{q}\|_2 \geq \sqrt{\left( \frac{1+\lambda}{1-\lambda} \right) \frac{C}{N}} + \epsilon \right) \leq C \exp\left( -\frac{1-\lambda}{1+\lambda} \cdot N\epsilon^2 \right),$$

*where* $C := C(\nu, 0, \infty) = \|\frac{\nu}{\pi}\|_\infty$ *is defined as in Proposition E.1 by setting* $n_0 = 0$ *and* $p = \infty$, *while* $\frac{\nu}{\pi}$ *represents the element-wise ratio between the vectors of probability distributions[12].*

*Proof.* First of all, we start by reporting an existing concentration result obtained by combining results in Hsu et al. (2012) and Fan et al. (2021). In particular, we have:

$$Pr_{\boldsymbol{\nu}}\left( \|\widehat{\mathbf{q}} - \mathbf{q}\|_2 \geq \mathbb{E}_{\boldsymbol{\nu}}\left[ \|\widehat{\mathbf{q}} - \mathbf{q}\|_2 \right] + \epsilon \right) \leq C \exp\left( -\frac{1}{h} \cdot \frac{1-\lambda}{1+\lambda} \cdot N\epsilon^2 \right), \tag{P.17}$$

where $C := C(\nu, 0, p)$ and $h$ are defined as in E.1, while we used $Pr_{\boldsymbol{\nu}}$ and $\mathbb{E}_{\boldsymbol{\nu}}$ to emphasize the dependence on the initial distribution $\boldsymbol{\nu}$.

To proceed with the analysis, we need to bound $\mathbb{E}_{\boldsymbol{\nu}}\left[ \|\widehat{\mathbf{q}} - \mathbf{q}\|_2 \right]$. In the following steps, we show how to do it using some ideas appearing in Fan et al. (2021). We have:

$$
\begin{aligned}
\mathbb{E}_{\boldsymbol{\nu}}\left[ \|\mathbf{q} - \widehat{\mathbf{q}}\|_2 \right]^2 &\leq \mathbb{E}_{\boldsymbol{\nu}}\left[ \|\mathbf{q} - \widehat{\mathbf{q}}\|_2^2 \right] \\
&= \mathbb{E}_{\boldsymbol{\nu}}\left[ \sum_{y=1}^d |\mathbf{q}_y - \widehat{\mathbf{q}}_y|^2 \right] \\
&= \sum_{y=1}^d \mathbb{E}_{\boldsymbol{\nu}}\left[ (\mathbf{q}_y - \widehat{\mathbf{q}}_y)^2 \right] \\
&= \sum_{y=1}^d \mathbb{E}_{\boldsymbol{\nu}}\left[ \left( \mathbf{q}_y - \frac{1}{n}\sum_{i=1}^n \mathbb{1}\{f(S_i) = y\} \right)^2 \right] \\
&= \sum_{y=1}^d \mathbb{E}_{\boldsymbol{\pi}}\left[ \frac{\boldsymbol{\nu}(S_1)}{\mathbf{q}(S_1)}\left( \mathbf{q}_y - \frac{1}{n}\sum_{i=1}^n \mathbb{1}\{f(S_i) = y\} \right)^2 \right] && \text{[Change of Measure]} \\
&\leq \sum_{y=1}^d \mathbb{E}_{\boldsymbol{\pi}}\left[ \left( \frac{\boldsymbol{\nu}(S_1)}{\mathbf{q}(S_1)} \right)^p \right]^{1/p} \mathbb{E}_{\boldsymbol{\pi}}\left[ \left( \mathbf{q}_y - \widehat{\mathbf{q}}_y \right)^{2h} \right]^{1/h} && \text{[Hölder]} \\
&= \left\| \frac{\boldsymbol{\nu}}{\boldsymbol{\pi}} \right\|_\infty \sum_{y=1}^d \mathbb{E}_{\boldsymbol{\pi}}\left[ \left( \mathbf{q}_y - \widehat{\mathbf{q}}_y \right)^2 \right] && [p = \infty,\ h = 1]
\end{aligned}
$$

---

[12]Differently from the distributions defined in Proposition E.1, here $\boldsymbol{\nu}$ and $\boldsymbol{\pi}$ are probability mass functions defined on a finite state space.

$$= \left\|\frac{\boldsymbol{\nu}}{\boldsymbol{\pi}}\right\|_\infty \sum_{y=1}^{d} var[\widehat{\mathbf{q}}_y] \qquad\qquad \text{[Variance Definition]}$$

$$\leq \left\|\frac{\boldsymbol{\nu}}{\boldsymbol{\pi}}\right\|_\infty \sum_{y=1}^{d} \left(\frac{1+\lambda}{1-\lambda}\right) \cdot \frac{\boldsymbol{q}_y}{n} \qquad\qquad \text{[Result from Fan et al. (2021)]},$$

$$\leq \left\|\frac{\boldsymbol{\nu}}{\boldsymbol{\pi}}\right\|_\infty \left(\frac{1+\lambda}{1-\lambda}\right) \cdot \frac{1}{n} \qquad\qquad \left[\text{Since } \sum_{y=1}^{d} \boldsymbol{q}_y = 1\right],$$

$$= \frac{C}{n}\left(\frac{1+\lambda}{1-\lambda}\right), \tag{P.18}$$

where in the derivation we made explicit the dependence of each term $\widehat{\mathbf{q}}_y$ from the sequence of stochastic underlying latent states $(S_i)_{i\in[n]}$, while $\mathbb{1}$ represents the indicator function, while $f(S)$ represents the outcome of a categorical distribution with size $d$ and parameters based on the state $S$[13].

From the obtained result, we can see that starting from a distribution that is different from the stationary one leads to a further multiplicative term $C$ in the final concentration result.

By using $p = \infty$ and $h = 1$ we obtain the bound appearing in P.18. Finally, by substituting this result into Equation P.17 we obtain the result in the claim. $\qquad\square$

### E.3 Bounds on the Error of the Estimated Belief

We present here the result appearing in Zhou et al. (2021) that controls the error in the estimated belief. They consider a setting with Bernoulli rewards $\mathbb{V} = \{0, 1\}$ which results in an action observation matrix $\boldsymbol{O} \in \mathbb{R}^{2I \times S}$. Since the rewards are Bernulli, the dimension of the action observation matrix can be halved, because half of the probabilities contained are complementary to the other half. In particular, we will use the reward matrix $\boldsymbol{\mu} \in \mathbb{R}^{I \times S}$ such that each of its elements is defined as in Equation 4 that we report here for simplicity.

$$\boldsymbol{\mu}(a, s) = \sum_{r \in \mathbb{V}} r \boldsymbol{O}\big((a, r), s\big) \qquad \forall\, a \in \mathbb{I},\ s \in \mathbb{S}.$$

**Proposition E.5.** *(Controlling the belief error (Zhou et al., 2021)) Assume to have a transition matrix $\boldsymbol{P}$ of size $S \times S$ with minimum entry $\epsilon > 0$ and reward matrix $\boldsymbol{\mu} \in \mathbb{R}^{I \times S}$. Let's assume to have an estimator $(\widehat{\boldsymbol{\mu}}, \widehat{\boldsymbol{P}})$ of the true model parameters $(\boldsymbol{\mu}, \boldsymbol{P})$. For an arbitrary reward-action sequence $\{r_{1:t}, a_{i:t}\}_{t\geq 1}$, let $\widehat{\mathbf{b}}_t$ and $\mathbf{b}_t$ be the corresponding beliefs in period t under $(\widehat{\boldsymbol{\mu}}, \widehat{\boldsymbol{P}})$ and $(\boldsymbol{\mu}, \boldsymbol{P})$ respectively. Then there exists constants $L_1$ and $L_2$ such that:*

$$\|\widehat{\mathbf{b}}_t - \mathbf{b}_t\|_1 \leq L_1 \|\widehat{\boldsymbol{\mu}} - \boldsymbol{\mu}\|_1 + L_2 \|\widehat{\boldsymbol{P}} - \boldsymbol{P}\|_F,$$

*where $L_1 = 4S(\frac{1-\epsilon}{\epsilon})^2 / \min\{\boldsymbol{\mu}_{\min}, 1 - \boldsymbol{\mu}_{\max}\}$, $L_2 = \frac{4S(1-\epsilon)^2}{\epsilon^3} + \sqrt{S}$, $\|\cdot\|_F$ is the Frobenius norm, $\boldsymbol{\mu}_{\max}$ and $\boldsymbol{\mu}_{\min}$ are the maximum and minimum element of the matrix $\boldsymbol{\mu}$ respectively.*

We adapt the previous result to our setting with a discrete set of rewards and with a known observation matrix $\boldsymbol{O}$. It can be expressed as follows.

**Proposition E.6.** *(Controlling the belief error in Switching Latent Bandits) Assume to have a transition matrix $\boldsymbol{P}$ of size $S \times S$ with minimum entry $\epsilon > 0$. Given a model with parameters $(\boldsymbol{O}, \boldsymbol{P})$, we assume to know the observation model $\boldsymbol{O}$ and to have an estimation of the transition matrix $\widehat{\boldsymbol{P}}$. For an arbitrary reward-action sequence $\{r_{1:t}, a_{i:t}\}_{t\geq 1}$, let $\widehat{\mathbf{b}}_t$ and $\mathbf{b}_t$ be the corresponding beliefs in period t under $(\boldsymbol{O}, \widehat{\boldsymbol{P}})$ and $(\boldsymbol{O}, \boldsymbol{P})$ respectively. Then there exists a constant $L$ such that:*

$$\|\widehat{\mathbf{b}}_t - \mathbf{b}_t\|_1 \leq L \|\widehat{\boldsymbol{P}} - \boldsymbol{P}\|_F,$$

*where $L = \frac{4S(1-\epsilon)^2}{\epsilon^3} + \sqrt{S}$ and $\|\cdot\|_F$ is the Frobenius norm.*

---

[13]In our context, the $f$ function represents the observation distribution given the used action and the underlying state. Assuming to condition on action $a$ (we refer to it as $f_a$), we will have $f_a(s) = \boldsymbol{O}\big((a, \cdot), s\big)$.

# F  Continuous Reward Distributions

In the main paper, we focused on the case where the set of possible observations is discrete and the distribution of the observations $\boldsymbol{O}\big((a,\cdot),s\big) \in \Delta(\mathbb{V})$ is categorical, for each latent state $s \in \mathbb{S}$ and action $a \in \mathbb{I}$. In this section, we show how we can extend the estimation procedure to also handle continuous reward distributions. Formally, if we consider having a Switching Latent Bandit problem with continuous rewards and a number $S$ of bandits and a number $I$ of actions available for each bandit, there will be $IS$ potentially different continuous reward distributions $\Pr(\cdot|s,a)$ for each latent state $s \in \mathbb{S}$ and action $a \in \mathbb{I}$. If we assume to discretize each reward distribution into $U$ consecutive intervals, we will have $U-1$ splitting points. By considering the ordered set of splitting points and taking two consecutive splitting points $u_h$ and $u_k$ for which holds that $u_h < u_k$, we can define the interval $\mathcal{I}_{hk} = (u_h, u_k]$. The probability that a realization from a continuous distribution $\Pr(\cdot|s,a)$ falls within interval $\mathcal{I}_{hk}$ is defined as:

$$\Pr(r \in \mathcal{I}_{hk}|s,a)) = \int_{u_h}^{u_k} \Pr(dr|s,a)\ dr.$$

Of course, if we are able to exactly compute the integrals in the previous formulation we will not introduce any error in the discretization process. By applying the same procedure for all the $U$ intervals identified, we can define the parameters of the new categorical distribution. This procedure is then applied to all the continuous probabilities $\Pr(\cdot|s,a)$ for each $s \in \mathbb{S}$ and $a \in \mathbb{I}$ using the same splitting points and we finally obtain a new action observation matrix of size $IU \times S$, which should of course satisfy Assumption 4.2.
From this point on, we can build the new reference matrix and we can proceed with the estimation procedure presented in Algorithm 1. Whenever a reward is observed during the estimation procedure, the count vector is updated by considering the interval to which the observed reward belongs.
It is an interesting problem to determine in this setting the number of suitable splits and the location of the split points that leads to an action observation matrix with higher $\sigma_{\min}(\boldsymbol{O})$.

Another issue arises when the environment comprises numerous but finite observations. In such scenarios, we can employ the inverse approach by clustering some observations, thereby reducing the scale of the problem. By selecting a number of clusters $C < V$, we can divide the observations into distinct groups. This allows us to utilize cluster-level probabilities (obtained by summing probabilities of the single observations) to construct a new action observation matrix and then proceed with the standard estimation procedure.