# OpenReview forum: "Switching Latent Bandits"
_TMLR — Accepted by TMLR_

### Review · Reviewer_UHGH · 2024-03-26

**Summary Of Contributions:**

The authors consider the problem of latent bandits where the reward of arms depend on an unobserved state, and where the learner knows the conditional distribution of the reward knowing both the state and the chosen action. The problem consists in estimating the state transition probability through the rewards. This also enable accurate estimation of the belief state.

The authors propose an algorithm which starts by (possibly) pruning the set of arms, then explores them uniformly to infer the state transition probability matrix by inverting the (linear) relationship between state transition probability matrix and rewards transition probability matrix.

**Audience:**

Yes

**Broader Impact Concerns:**

Not applicable.

**Claims And Evidence:**

Yes

**Requested Changes:**

See above.

**Strengths And Weaknesses:**

Strengths

- The paper is easy to understand
- The authors present both algorithms, regret upper bounds as well as numerical simulations to illustrate the efficacy of the approach

Weaknesses
- I have doubts about the regret definition/model/objective: the authors are purely concerned with making sure that the estimated belief state ${\hat b}_t$ is an accurate estimate of the actual belief state $b_t$, but they do not care at all about the collected rewards. In fact, if there exists a very informative action (that allows to quickly guess the state) but has 0 reward, playing it incurs no regret. Not only is this counter intuitive, but i'm not even sure that the problem at hand is a bandit problem. It is essentially just an estimation problem. The regret definition would make sense if the agent were forced to play the "estimated belief" policy maximizing $\mu(a)^\top {\hat b}_t$, but it is not the case here and the learner may play as she wishes.

- The strategy (uniform exploration) seems overly simplistic and probably highly suboptimal. The authors argue that, in order to counteract this problem, when there are too many non informative arms one can just exclude arms a priori, and concentrate on a well chosen, small subset of arms to apply uniform exploration on this subset, but I do not believe that this is true in general. Consider the following example where one has $K$ states $\mathbb{S}=\\{1,...,K\\}$, and K-1 arms $\mathbb{I}=\\{1,...,K-1\\}$, and the rewards are binary $\mathcal{V}=\\{0,1\\}$ such that $O((a,1),s) = 1$ if $a=s$ and $O((a,1),s) = 0$ if $a \ne s$. Essentially, sampling action $a$ is equivalent to asking the question "is the system in state a ?" and getting an exact answer. For the algorithm to be consistent, we cannot exclude any arm from the set of arms, otherwise if $a$ is excluded, it becomes impossible to guess (for instance) the probabilities $P(s,s')$ with $s=a$. So one has to sample uniformly over all the arms if we use the authors strategy. Now consider two fixed states $s_1$ and $s_2$, a positive number $\epsilon$ arbitrarily small and $P$ such that $P(s,s') \le \epsilon$ if $s,s' \not\in \\{s_1,s_2\\}$ (i.e the chain concentrates mostly on states $s_1$ and $s_2$). Then clearly, whenever we select actions that are not in $\{s_1,s_2\}$ we acquire little to no information about $P$ and collect no reward either. So only at most a fraction $4/K^2$ of the samples collected are useful, and the regret is at least $O(K^2)$ greater than what one would obtain with an optimal approach. I believe that, to design good algorithms for this problem (like in all bandit problems) one must sample adaptively to adjust the exploration strategy to the emerging information.
- Why is Assumption 4.1 necessary ? This seems like a very strong assumption and clearly unnecessary for ergodicity. For instance very simple Markov chains like birth and death processes do not verify this assumption. This assumption greatly reduces the applicability of the results.
- While the writing is very clear, I believe that the article is much too long. For instance Section "5 Proposed Approach" it is almost 3 pages of simple definitions and properties. Especially since the proposed approach is quite intuitive to a reader with a basic understanding of Markov chains: one simply estimates the transition probability matrix of rewards, and then one inverts a linear system to retrieve the transition probability of the states.


Minor remarks:
- "The regret of a policy $\theta$ on a bandit instance is defined as:" -> the authors do not define what a "policy" is at that point.
- "is the unique stationary distribution of the chain" -> it might be worth reminding the reader that the stationary distribution is the solution to $\pi = \pi P$ at this point of the text.
- "Each MAB is characterized by a finite set of discrete arms ... " This phrase is misleading because it sounds like there may be different sets of arms for different MAB instances, whereas here, all MABs have the same set of arms and rewards.
- "The distribution of rewards Pr(·|s, a) conditioned on MAB instance s and action a is categorical" By categorical do you mean discrete ?
- "Stationary Distribution of Consecutive States" Why not simply define $W(s,s') = \pi(s) P(s,s')$ ?

---

> ### Author Response · Authors · 2024-04-16
>
> We thank the reviewer for their time in reviewing our manuscript. In the following, we will address the main points raised in the comments.
>
> ## Objective Definition
>
> **Q.1 The regret definition would make sense if the agent were forced to play the "estimated belief" policy maximizing $\mu(a)^T \widehat{b}_t $, but it is not the case here and the learner may play as she wishes.**
>
> **A.1** In our setting, the goal of the agent is indeed to play the action maximizing $\mu(a)^T \widehat{b}_t$ as shown in the paragraph *Learning objective* (Section 4) and in particular in Equation 5 where we define how the action is chosen by the agent. The regret is instead defined against the oracle that updates the belief using the exact knowledge of both the observation and transition model and chooses the action maximizing $\mu(a)^T b_t$. The definition of regret we use in our problem is reported in Equation 6 and differs from the standard definition of regret reported in Equation 1.
>
> Since we do not have access to the transition model $\boldsymbol{P}$ and we need an estimate of it in order to update the belief (Equation 3), our approach is to use an Explore-Then-Commit type of algorithm (SL-EC algorithm) that uses the Explore phase to collect samples that allow estimating the transition model $\widehat{\boldsymbol{P}}$, while in the second phase the agent plays the action maximizing the quantity $\mu(a)^T \widehat{b}_t$ and updating the belief using the estimated transition model $\widehat{\boldsymbol{P}}$. The Explore part uses our devised estimation procedure (Algorithm 1) and during this part, we pay maximum regret since arms are pulled in a round-robin fashion.
>
> **Q.2 If there exists a very informative action (that allows to quickly guess the state) but has 0 reward, playing it incurs no regret.**
>
> **A.2** If the algorithm is in its Explore phase, it will always pay maximum regret (1 for each time step in the Explore phase). Differently, if the Commit phase has started, playing an action that has an expected reward of 0 would lead to a regret that is non-null if, a different action, ensures a higher expected reward $\mu(a)^T b_t$. In conclusion, **we always pay regret if we play a suboptimal action**.
>
>
> ## Exploration Strategy
>
> **Q1: The authors argue that, in order to counteract this problem, when there are too many non-informative arms one can just exclude arms a priori, and concentrate on a well-chosen, small subset of arms to apply uniform exploration on this subset, but I do not believe that this is true in general.**
>
> **A.1:** The arm selection procedure allows for reducing the number of arms selecting the ones that jointly provide more information. We do not get why the reviewer says that this is not generally true. As specified in Section 5.1.1, **the conditions under which this technique is applicable is that the observation matrix defined by the subset of selected arms should satisfy Assumption 4.2**. In the example presented by the reviewer, all the arms are informative and necessary to learn the transition model, thus preventing the use of the arm selection procedure. Indeed, it can be easily checked that by removing any of the arms from the observation matrix of the proposed example, Assumption 4.2 is no longer satisfied.
>
> **Q.2 Proposed Example**
>
> **A.2:** While we agree with the reviewer that more suitable exploration strategies can be devised for the presented example, we would like to clarify that our algorithm tackles the more general class of problems where similar conditions (those defined in the mentioned example) may not be verified.
>
> The reviewer proposes to use a technique that adaptively changes the exploration strategy based on the problem estimates. These approaches are interesting and related to the set of techniques based on Information-Directed Sampling (IDS) [1] which aims at balancing a measure of the regret suffered by a stochastic policy and a measure of the information gain obtained when playing that policy.
> For our specific setting, a way that allows using IDS techniques can be to consider a measure of regret that holds in expectation with respect to the estimated state distribution $\widehat{d}(s)$ of the chain, while the measure of information gain can be determined by the minimum singular value of a matrix that is obtained by combining the action probabilities of the chosen stochastic policy with the corresponding rows of the action observation matrix.
>
> While IDS approaches can be of practical interest, using them would not allow for improving our theoretical guarantees.
> We will show two ways of using an IDS-based algorithm in our setting and how they both fail to improve theoretical results.

---

> ### Author Response · Authors · 2024-04-16
>
> **First Case:** The first approach involves employing **the IDS strategy throughout the entirety of the interaction process**. This can be done by dividing the interaction into different episodes. The IDS policy is computed at the beginning of each episode and remains fixed throughout the duration of that episode.
> Samples collected during the episode are used to estimate the transition model and then a new IDS policy is computed using the updated estimates of the state visitation distribution $\widehat{d}(s)$. However, if we consider using the oracle that computes the real belief and plays the action maximizing $\mu(a)^T b_t$ (as the currently used oracle), **the IDS approach would suffer linear regret**, since the stochastic policy used in IDS would choose actions only based on the expected state distribution, without leveraging the information about the belief state and the underlying dynamics of the chain.
>
> **Second Case:** The second approach is devising a **two-phase algorithm that alternates between exploration and exploitation phases**. In this method, we would utilize the IDS procedure during the estimation phases and the belief-based policy that chooses the action maximizing $\mu(a)^T \widehat{b}_t$ during the exploitation phases. Specifically, the stochastic policy derived from the IDS approach is computed at the beginning of each exploration phase and remains fixed until its conclusion.
> While using the IDS policy during each estimation phase may reduce the regret compared to the uniform exploration strategy, this may come at the cost of having a poorer estimation of the transition model compared to our current approach, given the same number of used samples.
> This discrepancy becomes apparent in the following adversarial scenario. Let's consider an instance where arms are categorized into two subsets: informative arms (they give information about the underlying chain) but have almost zero-reward and a second subset of good-rewarding arms carrying no information about the underlying state (for example the observation probability induced by that arm is equal for all the states s). Using the IDS procedure during each exploration phase would thus define a policy that adequately balances between the two subsets. **However, this will result in a suboptimal estimation of the transition model due to the selection of less informative samples, consequently leading to higher regret during the subsequent exploitation phase**.
> Moreover, regardless of the specific problem instance, **the drawback of using the IDS approach is that its theoretical results hold in expectation with respect to the state visitation distribution, thus not considering the temporal relation between states that is induced by the chain**. Differently, the theoretical results of the exploitation part rely on a frequentist analysis. These results cannot be directly combined unless we choose to upper bound the regret of the IDS phases with the maximum value (as we do in our current analysis). **To sum up, we would again pay maximum regret in the exploration phase, and potentially suffer a higher regret during the exploitation phase, due to the suboptimality of the estimation of the IDS procedure.**
>
> To conclude, given that employing these techniques does not enhance the theoretical guarantees of our algorithm, we opted for a different approach that completely separates the exploration from the exploitation phase by using an Explore-Then-Commit methodology, while providing an offline arm selection procedure for the Exploration part.

---

> ### Author Response · Authors · 2024-04-16
>
> ## Assumption 4.1
>
> **Q: Why is Assumption 4.1 necessary? This seems like a very strong assumption and clearly unnecessary for ergodicity.**
>
> **A:** We agree with the reviewer that this assumption may be strong and is not necessary for ergodicity. **However, we require this condition to bound the 1-norm error between the belief computed using the real transition matrix and the one computed using the transition matrix estimated with Algorithm 1**. This assumption is mainly technical and inspired by the work of [2] that proposes this bound on the belief that we report in Proposition D.3. Indeed we use a result derived from it (reported in Proposition D.4) in the proof of Theorem 6.1, as shown in the lines preceding line (P.16) in Appendix B.2.
>
> This assumption is reported for the same purpose in many works dealing with partially observable problems in the average reward setting. To name a few, we consider the work of [3, 4, 5]. It has been employed as well in the more complex POMDP setting such as the work of [6].
> Other works that do not explicitly use this assumption employ a similar one, such as the work of [7], where their Assumption 3 shows how the estimation error of the belief decreases with a convenient rate with respect to the number of samples observed. Indeed, the statement made in their Assumption 3 has been proved to hold only when Assumption 4.1 is satisfied.
> We refer the reviewer to Section 4.1 for more details on this aspect.
>
> **Additionally, we highlight that since this assumption is only needed to bound the error in the estimated belief, we do not require it in the estimation procedure described in Algorithm 1**. Indeed, Lemma 6.1 which provides theoretical guarantees for Algorithm 1 does not need to satisfy Assumption 1, but it only requires the Markov chain to be ergodic.
>
>
> ## Article Length
>
> **Q: While the writing is very clear, I believe that the article is much too long.**
>
> **A:** We are happy that the writing and the presentation of the problem are clear. We chose to dedicate more space and details to this part following the suggestions received in the previous version of the work. Specifically, this part was not completely understood in the previous version and this aspect made us revise it to achieve greater clarity and eliminate any ambiguities in the notation and in the definitions.
> However, if necessary, we are ready to condense this section while preserving its clarity.
>
> ## Minor Remarks
>
> - **Definition of $\theta$**: we will certainly provide in the revised version the definition of policy in the common Bandit setting.
> - **Formulation of the unique distribution**: we thank the reviewer for the suggestion. We will move the formulation from Section 5.1 to the preliminaries.
> - **"Each MAB is characterized by a finite set of discrete arms ... "**: we thank the reviewer for the suggestion. We will change this sentence to "Each MAB is characterized by the same sets of finite arms and finite rewards." in order to avoid ambiguities.
> - **Categorical Distribution**: in this sentence, we refer to the distribution over the observations. Since the observations are discrete, the distribution we assume over these discrete values is of type *categorical*.
> - **Stationary Distribution of Consecutive States**: we thank the reviewer for the suggestion. We opted for the matrix notation in our definition but the expression suggested by the reviewer seems more direct and avoids the definition of matrix $\{\boldsymbol{\Pi}}$. We will modify it and present the definition in this suggested form.

---

> ### Author Response · Authors · 2024-04-16
>
> ## References
>
> [1] Daniel Russo and Benjamin Van Roy. Learning to Optimize via Information-Directed Sampling. In
> Advances in Neural Information Processing Systems 27, pages 1583–1591. 2014.
>
> [2] Yohann De Castro, Elisabeth Gassiat, and Sylvain Le Corff. Consistent estimation of the filtering and
> marginal smoothing distributions in nonparametric hidden markov models. IEEE Transactions on Information Theory, 63(8):4758–4777, 2017.
>
> [3] Xiang Zhou, Yi Xiong, Ningyuan Chen, and Xuefeng Gao. Regime switching bandits. In A. Beygelzimer, Y. Dauphin, P. Liang, and J. Wortman Vaughan (eds.), Advances in Neural Information Processing Systems, 2021.
>
> [4] Bowen Jiang, Bo Jiang, Jian Li, Tao Lin, Xinbing Wang, and Chenghu Zhou. Online restless bandits with
> unobserved states. In Proceedings of the 40th International Conference on Machine Learning, 2023.
>
> [5] Robert Mattila, Cristian Rojas, Eric Moulines, Vikram Krishnamurthy, and Bo Wahlberg. Fast and consistent learning of hidden Markov models by incorporating non-consecutive correlations. Proceedings of the 37th International Conference on Machine Learning. PMLR, 13–18 Jul 2020.
>
> [6] Yi Xiong, Ningyuan Chen, Xuefeng Gao, and Xiang Zhou. Sublinear regret for learning pomdps, 2022.
>
> [7] Mehdi Jafarnia Jahromi, Rahul Jain, and Ashutosh Nayyar. Online learning for unknown partially observable
> mdps. In Proceedings of The 25th International Conference on Artificial Intelligence and Statistics, 2022.

---

### Review · Reviewer_vJgb · 2024-04-11

**Summary Of Contributions:**

This paper studies a latent bandit problem where the latent state evolves according to an unknown Markov chain. The authors assume perfect knowledge of reward distribution of each arm, conditioned on the latent state. They introduce a novel estimation method of the transition matrix that leverage this knowledge. This result is later used in the proposed explore-then-commit algorithm SL-EC. This algorithm enjoys a $\tilde{O}(T^{2/3})$ regret bound and the authors illustrate that the algorithm has an empirical advantage over baselines.

**Audience:**

Yes

**Broader Impact Concerns:**

No concerns.

**Claims And Evidence:**

Yes

**Requested Changes:**

I have the following requested changes.


### Critical
- Discuss whether the scaling in $\sigma_{\min}$, $\pi_{\min}$, and $T$ is optimal or if it can be improved. If possible, please state a lower bound on the cumulative regret.
- Address the weaknesses in the empirical section (see weaknesses). An experiment where the reward model is misspecified would also be nice to see.

### Would improve the paper
- I think the paper would benefit from a discussion on applications where the reward model is perfectly known.
- It would improve the theory if the results wouldn't require the chain to start from the stationary distribution.
- Discuss why use explore-then-commit for this problem and what the challenges with implementing more competitive approaches, like those based on optimism or Thompson sampling, are.

**Strengths And Weaknesses:**

### Strengths:
- This paper studies an interesting problem that, to the best of my knowledge, is underexplored in the literature.
- The paper is well-written.
- The estimator seems novel.

### Weaknesses:
- The approach requires full knowledge of the reward model for each latent state which is restrictive. Are there any real-world problems where this is true? Does your algorithm work well even when we only have access to an approximation of the reward distributions ?
- The $T^{2/3}$ scaling is undesirable and I think it might be unavoidable when using explore-then-commit. Why go for explore-then-commit rather than approaches based UCB or Thompson sampling that usually enjoys much better regret? Are there any particular challenges with applying these approaches to this problem?
- Both $\sigma_{\min}$ and $\pi_{\min}$ can be very small in practice and my understanding is that this could make the bound in Thm 6.1 vacuous. This since it could happen that $T_0 > T$ even for moderately large $T$.  It seems like this undesirable scaling is due to the fact that we try to perfectly estimate $P$ and do we really need to do that to have good regret?
- The regret bound holds only under the assumption that the chain starts from its stationary distribution. The authors state that similar results can be shown without this assumption. Why not have this stronger result in the paper?
- The empirical evaluation can be improved.
    - SL-EC is not evaluated using the budget suggested by Eq 17 and instead uses scaled values. The amount of scaling is not presented in the paper. I understand that what is suggested by theory often is conservative in practice but having a version of SL-EC that uses the budget suggested by Eq 17 would help making sense of the theoretical results.
    - Plotting the R(t) for a single horizon, as in Figure 2, is not very informative. We know already that the algorithm has linear regret up to $t=T_0$ and then either zero regret or with ,low probability, linear regret. I suggest that the authors also plot the regret over several horizons, i.e $R(T)$, to illustrate the dependence in $T$ (since the budget $T_0$ is a function of $T$). This would tell us for which horizons the SL-EC has an advantage over baselines.
    - The confidence intervals in the empirical section looks too good to be true given the small number of seeds , e.g. see Fig 2 where only 5 seeds are used.  I wonder if this is due to the fact that the cumulative regret has a multimodal distribution and the CI wrongly assumes a Gaussian distribution. Note that with probability $1-\delta$ the algorithm suffers no regret after $T_0$ and with probability $\delta$ it suffers linear regret. Thus, with probability $(1-\delta)^k$, k being the number of seeds, the sample variance in instant regret for any $t>T_0$ will be zero which will result in an incorrect CI around cumulative regret that has a very small width. I suggest that the authors run the experiments using more random seeds.

---

> ### Author Response · Authors · 2024-04-26
>
> We thank the reviewer for their time in reviewing our manuscript. In the following, we will tackle the main points raised in the comments.
>
> ## Real-World Problems
>
> **Q: The approach requires full knowledge of the reward model for each latent state which is restrictive. Are there any real-world problems where this is true? Does your algorithm work well even when we only have access to an approximation of the reward distributions?**
>
> **A:** Our work builds on the assumption of perfectly knowing the reward model and we provide theoretical guarantees under this condition. **However, our approach works empirically well also with approximate reward models.** We provide numerical simulations showing the efficacy of our approach under different misspecification levels of the reward model. We refer to Section 2 of the document reported at the end of our answers for experiments and details on this aspect. From the results reported in the experiments, we can conclude that our approach can be as well extended to scenarios with approximate reward models. Of course, the quality of the final transition model will depend on the approximation of the provided reward model.
> This aspect allows extending the applicability of our approach also to approximate observation models.
>
> There are various scenarios in which it is possible to assume knowledge (or approximate knowledge) of the observation model.
>
> Some of these include cases where the latent states characterize different working conditions of the physical environment. In these cases, the reward model of each latent state can be either directly determined by the physical modeling of the environment or learned using simulation-based approaches. Once the reward models are available, they can be used online in order to learn the transition model of the system. Problems of these types can be found in the field of meteorology or the domain of resource allocation. We consider for example the case of electricity allocation in an urban area, where each arm corresponds to a specific configuration of energy allocation while the reward model corresponds to the optimality of the configuration. The choice of the optimal allocation needs to adapt to the underlying state of the system that is subject to several factors such as meteorological conditions, consumption habits of the inhabitants, seasonality.
>
> Another scenario where this assumption is realistic is in the field of recommender systems. Each user can have different preferences (latent state) and these preferences can be revealed while interacting with the environment. Realistically, they can change over time. The knowledge of the reward models is justified by the large availability of offline data that allows learning offline high-quality models of user behaviour.
>
> Another realistic scenario that we can consider is when a model of the environment has already been learned (perhaps both the reward and the transition models are available) but the dynamics of the system have changed. This includes various real-world scenarios where non-stationarity is present. If the non-stationarity impacts the transition model, previous knowledge of the reward model can be retained through the use of our approach, without requiring to estimate everything from scratch, as done instead with Spectral Decomposition approaches.
>
>
> ## Applications of UCB and TS Approaches
>
> **Q: Why go for explore-then-commit rather than approaches based UCB or Thompson sampling that usually enjoys much better regret? Are there any particular challenges with applying these approaches to this problem?**
>
> **A:** The choice of presenting an algorithm of the type Explore-Then-Commit is mainly due to the simplicity of the approach with respect to UCB or TS types of algorithms. Indeed, even if for the moment we are not able to provide a lower bound for the considered problem, **we strongly believe that this result cannot be further improved within the considered class of problems**. In the following, we motivate our claim by providing two main reasons.
>
> Ideally, in order to obtain a result in terms of regret that has better scaling with respect to time (as commonly happens with UCB or TS), we would need to take actions that jointly improve our model estimates and are able to exploit the already available knowledge. In our setting, this would mean that we should be able to acquire information while playing the belief-based policy that maximizes at each time step the quantity $\mu(a)^T \widehat{b}_t$ with $\widehat{b}_t$ being computed using the last estimate of the transition matrix $\widehat{\boldsymbol{P}}$. However, this is prevented by the following facts:

---

> ### Author Response · Authors · 2024-04-26
>
> 1. The first one is that our estimation algorithm (Algorithm 1) works under samples collected using a uniform (round-robin) strategy. Even if it can be adapted to work with stochastic policies having different probabilities of pulling each arm, the policies that can be used for estimation are those that are not conditioned on past observations. Given this fact, **we are not able to provide an estimate of the transition model using samples collected from a belief-based policy.
> This aspect is typical in the Partially Observable setting**, where estimation approaches mainly work with observation-independent policies as done for example in [1, 2] which employ Spectral Decomposition techniques with uniform policies. An exception to this case is represented by the work of [1] where SD approaches are extended to work with samples coming from memoryless policies, but still unable to work under belief-based ones.
> Hence, **we are required to split the problem between purely explorative and purely exploitative phases**. These approaches include algorithms of type Explore-Then-Commit (as our SL-EC algorithm) but also algorithms that alternatively switch between explorative and exploitative phases (as the SEEU algorithm employed in [2]).
> **However, both types of algorithms typically suffer a regret of order $\mathcal{O}(T^{2/3})$ with respect to time $T$**. The previous considerations have led us to opt for a simple type of algorithm based on the Explore-Then-Commit approach.
>
> 2. **The second reason is the most relevant in our opinion and is mainly due to the identifiability condition reported in our Assumption 4.2.** This assumption basically states that the information coming from pulling all the arms allows for the identification of the latent states. In practice, we are always ensured that when all the arms are played, there is enough information to improve the model estimate.
> Let's now assume that there exists an estimation algorithm that is stronger than our Algorithm 1 and can make estimates of the transition model even if samples are collected using a belief-based policy. Even in this case, under Assumption 4.2, we would not be able to provide better regret guarantees. This is because there exist instances in this class of problems where highly rewarding arms (potentially selected by the belief-based policy) may not provide any information at all to improve the model estimate. For these instances, **a forced exploration phase is needed** in order to acquire samples that improve the estimation of the model.
> In practice, **the challenge here lies in the fact that we are not guaranteed to acquire information at each step $t$ of interaction and this aspect increases the difficulty of the problem**. This feature is also crucial in the (more challenging) POMDP setting with $\alpha$-weakly revealing observations. In the work of [3] it is indeed shown that when it is possible to acquire information at each step (the setting is named undercomplete) it is possible to devise an algorithm with regret $\mathcal{O}(\sqrt{T})$, while in the more challenging overcomplete (where you are guaranteed to acquire information only from multiple steps, which translates to our scenario), a forced exploration phase is induced in the devised algorithm, thus reaching a regret of order $\mathcal{O}(T^{2/3})$.
> Though the POMDP setting is different from the one we consider, the one-step identifiability problem is present both in our setting and in the overcomplete one. Since a lower bound of order $\mathcal{O}(T^{2/3})$ has been proved for the overcomplete case [4], we strongly believe that a similar dependence with respect to time holds as well under our assumptions.
>
> Dealing more specifically with approaches based on TS, to the best of our knowledge, there are no approaches that are able to directly estimate the parameters of the model using a TS-based policy. In particular, we provide two example:
> - **the work of [5] develops a TS-based policy under the assumption of having a model that can be estimated offline using classical SD techniques** (see their discussion after Theorem 2 reported in their paper);
>
> - another interesting work using TS approaches is the one of [6] which tackles the more challenging POMDP setting (to which our setting can be mapped if we assume that the transition matrices of each action in the POMDP are equal and correspond to our MC). They are not able to use TS approaches to estimate the parameters of the model. Indeed, **the theoretical results they obtain hold under the assumption of having a consistent estimator of the transition model, which is not directly obtained with the approach they present** (see their Assumption 4).
>
> Considering this aspect, our approach improves over these works since we show a provably consistent estimator of the transition model, without explicitly requiring an assumption on its consistency.

---

> ### Author Response · Authors · 2024-04-26
>
> To conclude, there are no techniques based on TS policies that are able to provide guarantees over the parameters of the model and this aspect would again lead to the need for purely explorative phases, thus resorting to the considerations made in point 1.
>
> ## Dependency on $\sigma_{\min}$ and $\pi_{\min}$ and Full Estimation of the Transition Model
> **Q: Both $\sigma_{\min}$ and $\pi_{\min}$ can be very small in practice and my understanding is that this could make the bound in Thm 6.1 vacuous. This since it could happen that $T_0 > T$ even for moderately large $T$. It seems like this undesirable scaling is due to the fact that we try to perfectly estimate $P$ and do we really need to do that to have good regret?**
>
> **A:** The need to perfectly estimate the transition model $\boldsymbol{P}$ is a common characteristic in the infinite-horizon Partially Observable setting. Indeed, the available estimation procedures (Spectral-based approaches and our approach as well) only provide results in terms of the Frobenious norm or operator norm of the estimation error of the **entire transition model**. This aspect can be observed in [1, 2, 7, 8]. They all provide estimation guarantees that depend on the minimum state probability $\pi_{\min}$. This term is unavoidable and is mainly due to the partially observable nature of the problem since the fact that the state cannot be directly observed prevents defining error bounds for each row (state) of the transition matrix, as instead happens in the fully observable setting (see for example [9]). The dependence on $\pi_{\min}$ encodes the higher complexity of the partially observable setting with respect to the fully observable one and is used to move from the bound per row achievable in the fully observable setting to the bound for the whole matrix typical of the partially observable one.
>
> Regarding the dependency on $\sigma_{\min}$, it appears as well in all estimation problems with partial observability [1,2,3,4,7,8,10,11]. It is related to the identifiability of the different states given the information provided by the observations. Low values of $\sigma_{\min}$ make the estimation problem more difficult, while the nullity of this term defines instances where some states cannot be distinguished from one another, thus making standard estimation approaches fail. **The relevance of this dependence has led to the definition of a new tractable class of problems in the partially observable setting, called $\alpha$-weakly revealing, where $\alpha$ represents a lower bound to the minimum singular value $\sigma_{\min}$**. Furthermore, lower bounds in terms of regret have been provided for the revealing POMDP setting, showing that the dependency on the identifiability of the states represented by the quantity $\sigma_{\min}$ is unavoidable.
>
> Having discussed the dependence of these two terms, we refer the reviewer to Appendix C presenting a comparison between standard Spectral Decomposition techniques and our estimation approach. In particular, we show how using the knowledge of the observation model we improve the order of dependence on all the problem parameters, thus including $\sigma_{\min}$ and $\pi_{\min}$.
>
> ## Assumption of Starting from the Stationary Distribution
>
> **Q: The regret bound holds only under the assumption that the chain starts from its stationary distribution. The authors state that similar results can be shown without this assumption. Why not have this stronger result in the paper?**
>
> **A:** We opted for this assumption since the initial distribution of the chain easily approaches the stationary one when a few steps of the chain occur. Indeed, since we assume that the chain is ergodic, it is able to reach its stationary distribution geometrically fast. As specified in the main paper, results considering starting from a distribution that is different from the initial one are already available for ergodic chains and can be adaped to our setting.
>
> We will certainly provide more details on this aspect and define theoretical guarantees for the more general case in a revised version of the work. Thank you for the suggestion.

---

> ### Author Response · Authors · 2024-04-26
>
> ## Empirical Evaluation
>
> **Q.1: SL-EC is not evaluated using the budget suggested by Eq 17 and instead uses scaled values. The amount of scaling is not presented in the paper. I understand that what is suggested by theory often is conservative in practice but having a version of SL-EC that uses the budget suggested by Eq 17 would help making sense of the theoretical results.**
>
> **A.1:** The choice of not using the value of $T_0$ prescribed by theory is mainly due to the fact that, as the reviewer states, the results suggested by the theory are typically highly conservative and the value of $T_0$ tends to have high values with respect to the total horizon $T$ when $T$ is not large enough. Choosing the value of $T_0$ suggested by Eq.17 would have required experiments with a horizon length in the order of $10^7-10^8$ which would have in turn required a large amount of time and computational resources for running the experiments.
>
> Indeed, for practical reasons, we chose to employ a smaller value for the exploration horizon $T_0$ which is the one suggested by Eq.17 divided by the quantity $(10L)^{2/3}$. This value allows showing the benefits of our estimation procedure avoiding having too long interaction phases.
>
> **We remark that the choice of scaling the values obtained by theory is common in the scientific literature and allows for preserving the theoretical guarantees. Indeed, this caveat mostly translates into bigger multiplicative constants in the final regret bound or in similar bounds but holding with smaller probability**. For our specific case, it can be shown that using the scaled value of $T_0$ leads to an incurred regret with a multiplicative constant that is $(10L)^{1/3}$ higher than the one presented in Theorem 6.1.
>
> This topic is also discussed in [1] (see their Remark 3) when computing the optimistic POMDP model and is applied as well in the experimental sections of [1] and [2].
>
> If the motivations mentioned above have not fully addressed the concerns of the reviewer, we will certainly provide the experiments with values of $T_0$ suggested by Eq.17 in a revised version of the work.
>
> **Q.2: I suggest that the authors also plot the regret over several horizons, i.e $R(T)$, to illustrate the dependence in $T$ (since the budget $T_0$ is a function of $T$). This would tell us for which horizons the SL-EC has an advantage over baselines.**
>
> **A.2:** As suggested by the reviewer, we run experiments using different horizons. For the reasons mentioned above, we adopted the value of $T_0$ scaled by the quantity $(10L)^{2/3}$. We recall that the length of the exploration horizon depends on the characteristics of the specific bandit instance considered. There are instances where the benefits of our approach can be observed in smaller horizons since short exploration phases can suffice when states are easily identifiable (high $\sigma_{\min}$) and viceversa, when the problem is more complex, the benefits are more evident in the long run.
>
> In addition, we executed another set of experiments showing the efficacy of the SL-EC algorithm even when the exploration horizon is much smaller than the one suggested by theory. We manually set the exploration horizon over different runs and tested it on a Switching Bandit instance with $S=4$ states, $I=8$ actions and $V=5$ observations. The mentioned experiments are reported in the document that can be found in the last comment.
>
> **Q.3: The confidence intervals in the empirical section looks too good to be true given the small number of seeds.**
>
> **A.3:** The low values for the confidence intervals can be attributed to different factors. One of them is the intrinsic stochasticity induced by the specific instance of Switching Latent Bandit used in the experiment, some instances may indeed induce more stochasticity in the process than others and this may lead to larger confidence intervals. Another relevant aspect is the fact that the choice of including in the plot the regrets of algorithms having worse performances, such as SW-UCB, $\epsilon$-greedy or Exp3.S results in having a wide range of values to cover along the y-axis, that lead to a bad visualization of the confidence intervals. Indeed, this phenomenon is mitigated in the experiments performed on the different interaction horizons (see Figure 1.2 in the linked document).
> In the experiments reported we increased the number of runs to 20.
>
> ## Link to the document: [Document with Experiments](https://docs.google.com/document/d/e/2PACX-1vSsfTc0p6DJEKOOhOfS1DALUXjeYHyhw19PhZruOfeGi2hgqFBJyqolai8AnR8XKnJPEYqKeAEJDgvR/pub)

---

> ### Author Response · Authors · 2024-04-26
>
> ## References
>
> [1] Kamyar Azizzadenesheli, Alessandro Lazaric, and Anima Anandkumar. Reinforcement learning of pomdps using spectral methods. In Annual Conference Computational Learning Theory, 2016.
>
> [2] Xiang Zhou, Yi Xiong, Ningyuan Chen, and Xuefeng Gao. Regime switching bandits. In A. Beygelzimer, Y. Dauphin, P. Liang, and J. Wortman Vaughan (eds.), Advances in Neural Information Processing Systems, 2021.
>
> [3] Qinghua Liu, Alan Chung, Csaba Szepesvari, and Chi Jin. When is partially observable reinforcement learning not scary? In Proceedings of Thirty Fifth Conference on Learning Theory, volume 178, pages 5175–5220. PMLR, 02–05 Jul 2022.
>
> [4] Chen, F., Wang, H., Xiong, C., Mei, S., & Bai, Y. (2023, July). Lower bounds for learning in revealing pomdps. In International Conference on Machine Learning (pp. 5104-5161). PMLR.
>
> [5] Joey Hong, Branislav Kveton, Manzil Zaheer, Yinlam Chow, Amr Ahmed, Mohammad Ghavamzadeh, and Craig Boutilier. Non-stationary latent bandits. CoRR, abs/2012.00386, 2020.
>
> [6] Mehdi Jafarnia Jahromi, Rahul Jain, and Ashutosh Nayyar. Online learning for unknown partially observable mdps. In Proceedings of The 25th International Conference on Artificial Intelligence and Statistics, 2022.
>
> [7] Animashree Anandkumar, Rong Ge, Daniel Hsu, Sham M. Kakade, and Matus Telgarsky. Tensor decompositions for learning latent variable models. J. Mach. Learn. Res., 15(1):2773–2832, jan 2014.
>
> [8] Yi Xiong, Ningyuan Chen, Xuefeng Gao, and Xiang Zhou. Sublinear regret for learning pomdps, 2022.
>
> [9] T. Jaksch, R. Ortner, and P. Auer. Near-optimal regret bounds for reinforcement learning. J. Mach. Learn. Res., 11:1563–1600, aug 2010.
>
> [10] Chi Jin, Sham M Kakade, Akshay Krishnamurthy, and Qinghua Liu. Sample-efficient reinforcement learning of undercomplete POMDPs. NeurIPS, 2020.
>
> [11] Zhaohan Daniel Guo, Shayan Doroudi, and Emma Brunskill. A PAC RL algorithm for episodic
> POMDPs. In Artificial Intelligence and Statistics, pages 510–518. PMLR, 2016.

---

> > ### Comment · Reviewer_vJgb · 2024-06-17
> > **A few comments.**
> >
> > Thank you for your comments.
> >
> >  I have a few more comments and questions.
> > - The new experiments attached look good and I think it makes sense to have them in a revised version. The same goes for the table that helps position the work relative to other work.
> > - Thank you for clarifying the choice of $T_0$.  This scaling should probably be mentioned in the main paper.
> > - In the definition of the regret (eq:6) is the $\max_a$ in the second term correct? An agent doesn’t necessarily play the argmax at every step (e.g. UCB and TS) and this regret definition seems specific to your explore-then-commit strategy.
> > - I agree with reviewer UHGH’s comments regarding the sub-optimality of uniform exploration and I think the paper would benefit from discussing it. There is some work on pure-exploration in the latent bandit model that the authors might find useful (see Kinyanjui et al. 2023).
> >
> >
> >
> > Reference:
> >
> > Fast Treatment Personalization with Latent Bandits in Fixed-Confidence Pure Exploration,Newton Mwai Kinyanjui and Emil Carlsson and Fredrik D. Johansson, Transactions on Machine Learning Research (TMLR) 2023.

---

> ### Author Response · Authors · 2024-06-17
>
> We thank the reviewer for their response and time spent in reviewing our manuscript. Regarding the first two points, we are happy that the provided answers addressed the concern of the reviewer, we will certainly include them in a revised version of the work together with other useful points highlighted during the discussion period.
>
> ## Regret Definition
> Concerning Equation 6, its formulation is correct since we compare against the oracle maximizing the instantaneous expected reward at each step, (the one which pulls $ \arg \max_{a \in \mathbb{A}} \langle \boldsymbol{\mu}(a), \boldsymbol{b}_{t} \rangle $). We name now this policy as the "Optimal Belief-based Action" and we will refer to it in these terms in the comparison table (by revising the table, we noticed that we wrongly reported this aspect along the "Oracle" row, and we have now fixed it).
> We would like to highlight two main aspects:
> 1. The optimal policy for this setting is the optimal POMDP policy (since our problem can easily be cast to a POMDP, as shown in [2]), which is different from the "Optimal Belief-based Action" we chose. However, **computing the optimal POMDP policy given the model is computationally intractable, and this aspect has led us to the choice of an oracle strategy as the one defined in Eq.5**, which is instead much easier to compute.
> 2. The regret order we suffer with our policy when compared against our oracle (Eq. 6) is $\widetilde{\mathcal{O}}(T^{2/3})$. We would like to highlight that **if instead we had used the optimal POMDP policy computed on the model estimated through Algorithm 1 and if we had compared it against the optimal POMDP policy** (see Equation 1 in [2] for the precise regret definition), **we would as well have suffered a regret of order $\widetilde{\mathcal{O}}(T^{2/3}) $ with respect to time**. Indeed, without diving too much into the details, the regret analysis in both cases would boil down to upper bounding a term of type $ T_0 + \sum_{t=T_0+1}^T \|\|\boldsymbol{\mu}(a)\|\|_\infty \|\|\boldsymbol{b}_t - \widehat{\boldsymbol{b}}_t\|\|_1$ (the regret analysis in [2] can be reduced to a similar form) where the error in the belief is bounded by the relation controlling the belief error (reported in Proposition D.3). If requested, we will include more details about this aspect in a revised version of the work.
>
> ## Sub-optimality of the Uniform Exploration Approach
> We thank the reviewer for pointing out the work of (Kinyanjui et al. 2023). The main issue with techniques like the one in (Kinyanjui et al. 2023) is their strong dependency on the stationary setting; it is indeed not immediately clear how they can be adapted to a non-stationary latent model as the one we consider. Furthermore, algorithms of this type update the action selection probability at each step (when new information is acquired): **in our setting, updating the action probability at each step modifies the induced stationary distribution, and this aspect can cause problems in the estimation procedure.**
>
> In our scenario, it is indeed relevant to select a priori an action policy and keep it constant during the interaction with the environment (as done for example by our Algorithm 1 with the proposed Offline Arm Selection procedure or as also described in the "First Case" and "Second case" methods reported in the answer to Reviewer UHGH). Other approaches that establish a fixed policy a priori are those based on optimal experimental design [12]; however, it is not clear how they can be used in this setting. Furthermore, we highlight that improving the estimation procedure with such techniques would not affect the order of the regret with respect to time (for the reasons discussed in the answer "Applications of UCB and TS Approaches" provided above).
> As suggested by the reviewer, we will discuss the presented aspect in a revised version of the manuscript.
>
>
> ## References
> [12]   J. Kiefer and J. Wolfowitz. The equivalence of two extremum problems. Canadian
> Journal of Mathematics, 12(5):363–365, 1960.

---

> > ### Comment · Reviewer_vJgb · 2024-06-24
> >
> > Thank you for the clarification. I have no further comments at this point.

---

### Review · Reviewer_yNot · 2024-06-03

**Summary Of Contributions:**

This paper introduces a novel approach to address the switching latent bandits problem, where an agent must solve multiple multi-armed bandit (MAB) problems evolving according to an unknown underlying Markov chain. The proposed explore-then-commit algorithm effectively handles this challenge by estimating the transition matrix during exploration and selecting actions optimally based on the believed latent state during exploitation. The theoretical analysis establishes a regret bound of $O(T^{2/3})$, demonstrating the algorithm's competitiveness compared to existing approaches. Empirical evaluations validate its effectiveness through numerical simulations and real data experiments, showcasing superior performance over alternative methods.

**Audience:**

Yes

**Broader Impact Concerns:**

The work is predominantly theoretical, I do not see any specific broader impact concerns.

**Claims And Evidence:**

Yes

**Requested Changes:**

Please considered the detailed feedback provided under Weaknesses above, specifically the concerns around regret optimality, modeling assumptions, novelty against existing techniques (with a table), and justifying motivations.

**Strengths And Weaknesses:**

Strength:

The problem setup is interesting, and the paper is well-structured. The writing is comprehensive, and notations are clear. Experiments are reported on synthetic and real data.

Weaknesses:

Algorithm and Regret Bound: I am concerned about the $O(T^{2/3})$ regret guarantee, which is standard for Explore and Commit algorithms, and the analysis is straightforward. I am missing the novelty of the algorithmic ideas and analysis techniques, especially given the work is predominantly theoretical. I am wondering why the authors did not try a UCRL type approach (UCB-based method) to achieve $O(\sqrt{T})$ bound? This appears to be a major limitation of the work. In fact, few related works, e.g., Azizzadenesheli et al. (2016) achieved $O(\sqrt{T})$ regret, and it is unclear the contribution of the current work over these related works.

It would be beneficial to have a table listing all the related works, their modeling assumptions, objectives, and performance guarantees to understand the precise contribution of this work.

Dependencies on the problem parameters: In the final regret bound of Theorem 6.1, the dependency of the problem-dependent parameters, e.g., $\epsilon$, $\sigma_{\min}$, $\lambda$, etc., requires more discussion. Are these dependencies tight? For example, I am surprised by the $O(1/\epsilon^2)$ dependency; why do we need to explore all $P(s,s')$, especially if the rewards associated with these states are small? Seems non-intuitive. Some discussions on the tightness of the derived regret bounds would be useful.

Assumptions: Following from the above point, it will also be important to understand the necessity of the assumptions, e.g., Assumptions 4.1 and 4.2 --- can they be relaxed if the dependencies on the parameters are not tight? Also, how practical are these assumptions for real-world problems? Can you give a real-world problem scenario where such examples are satisfied?

In general, the motivation of the setting needs to be explained better; the one-paragraph discussion on the financial market and online advertising is not enough to understand the exact mapping of the problem framework to these practical examples; a better and more detailed explanation would be useful.

Minor: The citation Kwon et al. (2022) seems to have typos. Check the formatting of the bibliography.

---

> ### Author Response · Authors · 2024-06-06
>
> We thank the reviewer for their time in reviewing our manuscript. In the following, we will address the main points raised in the comments.
>
> ## Table comparing with Related Works
> By following the suggestion provided by the reviewer, we defined a table highlighting the main differences with respect to Related Works. The table can be found in the last section of the document at the following link:
> [Document with Comparison Table](https://docs.google.com/document/d/e/2PACX-1vSsfTc0p6DJEKOOhOfS1DALUXjeYHyhw19PhZruOfeGi2hgqFBJyqolai8AnR8XKnJPEYqKeAEJDgvR/pub).
>
> ## Choice of Explore-Then-Commit Algorithm
> The choice of an Explore-Then-Commit type of algorithm is mainly because UCB and TS approaches are not able to improve the achieved regret result. Indeed, since we strongly believe that this result cannot be further improved within the considered class of problems, we opted for a simpler EC algorithm. In the following, we motivate our claim by providing two main reasons.
>
> Ideally, in order to obtain a better result in terms of regret (as happens with more sophisticated algorithms such as UCB or TS), we would need to take actions that exploit the already available knowledge while also improving the estimate of the model. In our setting, this would mean that we should be able to acquire information while playing the belief-based policy that maximizes at each time step the quantity $\mu(a)^T \widehat{b}_t$ (with $\widehat{b}_t$ being computed using the last estimate $\widehat{\boldsymbol{P}}$ of the transition matrix). However, this aspect is prevented by the following facts:
>
> 1. The first one is that our estimation algorithm (Algorithm 1) works under samples collected using a uniform strategy, thus not conditioned on past observations. Given this fact, **we are not able to provide an estimate of the transition model using samples collected from a belief-based policy**, being it a function of past observations. However, **this is a common problem in the Partially Observable setting** since most estimation approaches work with observation-independent policies as done for example in [1, 2] which adopt Spectral Decomposition techniques by employing uniform policies for data collection. An exception to this case is represented by the work of [1] where SD approaches are extended to work with samples coming from memoryless policies, but still unable to work under belief-based ones.
> From the previous consideration, it turns out that **in order to estimate the model parameters, we are required to split the problem into purely explorative phases (uniform policies) and purely exploitative ones (belief-based policy)**. Approaches of this type include algorithms of type Explore-Then-Commit (as our SL-EC algorithm) but also algorithms that alternatively switch between explorative and exploitative phases (as the SEEU algorithm employed in [2]). Since both types of algorithms typically suffer a regret of order $\mathcal{O}(T^{2/3})$ (see [2]), we opted for the simpler Explore-Then-Commit approach for this work.
>
> 2. The second reason is the most relevant, mainly due to the identifiability condition reported in Assumption 4.2. This assumption states that **the entire set of arms contains information for identifying the latent states**. However, this condition also includes cases where only a specific subset of arms may be informative for determining the latent states, while the remaining ones may not.
> Let's now assume that there exists an estimation algorithm that is stronger than our Algorithm 1 and can make estimates of the transition model even if samples are collected using a belief-based policy. Even in this case, under Assumption 4.2, we would not be able to provide better regret guarantees. This is because this class of problems includes worst-case instances where **we are guaranteed to acquire information only by pulling all the available arms**. Using instead a belief-based policy which selects arms based on their associated reward may lead to exclude actions that are necessary for model estimation. The same problem appears in a slightly different form in the more challenging POMDP setting with $\alpha$-weakly revealing observations. In [3], the authors show that when it is possible to acquire information at each step (undercomplete setting) it is possible to devise an algorithm with regret $\mathcal{O}(\sqrt{T})$. In contrast, in the more challenging overcomplete setting where you are guaranteed to acquire information only from multiple steps (this case can be translated to our scenario), a forced exploration phase is induced in the devised algorithm, having the effect of leading to a regret of order $\mathcal{O}(T^{2/3})$.
> Since the one-step identifiability problem is present both in our setting and in the overcomplete one, and a lower bound of order $\mathcal{O}(T^{2/3})$ has been proved for the overcomplete case [4], we strongly believe that a similar dependence with respect to time holds as well in our setting.

---

> ### Author Response · Authors · 2024-06-06
>
> ## Inconsistency of TS-based Approaches
>
> Dealing more specifically with approaches based on TS, to the best of our knowledge, there are no approaches that are able to directly estimate the parameters of the model using a TS-based policy. In particular, we refer to two works that share similarities with our setting:
>
> 1. The work of [5] tackles a similar switching Latent Bandit setting and considers having a prior of both the observation and the transition model. The computation of the prior relies on offline model estimation using classical SD techniques (see their discussion after Theorem 2 reported in their paper). The algorithm they provide is computationally inefficient since it makes a Bayesian update of the model parameters and is not able to guarantee a consistent estimation of the model.
>
> 2. The interesting work of [6] employs TS approaches in the more challenging POMDP setting under the assumption of knowing the observation model. Our setting can be easily mapped to theirs if we assume that the transition matrices of each action in the POMDP are equal and correspond to our Markov Chain. They use a TS policy to collect samples and estimate a model from these samples. However, **in order to provide regret guarantees, they assume that the Bayesian update of the model provides a consistent estimator (see their Assumption 4)**. However, this condition holds in the fully-observable setting and has not been verified for partially-observable ones.

---

> > ### Author Response · Authors · 2024-06-06
> >
> > ## Dependencies on the problem Parameters
> >
> > 1. **Dependency on $\epsilon$**: The dependency on this term derives from the $L$ term appearing in Theorem 6.1 which is itself **derived from a result from [12] that bounds the 1-norm error in the estimated belief $\|b - \widehat{b}\|_1$ with the error in the estimated model $\|\boldsymbol{P} - \widehat{\boldsymbol{P}}\|_F$.** This result holds when all the elements of the transition matrix are positive and this fact made us introduce Assumption 4.1. This result on the belief error (see Proposition D.3) has been used for the same purpose in [1,8,13] and has been adapted to our setting in Proposition D.4. We believe that this dependency can be improved (or even removed), however the only known result that allows bounding the error in the estimated belief is the one presented in Propositions D.3, which contains the dependency on the $\epsilon$ term.
> >    **We would like to highlight that the dependency on $\epsilon$ does not appear in Lemma 6.1** providing theoretical guarantees for the estimation of the transition model (Algorithm 1); as a consequence, Lemma 6.1 does not need Assumption 1 to hold but only requires the ergodicity of the chain.
> >
> > 2. **Dependency on $\pi_{\min}$**: The dependency on the minimum state stationary distribution $\pi_{\min}$ is common in the infinite-horizon Partially Observable setting and in HMM estimation problems. This aspect can be observed in [1, 2, 7, 8] where all theoretical results have a dependency on $\pi_{\min}$. This term is unavoidable and is mainly due to the partially observable nature of the problem since the fact that the state cannot be directly observed prevents defining error bounds for each row (state) of the transition matrix, as instead happens in the fully observable setting (see for example [9]). **The dependence on $\pi_{\min}$ encodes the higher complexity of the partially observable setting with respect to the fully observable one** and is used to move from the bound per row achievable in the fully observable setting to the bound for the whole matrix typical of the partially observable one. Lemma 6.1 shows a scaling of order $\mathcal{O}(1/\pi_{\min})$ of our estimation algorithm, which improves over the Spectral approaches whose scaling has order $\mathcal{O}(1/\pi_{\min}^{7/2})$.
> >
> > 3. **Dependency on $\sigma_{\min}(\boldsymbol{O})$**: Assumption 4.2 requires the action observation matrix to be full-column rank, which means that $\sigma_{\min}(\boldsymbol{O}) > 0$. This is a standard assumption used in these types of problems and appears in all estimation problems with partial observability [1,2,3,4,7,8,10,11]. It is related to the identifiability of the different states given the information provided by the observations. Low values of $\sigma_{\min}(\boldsymbol{O})$ make the estimation problem more difficult, while the nullity of this term defines instances where some states cannot be distinguished from one another, thus making standard estimation approaches fail.
> >    The relevance of this dependence has led to the definition of a new tractable class of problems in the partially observable setting, called $\alpha$-weakly revealing, where $\alpha$ represents a lower bound to the minimum singular value $\sigma_{\min}(\boldsymbol{O})$ of the observation matrix. Furthermore, lower bounds in terms of regret have been provided for the revealing POMDP setting [4], showing that **the dependency on the identifiability of the states represented by the quantity $\sigma_{\min}(\boldsymbol{O})$ is unavoidable**.
> >
> > 4. **Dependency on $\lambda$**: The $\lambda$ term appearing in the formulation represents the modulus of the second largest eigenvalue and derives from the fact that observed samples are not independent but come from a Markov Chain. The dependency on this term is thus unavoidable and we believe it to be tight for the considered setting. It derives from recent results from the work of [14] which improve the existing concentration results of samples coming from Markov Chains. Indeed, differently from standard results on Markov chain estimation which have a dependency of order $\mathcal{O}(1 / (1 - \lambda))$ (see [1]), Lemma 6.1 shows that our approach scales with an order of $\mathcal{O}(\sqrt{1 / (1 - \lambda)})$, which is further improved by the exponent $2I^2$ to the $\lambda$ due to the adopted round-robin policy (for details on the exponential term you might refer to the proof of Lemma 6.1 in Appendix B.1).
> >
> > **We refer the reviewer to Appendix C for a thorough comparison between standard Spectral Decomposition techniques and our estimation approach**. In particular, we show how using the knowledge of the observation model we improve the order of dependence on all the problem parameters, thus including $\sigma_{\min}$ and $\pi_{\min}$.

---

> > > ### Author Response · Authors · 2024-06-06
> > >
> > > ## Difference with Azizzadenesheli et al. 2016 ([1])
> > >
> > > There are several differences with respect to the work of Azizzadenesheli et al. (2016) ([1]). Mainly:
> > >
> > > 1. They consider the more classical POMDP setting without knowledge of the observation or transition model and use SD approaches to estimate both models. Differently, in our work, we assume a Switching Latent Bandit setting and devise a new estimation technique tailored for learning the transition model while exploiting the knowledge of the observation model.
> > >
> > > 2. As can be seen in the Table, **the regret in [1] is defined with respect to the optimal memoryless stochastic policy, which is a weaker oracle than the optimal deterministic belief-based policy we use**. The set of memoryless policies is less powerful than the set of belief-based policies since the former ones are Markovian with respect to the last observation received. Furthermore, the oracle is *stochastic* since **the set of considered policies have a minimum probability of taking each action**. This aspect favors model estimation and is fundamental to applying Spectral techniques and obtaining a regret of order $\mathcal{O}(\sqrt{T})$.
> > >
> > > 3. While we use an EC algorithm, an optimistic approach is used in [1] with each episode being characterized by an increasing length. The optimistic approach is supported by the fact that the used policies are exploratory enough to allow state identification since every action is constantly chosen with a minimum probability.
> > >
> > > 4. Concerning the assumptions, the minimum transition probability assumption (our Assumption 4.1) is not needed in [1] since, being their policies memoryless, no notion of belief state is employed. In this way, the result of [12] on the bound of the error of the estimated belief is not used. Concerning the identifiability assumption, they also have a full-rank assumption on the observation matrix, which is required by Spectral Decomposition techniques. In addition, they require the transition matrix to be invertible, as a necessary condition to apply SD approaches.
> > >
> > > ## Necessity of Assumptions
> > >
> > > Following the discussion developed in the points above, it turns out that **Assumption 4.2 is necessary for our setting since otherwise state identification is not possible**. As highlighted above, it is used (either in the $\alpha$-revealing or full-rank version) in all problems providing estimation guarantees in the partially observable setting. The dependency on the minimum singular value $\sigma_{\min}(\boldsymbol{O})$ appears as well in the formulation of the lower bounds for the settings in which the difficulty of the problem has been characterized, as shown in [4].
> > >
> > > Concerning instead Assumption 4.1, as stated above, it is a technical assumption used to bound the estimation error of the belief vector employed during the exploitation phase of the SL-EC algorithm.
> > > We recall that this assumption is not necessary for the estimation algorithm (Algorithm 1), as shown in Lemma 6.1. Concerning instead the SL-EC algorithm, as discussed in the point on the *Dependency on $\epsilon$*, this assumption could likely be avoided, however no other results bounding the error of the estimated belief vector with the error of the estimated model are currently available than the one of [12], which requires the use of Assumption 4.1.

---

> > > > ### Author Response · Authors · 2024-06-06
> > > >
> > > > ## Real-world problems
> > > >
> > > > There are different scenarios where the two Assumptions we use can be satisfied. First of all, Assumption 4.1 on the minimum transition probability ensures that the model does not completely rule out any transitions, which can be beneficial for capturing the inherent randomness or rare events in the data. Some examples are:
> > > >
> > > > - **Weather Prediction**: the hidden states could represent different weather types (e.g., sunny, rainy, cloudy). Even though some transitions are more likely, any weather transition is possible due to the unpredictability of these patterns.
> > > >
> > > > - **Transportation Systems**: such as traffic flows where the hidden states represent different traffic conditions (e.g., free-flowing, congested, stopped). These patterns can change unexpectedly due to accidents, weather conditions, or external events.
> > > >
> > > > Concerning Assumption 4.2, it is likely to be satisfied when the observation space is rich with respect to the dimensionality of the state space. Furthermore, we recall that the weakly-revealing assumption includes as well special classes of observation models where each observation is generated only by one latent state (in the POMDP setting they are referred to as Block MDPs): they model different scenarios such as navigation tasks and image-based robotics tasks.
> > > >
> > > > An example where both assumptions are satisfied can be:
> > > >
> > > > - **A robot navigating in an environment**: The hidden states could be represented by distinct locations or regions within the environment. We can assume that the robot is equipped with sensors of different (known) qualities and having diverse (known) properties, thus providing different types of signals (observations). The choice over which sensor to use at each time step is the action taken. The robot can potentially move from any location to any other location, even if some transitions are very unlikely (e.g., due to obstacles or distance). Including a minimum probability for all state transitions ensures the model can account for rare movements or errors in navigation, while the variability of the different sensors ensures the full-rank condition and the identifiability of the underlying latent states.
> > > >
> > > > ### More Details on the Financial Example
> > > >
> > > > Stock prices and other financial time series often exhibit patterns that can be attributed to different market conditions or "regimes." These regimes might include bull markets (characterized by rising prices), bear markets (characterized by falling prices), and periods of high or low volatility. The actual regime at any given time is not directly observable (hidden state), but we can infer it from observable data. In this example, the different actions available are the decisions whether to sell or buy different amounts of stocks and the observations can be the different returns, trading volumes, or stock prices.
> > > >
> > > > ### More Details on the Online Advertising Example
> > > >
> > > > In this example, the hidden states can be represented by the interests of users, which are not observable. The interests may vary according to seasonal patterns, new trends, or exogenous variables that modify the market, and we can model it as a Markov chain. The actions are the different types of content that can be displayed to the users, while the observations may be represented by factors such as conversions or interactions with the ads (metrics such as click-through rate).
> > > >
> > > >
> > > > ## Minor
> > > > We thank the reviewer for the spotted typo. We will certainly remove it in a revised version of the work.

---

> > > > > ### Author Response · Authors · 2024-06-06
> > > > >
> > > > > ## References
> > > > >
> > > > > [1] Kamyar Azizzadenesheli, Alessandro Lazaric, and Anima Anandkumar. Reinforcement learning of pomdps using spectral methods. In Annual Conference Computational Learning Theory, 2016.
> > > > >
> > > > > [2] Xiang Zhou, Yi Xiong, Ningyuan Chen, and Xuefeng Gao. Regime switching bandits. In A. Beygelzimer, Y. Dauphin, P. Liang, and J. Wortman Vaughan (eds.), Advances in Neural Information Processing Systems, 2021.
> > > > >
> > > > > [3] Qinghua Liu, Alan Chung, Csaba Szepesvari, and Chi Jin. When is partially observable reinforcement learning not scary? In Proceedings of Thirty Fifth Conference on Learning Theory, volume 178, pages 5175–5220. PMLR, 02–05 Jul 2022.
> > > > >
> > > > > [4] Chen, F., Wang, H., Xiong, C., Mei, S., & Bai, Y. (2023, July). Lower bounds for learning in revealing pomdps. In International Conference on Machine Learning (pp. 5104-5161). PMLR.
> > > > >
> > > > > [5] Joey Hong, Branislav Kveton, Manzil Zaheer, Yinlam Chow, Amr Ahmed, Mohammad Ghavamzadeh, and Craig Boutilier. Non-stationary latent bandits. CoRR, abs/2012.00386, 2020.
> > > > >
> > > > > [6] Mehdi Jafarnia Jahromi, Rahul Jain, and Ashutosh Nayyar. Online learning for unknown partially observable mdps. In Proceedings of The 25th International Conference on Artificial Intelligence and Statistics, 2022.
> > > > >
> > > > > [7] Animashree Anandkumar, Rong Ge, Daniel Hsu, Sham M. Kakade, and Matus Telgarsky. Tensor decompositions for learning latent variable models. J. Mach. Learn. Res., 15(1):2773–2832, jan 2014.
> > > > >
> > > > > [8] Yi Xiong, Ningyuan Chen, Xuefeng Gao, and Xiang Zhou. Sublinear regret for learning pomdps, 2022.
> > > > >
> > > > > [9] T. Jaksch, R. Ortner, and P. Auer. Near-optimal regret bounds for reinforcement learning. J. Mach. Learn. Res., 11:1563–1600, aug 2010.
> > > > >
> > > > > [10] Chi Jin, Sham M Kakade, Akshay Krishnamurthy, and Qinghua Liu. Sample-efficient reinforcement learning of undercomplete POMDPs. NeurIPS, 2020.
> > > > >
> > > > > [11] Zhaohan Daniel Guo, Shayan Doroudi, and Emma Brunskill. A PAC RL algorithm for episodic POMDPs. In Artificial Intelligence and Statistics, pages 510–518. PMLR, 2016.
> > > > >
> > > > > [12] Yohann De Castro, Elisabeth Gassiat, and Sylvain Le Corff. Consistent estimation of the filtering and marginal smoothing distributions in nonparametric hidden markov models. IEEE Transactions on Information Theory, 63(8):4758–4777, 2017.
> > > > >
> > > > > [13] Bowen Jiang, Bo Jiang, Jian Li, Tao Lin, Xinbing Wang, and Chenghu Zhou. Online restless bandits with unobserved states. In Proceedings of the 40th International Conference on Machine Learning, 2023.
> > > > >
> > > > > [14] Jianqing Fan, Bai Jiang, and Qiang Sun. Hoeffding’s inequality for general markov chains and its applications
> > > > > to statistical learning. Journal of Machine Learning Research, 22(139):1–35, 2021.

---

### Author Response · Authors · 2024-06-07
**Visibility manually changed to "Everyone"**

Dear all,

we write this comment since we noted that the visibility of the answers to the reviews was not automatically set to "Everyone" at the beginning of the discussion period, but we had to switch to the "Everyone" option manually. We just wanted to ensure that everybody got a notification about it and can read the answers.

Hoping for an interesting and engaging discussion, we thank you for your attention.

Best regards,
the Authors

---

### Author Response · Authors · 2024-06-21

We thank all reviewers for their constructive feedback. We have now uploaded a revised version of the work containing the suggested changes and the newly conducted experiments. The lines and sections added or modified are reported in red.

If the reviewers have additional questions or clarifications, we're happy to discuss more.

---

### Decision · Action_Editor_9HQP · 2024-08-07

**Recommendation:** Accept as is

**Comment:**

While initially this paper wasn't unanimously appreciated by the reviewers, the author response has convinced one of the reviewers to change their minds and eventually support acceptance. One reviewer remained skeptical about the significance of the results, and in particular the strength of the assumptions that the authors have made. While I very much share these concerns and believe that the strength of the results leaves much to be desired, the paper does live up to the TMLR criteria for publication, and I am thus recommending the paper for acceptance. I nevertheless encourage the authors to keep working on this problem until a more satisfactory result can be achieved.

**Audience:**

Some members of the community working on multi-armed bandit problems and RL theory will surely find this paper to be interesting.

**Claims And Evidence:**

All results reported in the paper are supported by rigorous proofs. Neither the reviewers or myself have identified any issues with the proofs. (Most of the concerns were concerning the significance of the results, which is an independent issue altogether.)